# A dynamic approach to three-dimensional radiative transfer in subkilometer-scale numerical weather prediction models: the dynamic TenStream solver v1.0

Richard Maier[1], Fabian Jakub[1], Claudia Emde[1,2], Mihail Manev[1], Aiko Voigt[3], and Bernhard Mayer[1]

[1]Meteorologisches Institut, Ludwig-Maximilians-Universität München, Munich, Germany
[2]Institut für Physik der Atmosphäre, Deutsches Zentrum für Luft- und Raumfahrt, Oberpfaffenhofen, Germany
[3]Institut für Meteorologie und Geophysik, Universität Wien, Vienna, Austria

**Correspondence:** Richard Maier (richard.maier@physik.uni-muenchen.de)

**Abstract.** The increasing resolution of numerical weather prediction models makes inter-column three-dimensional (3D) radiative transport more and more important. However, 3D radiative transfer solvers are still computationally expensive, largely preventing their use in operational weather forecasting. To address this issue, Jakub and Mayer (2015) developed the TenStream solver. It extends the well-established two-stream method to three dimensions by using ten instead of two streams to describe the transport of radiative energy through Earth's atmosphere. Building upon this method, this paper presents the dynamic TenStream solver, which provides a further acceleration of the original TenStream model. Compared to traditional solvers, this speed-up is achieved by utilizing two main concepts: First, radiation is not calculated from scratch every time the model is called. Instead, a time-stepping scheme is introduced to update the radiation field based on the result from the previous radiation time step. Secondly, the model is based on incomplete solves, performing just the first few steps of an iterative scheme towards convergence every time it is called. Essentially, the model thereby just uses the ingoing fluxes of a grid box to update its outgoing fluxes. Combined, these two approaches put radiative transfer much closer to the way advection in the dynamical core of an NWP model is handled, as both use previously calculated results to update their variables and thereby just require access to the neighboring values of an individual grid box, facilitating model parallelization. To demonstrate the feasibility of this new solver, we apply it to a precomputed shallow cumulus cloud time series and test its performance both in terms of speed and accuracy. In terms of speed, the dynamic TenStream solver is shown to be about three times slower than a traditional 1D $\delta$-Eddington approximation, but noticeably faster than currently available 3D solvers. To evaluate the accuracy of the dynamic TenStream solver, we compare its results, as well as calculations carried out by a 1D $\delta$-Eddington approximation and the original TenStream solver, to benchmark calculations performed with the 3D Monte Carlo solver MYSTIC. We demonstrate that on the grid box level, dynamic TenStream is able to calculate heating rates and net irradiances at domain boundaries very close to those obtained by the original TenStream solver, thus offering a much better representation of the MYSTIC benchmark than the 1D $\delta$-Eddington results. By calling the dynamic TenStream solver less frequently than the $\delta$-Eddington approximation, we furthermore show that our new solver produces significantly better results than a 1D $\delta$-Eddington approximation carried out with a similar computational demand. At these lower calling frequencies, however, the incomplete solves in the dynamic TenStream solver also lead to the build-up of a bias with time, which becomes larger the lower the calling frequency is.

# 1 Introduction

Sources and sinks of radiative energy in the atmosphere are the main drivers of both weather and climate. They are quantified by heating rates and net surface irradiances and are calculated using radiative transfer models, which describe the transport of radiative energy through Earth's atmosphere, ideally allowing for full three-dimensional (3D) transport of energy. Depending on scale, we can differentiate between two different regimes of 3D radiative transport: On the model grid scale, 3D radiative transfer allows for horizontal transport of energy between adjacent model columns, whereas on the sub-grid scale, it refers to the three-dimensional transport of radiative energy within a heterogeneous model grid box. The calculation of both of these effects is computationally expensive, largely preventing their representation in operational weather forecasting. This is why up to this date, numerical weather prediction (NWP) models still use one-dimensional (1D) independent column approximations (ICA), such as the Monte Carlo Independent Column Approximation (McICA; Pincus et al. (2003)) currently employed at both DWD and ECMWF (DWD, 2021; Hogan and Bozzo, 2018). These models assume that radiative transport between grid boxes only takes place in the vertical and neglect any horizontal transport of energy – both in between different model columns and within individual model grid boxes.

However, both of these effects have been shown to be important for the correct calculation of radiative transfer in the atmosphere. While sub-grid scale 3D effects primarily act at coarser resolutions, where an individual grid box incorporates both cloudy and clear-sky regions and should thus not be treated homogeneously, the increasing horizontal resolution of numerical weather prediction models makes inter-column radiative transfer more and more important (O'Hirok and Gautier, 2005). Especially in large eddy simulations (LES) with hectometer-scale resolutions, inter-column radiative transport has been shown to affect both the organization and development of clouds. Klinger et al. (2017) for example showed that its consideration in the thermal spectral range leads to systematically larger cooling and much stronger organizational effects than in simulations driven by 1D radiative transfer. In accordance with that, Jakub and Mayer (2017) demonstrated that inter-column 3D radiative transfer in the solar spectral range may lead to the formation of cloud streets that are to this extent not found in 1D simulations.

To account for these increasingly important effects, a lot of effort in recent years was put into making 3D radiative transfer models computationally more feasible. Targeted towards sub-grid scale 3D effects, the Speedy Algorithm for Radiative Transfer through Cloud Sides (SPARTACUS; Schäfer et al. (2016); Hogan et al. (2016)) for example provides a fast method to calculate 3D radiative effects at the resolutions of currently employed global atmospheric models. To this end, it introduces additional terms to the well-established two-stream scheme to account for the radiative transport between cloudy and clear regions inside an individual model column. On the other hand, a lot of work went into the speed-up of inter-column radiative transport at subkilometer-scale resolutions, where model grid boxes can be gradually treated homogeneously. A large group of these models simplifies the expensive angular part of 3D radiative transfer calculations by just using a discrete number of angles (e.g., Lovejoy et al., 1990; Gabriel et al., 1990; Davis et al., 1990). Most recently, the TenStream solver (Jakub and Mayer, 2015) built upon this idea. It is capable of calculating 3D radiative fluxes and heating rates in both the solar and the thermal spectral range. To do so, it extends the 1D two-stream formulation to ten streams to consider horizontal transport of energy. Besides the TenStream solver, the Neighboring Column Approximation (NCA; Klinger and Mayer (2016, 2020)) provides a fast analytical

method for calculating inter-column 3D heating rates in the thermal spectral range. For that purpose, it estimates cloud side effects by taking just the direct neighbors of a specific grid box into account. Apart from these two approaches, significant progress has also been made in accelerating highly accurate 3D Monte Carlo solvers for the use in LES models, with Veerman et al. (2022) for example speeding up the method through the use of graphics processing units (GPUs). This allowed them to perform LES simulations driven by a full Monte Carlo solver for the first time ever. However, despite all these efforts, all of these solvers are still too slow to be used operationally. For example, the GPU-accelerated Monte Carlo solver of Veerman et al. (2022) is at least 6.4 times slower than the two-stream model they compare it to. And even while specifically designed for the use in NWP models, the SPARTACUS model is still 5.8 times slower than the McICA paramerization currently used at ECMWF (Hogan and Bozzo, 2018). This high computational burden prohibits the use of all of these models in operational forecasting, especially given that radiation is already called far less often than the dynamical core of NWP models.

To address this high computational cost of current 3D solvers, this paper presents a first step towards a new, "dynamic" 3D radiative transfer model. Currently designed for the use at subkilometer-scale horizontal resolutions, where model grid boxes can be assumed to be homogeneous, this new, fully three-dimensional model is based on the TenStream solver. It accelerates inter-column 3D radiative transfer towards the speed of currently employed 1D solvers by utilizing two main concepts. First, the model does not calculate radiation from scratch every time it is called, but treats radiation more like dynamics by using a time-stepping scheme to update the radiation field based on the result from the previous time step. Secondly, the model is based on incomplete solves, performing just the first few steps towards convergence every time it is called.

A detailed description of this method can be found in Sect. 2 of this paper. In Sect. 3, we then introduce a precomputed LES shallow cumulus cloud time series and further methodology to assess the quality of the new dynamic TenStream solver both in terms of speed and accuracy, also considering different calling frequencies. To this end, we compare it to both a traditional 1D $\delta$-Eddington approximation, the original TenStream solver as well as a benchmark simulation provided by the 3D Monte Carlo solver MYSTIC (Mayer, 2009). The results of this evaluation are presented in Sect. 4. The paper ends with a summary and outlook given in Sect. 5.

## 2  Towards dynamic treatment of radiation

Our goal is to create a 3D radiative transfer solver that calculates radiative fluxes and heating rates at a significantly faster speed than other inter-column 3D solvers, while also delivering a noticeable improvement in terms of accuracy over currently employed 1D radiation schemes. Here, we explain the foundation and functionality of our newly developed dynamic TenStream solver, that aims to achieve these targets using a time-stepping scheme and incomplete solves.

### 2.1  The original TenStream model

We build upon the TenStream model (Jakub and Mayer, 2015), which extends the established two-stream formulation to three dimensions. Figure 1 shows the definition of its streams, i.e., radiative fluxes (in units of W), for a single rectangular grid box, with the indices $(i, j, k)$ indicating the position of the box in a Cartesian grid of size $N_x \cdot N_y \cdot N_z$.

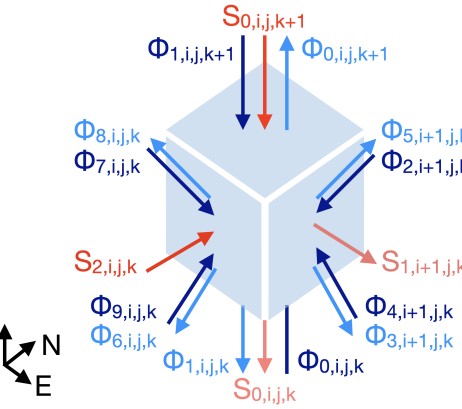

**Figure 1.** Schematic illustration of all fluxes entering and exiting a rectangular grid box $(i,j,k)$ in the TenStream solver and their respective indices. Diffuse fluxes are shown in blue, while fluxes of direct radiation are displayed in red. Fluxes entering the grid box are shown in a darker tone than the ones exiting. The two pairs of diffuse fluxes on each of the sideward oriented faces of the cuboid point into and out of the upper and lower hemispheres, respectively. Fluxes on the sides of the cuboid facing to the north and west are not visible.

Ten streams ($\Phi_0$, $\Phi_1$, ..., $\Phi_9$; depicted in blue) are used to describe the three-dimensional transport of diffuse radiation. As in the two-stream formulation, two of them ($\Phi_0$ (upward) and $\Phi_1$ (downward)) characterize the transport in the vertical, whereas four additional streams are introduced to describe the transport in each of the two additional horizontal dimensions. The transport of direct radiation, i.e., radiation originating from the Sun that has not yet interacted with the atmosphere, is treated separately using the three additional streams $S_0$, $S_1$ and $S_2$, one for each dimension (shown in red in Fig. 1). Using these streams, the radiative transport through a single grid box $(i,j,k)$ in the case of Sun shining from the south-west can be expressed by the following matrix equation:

$$
\underbrace{\begin{pmatrix} \Phi_{0,\,i\quad,\,j\quad,\,k+1} \\ \Phi_{1,\,i\quad,\,j\quad,\,k} \\ \vdots \\ \Phi_{9,\,i\quad,\,j+1,\,k} \\ S_{0,\,i\quad,\,j\quad,\,k} \\ S_{1,\,i+1,\,j\quad,\,k} \\ S_{2,\,i\quad,\,j+1,\,k} \end{pmatrix}}_{=\boldsymbol{\Phi}_{\mathrm{out},i,j,k}} = \underbrace{\begin{pmatrix} a_{00,i,j,k} & \cdots & a_{09,i,j,k} & b_{00,i,j,k} & b_{01,i,j,k} & b_{02,i,j,k} \\ a_{10,i,j,k} & \cdots & a_{19,i,j,k} & b_{10,i,j,k} & b_{11,i,j,k} & b_{12,i,j,k} \\ \vdots & & \vdots & \vdots & \vdots & \vdots \\ a_{90,i,j,k} & \cdots & a_{99,i,j,k} & b_{90,i,j,k} & b_{91,i,j,k} & b_{92,i,j,k} \\ 0 & \cdots & 0 & c_{00,i,j,k} & c_{01,i,j,k} & c_{02,i,j,k} \\ 0 & \cdots & 0 & c_{10,i,j,k} & c_{11,i,j,k} & c_{12,i,j,k} \\ 0 & \cdots & 0 & c_{20,i,j,k} & c_{12,i,j,k} & c_{22,i,j,k} \end{pmatrix}}_{=\boldsymbol{T}_{i,j,k}} \cdot \underbrace{\begin{pmatrix} \Phi_{0,\,i\quad,\,j\quad,\,k} \\ \Phi_{1,\,i\quad,\,j\quad,\,k+1} \\ \vdots \\ \Phi_{9,\,i\quad,\,j\quad,\,k} \\ S_{0,\,i\quad,\,j\quad,\,k+1} \\ S_{1,\,i\quad,\,j\quad,\,k} \\ S_{2,\,i\quad,\,j\quad,\,k} \end{pmatrix}}_{=\boldsymbol{\Phi}_{\mathrm{in},i,j,k}} + \underbrace{\begin{pmatrix} e_{0,i,j,k}\cdot B_{\mathrm{eff},0,i,j,k} \\ e_{1,i,j,k}\cdot B_{\mathrm{eff},1,i,j,k} \\ \vdots \\ e_{9,i,j,k}\cdot B_{\mathrm{eff},9,i,j,k} \\ 0 \\ 0 \\ 0 \end{pmatrix}}_{=\boldsymbol{B}_{i,j,k}} \tag{1}
$$

In there,

- the vector $\boldsymbol{\Phi}_{\mathrm{in},i,j,k}$ consists of all the radiative fluxes entering grid box $(i,j,k)$. For reasons of clarity, will use the expression $\Phi_{\mathrm{in},m,i,j,k}$ to address an individual entry $m$ of this vector, implying that for example $\Phi_{\mathrm{in},10,i,j,k}$ equals $S_{0,i,j,k+1}$ in case of Sun shining from the south-west.

– the matrix $\boldsymbol{T}_{i,j,k}$ describes the scattering and absorption of the ingoing radiation $\boldsymbol{\Phi}_{\mathrm{in},i,j,k}$ on its way through the grid box, with $a_{00,i,j,k}$ for example quantifying the fraction of the upward flux entering the grid box at the bottom ($\Phi_{0,i,j,k}$) that exits the box in the same direction through the top ($\Phi_{0,i,j,k+1}$). While the "a"-coefficients describe the transport of diffuse radiation, the "b"-coefficients quantify the fraction of direct radiation that gets scattered, thus providing a source term for the ten diffuse streams. The "c"-coefficients describe the amount of direct radiation that is transmitted through the grid box without interacting with the medium. All of these transport coefficients depend on the optical properties (optical thickness, single scattering albedo, asymmetry parameter, grid box aspect ratio and angle of solar incidence) of the particular grid box. They are precomputed using Monte Carlo methods and stored in look-up tables (Jakub and Mayer, 2015). We will use the expression $t_{mn,i,j,k}$ to refer to the entry in row $m$ and column $n$ of the full matrix $\boldsymbol{T}_{i,j,k}$.

– the vector $\boldsymbol{B}_{i,j,k}$ quantifies the amount of thermal radiation that is emitted into the direction of every one of the ten diffuse streams. Its entries $B_{m,i,j,k}$ are calculated by multiplying the black body radiation that is emitted into the corresponding direction ($B_{\mathrm{eff},m,i,j,k}$) by the emissivity of the grid box in that direction. According to Kirchhoff's law, this emissivity of a grid box into a certain direction is the same as the absorptivity of radiation coming from that direction, which in turn is one minus the transmittance in that direction. For example, the emissivity $e_{0,i,j,k}$ of grid box $(i,j,k)$ in upward direction is equal to the fraction of the downward facing radiative flux $\Phi_{1,i,j,k+1}$ that is absorbed on the way through that grid box, which in turn is one minus the sum of all fractions $a_{n1,i,j,k}$ of $\Phi_{1,i,j,k+1}$ exiting grid box $(i,j,k)$, i.e.,

$$e_{0,i,j,k} = 1 - \sum_{n=0}^{9} a_{n1,i,j,k}$$

where $a_{n1,i,j,k}$ refers to the corresponding entries in the second column of matrix $\boldsymbol{T}_{i,j,k}$.

– the vector $\boldsymbol{\Phi}_{\mathrm{out},i,j,k}$ consists of all radiative fluxes exiting the grid box $(i,j,k)$. For every stream, it contains all the radiative energy that has not interacted with the grid box on its way through, plus, in case of the diffuse streams, the radiative energy that has been scattered and emitted into that direction along that way. Similar to the ingoing flux vector, we use the expression $\Phi_{\mathrm{out},m,i,j,k}$ to refer to an entry $m$ of the full vector $\boldsymbol{\Phi}_{\mathrm{out},i,j,k}$.

The combined equations for all the $N_x \cdot N_y \cdot N_z$ grid boxes make up a large system of coupled linear equations that must be provided with boundary conditions at the edges of the domain. At the top and bottom, these are determined by the incoming solar radiation on one side and by ground reflection and emission on the other:

$$S_{0,i,j,N_z+1} = E_0 \cdot \cos\theta_{\mathrm{inc}} \cdot \Delta x \cdot \Delta y \qquad \text{(incoming solar radiation at the top)}$$

$$\Phi_{0,i,j,0} = A_g \cdot (\Phi_{1,i,j,0} + S_{0,i,j,0}) + (1 - A_g) \cdot \pi \cdot B_g \cdot \Delta x \cdot \Delta y \qquad \text{(reflection and emission at the ground)}$$

In here, $E_0$ denotes the extraterrestrial solar irradiance (in units of $\mathrm{W\,m^{-2}}$), $\theta_{\mathrm{inc}}$ the solar zenith angle, $A_g$ the ground albedo, $B_g$ the emitted black body radiance of the ground (in units of $\mathrm{W\,m^{-2}\,sr^{-1}}$) and $\Delta x$ and $\Delta y$ the horizontal grid box lengths (in units of m). The boundary conditions employed at the sides of the domain depend on the model configuration and can either be cyclic or provided by neighboring subdomains. The resulting system of linear equations can then be solved by various

numerical methods. In the original TenStream solver, they are provided by the parallel linear algebra library PETSc (Balay et al., 2023).

## 2.2 Introducing time-stepping and incomplete solves: the dynamic TenStream solver

However, solving this large system of linear equations is a difficult task, especially when it needs to be parallelized for large NWP simulations. The main reason behind this difficulty is the fundamentally different approach on how radiation and dynamics are treated in numerical models. On the one hand, solving the equations of motion that govern advection in the dynamical core of an NWP model represents an initial value problem that has no known analytical solution. Hence, these equations are discretized in space and time and solved by a time-stepping scheme, where model variables are gradually propagated forward in time by applying the discretized equations onto values obtained at previous time steps (Holton and Hakim, 2012). An individual grid box thereby only needs information about itself and its nearby surroundings, facilitating model parallelization. Radiative transfer on the other hand is treated as a boundary value problem, where information is not gradually propagated through the domain, but rather spread almost instantaneously at the speed of light, involving the entire model grid. Three-dimensional radiative transfer can thus easily break model parallelization, as a radiative flux at any position in the domain can theoretically depend on all other radiative fluxes throughout the domain. This can be seen by looking at the coupled structure of the equations in the original TenStream solver in Eq. (1).

### 2.2.1 The Gauß-Seidel method

We tackle this problem by treating radiation similar to initial value problems. To this end, we build upon the TenStream linear equation system revisited in Sect. 2.1 and examine its solution with the Gauß-Seidel method, as it is described in e.g., Wendland (2017). According to this iterative method, a system of linear equations must be transformed in a way that there is one equation solved for every unknown variable. This form is given by the equations in Eq. (1) with the unknown variables being all the radiative fluxes in the entire domain. Providing a first guess for all of these variables, one then iterates through all these equations and sequentially updates all the radiative fluxes on the left-hand side of the equations by applying either the first guess, or, if already available, the updated values to the corresponding variables on the right-hand side of the equations. Applied to the TenStream equations, this means that one gradually iterates through all the grid boxes of the entire domain. For every grid box, one then calculates updated values for the outgoing fluxes $\Phi_{\text{out},m,i,j,k}^{(l+1)}$ on the left side of Eq. (1) by applying either already updated ingoing fluxes $\Phi_{\text{in},m,i,j,k}^{(l+1)}$, or, if not yet available, their values $\Phi_{\text{in},m,i,j,k}^{(l)}$ from the previous Gauß-Seidel iteration step to the variables on the right side of the equations:

$$\Phi_{\text{out},m,i,j,k}^{(l+1)} = \sum_{n=0}^{9+3} t_{mn,i,j,k} \cdot \begin{cases} \Phi_{\text{in},n,i,j,k}^{(l+1)} & \text{if } \Phi_{\text{in},n,i,j,k}^{(l+1)} \text{ has already been calculated} \\ \Phi_{\text{in},n,i,j,k}^{(l)} & \text{otherwise} \end{cases} + B_{m,i,j,k} \quad (2)$$

In here, the indices $m$ and $n$ denote an individual entry of the outgoing flux vector $\boldsymbol{\Phi}_{\text{out},i,j,k}$, the ingoing flux vector $\boldsymbol{\Phi}_{\text{in},i,j,k}$ or the thermal source vector $\boldsymbol{B}_{i,j,k}$, whereas $l$ quantifies the Gauß-Seidel iteration step and $t_{mn,i,j,k}$ refers to the corresponding entry in matrix $\boldsymbol{T}_{i,j,k}$ in Eq. (1). Completing this procedure for all the grid boxes and boundary conditions accomplishes one

Gauß-Seidel iteration. One can then repeat this procedure with the updated radiative fluxes serving as the new first guess, until the values eventually converge to the solution of the linear equation system. The thermal source terms are not part of the first guess and have to be calculated from scratch following the pattern outlined in Sect. 2.1 before starting with the Gauß-Seidel algorithm.

### 2.2.2 Dynamic treatment of radiation

We use the Gauß-Seidel method to significantly speed up 3D radiative transfer calculations by utilizing two main concepts: a time-stepping scheme and incomplete solves.

    To introduce the time-stepping scheme, we make use of the fact that the Gauß-Seidel algorithm requires us to choose an initial guess from where to start. So instead of solving the whole TenStream linear equation system from scratch every time, we use the result obtained at the previous call of the radiation scheme as a starting point of the algorithm. Assuming that the

field of optical properties determining the radiative fluxes has not changed fundamentally between two calls of the radiation scheme, this first guess should already be a good estimator of the final result. However, for the very first call of the radiation scheme, we cannot use a previously calculated result. In order to choose a reasonable starting point of the algorithm for this first call as well, though, we could use a full TenStream solve. However, such a solve would be computationally expensive and rely on numerical methods provided by the PETSc library, that we want to get rid of with our new solver to allow for easier

integration into operational models. So instead of performing a full TenStream calculation, we decided to solve the TenStream linear equation system for a clear sky situation as a starting point. This is the spin-up mentioned in Fig. 2. Since there is no horizontal variability in the cloud field in a clear-sky situation and our model does not feature any horizontal variability in the background atmosphere, we can perform this calculation for a single vertical column at a dramatically increased speed compared to a calculation involving the entire model grid. We cannot use a 1D solver for that, however, because we also need

to pass initial values to the sideward facing fluxes in the TenStream equation system. Assigned to the radiative fluxes of all vertical columns in the entire domain, these values then provide a first guess for all the TenStream variables that can be assumed to be much closer to the final result than starting with values of zero – even if the background atmosphere was not horizontally homogeneous and we would have to take the average of that background first.

    Based on the idea that the radiative field does not fundamentally change between two calls of the radiation scheme, we

furthermore just perform a limited number $N$ of iterations of the Gauß-Seidel algorithm every time the radiation scheme is called, essentially not letting it fully converge. Unless the radiative fluxes have changed dramatically compared to the last calculation, adjusting the variables towards the new solution should already provide a good approximation of the full solution, especially since it incorporates inter-column 3D effects, unlike the 1D independent column solutions used nowadays.

    The combination of these two efforts is visualized in Fig. 2. Instead of calculating a full 1D solution from scratch every

time radiation is called, our dynamic approach uses the previously obtained result as the starting point of a new incomplete 3D solve. This treatment of radiation puts it much closer to the way initial value problems like advection in the dynamical core of an NWP model are handled. Both use previously calculated results to update their variables. And looking at an individual grid box, updating the outgoing fluxes by applying Eq. (2) only requires access to the fluxes entering that exact same grid box and

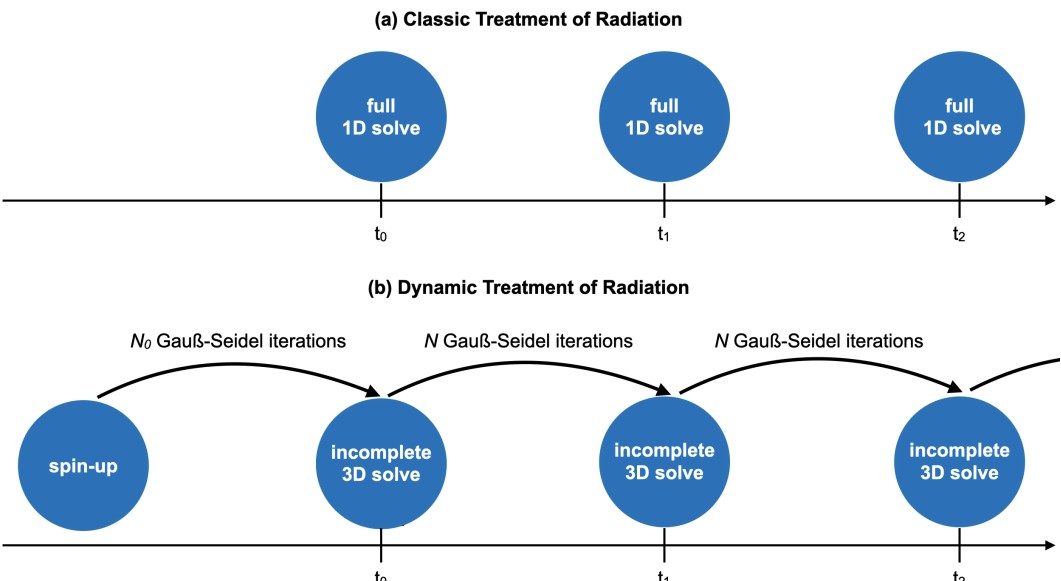

**Figure 2.** Schematic illustration of the dynamic treatment of radiation compared to the classic treatment. Instead of performing full 1D solves from scratch every time the radiation scheme is called, we use the result obtained at the last call as a starting point of an incomplete 3D solve, adjusting the previously calculated radiative fluxes towards the new full 3D solution.

thus only to neighboring values, just like in the discretized equations describing advection in the dynamical core of an NWP
model.

But even though the calculation of updated outgoing fluxes only requires access to fluxes entering the exact same grid box, this update process can indeed involve more distant grid boxes, since their calculation uses ingoing fluxes calculated in the very same Gauß-Seidel iteration wherever possible. And since these ingoing fluxes are outgoing fluxes of a neighboring grid box that may have also been calculated using already updated radiative fluxes, information can spread across the domain wherever possible, involving e.g. entire subdomains in NWP models. This is visualized in Fig. 3, which shows the first few steps of a Gauß-Seidel iteration in two dimensions only. Looking for example at the third step, outgoing fluxes of the upper-right grid box (highlighted in grey) are updated using the corresponding ingoing fluxes. Thereby, the ingoing flux of direct radiation entering the grid box on the left-hand side for example already contains radiative transfer through the two grid boxes on its left-hand side. This shows that the iteration direction through the grid boxes within a Gauß-Seidel iteration is crucial, as information can spread much faster in the direction one iterates through the grid boxes. Since the Gauß-Seidel algorithm allows us to freely choose the order in which to proceed through the system of linear equations, we can use this order to our advantage. First, we use the fact that whereas diffuse radiation spreads into all directions simultaneously, direct radiation clearly propagates in the direction of the Sun. Hence, for the solar spectral range, we first iterate through the grid boxes in the direction given by solar incidence in the horizontal and then from top to bottom in the vertical, as it is indicated by the dashed brown arrow in Fig. 3. In contrast to this two-dimensional example, both horizontal dimensions are affected by the position of the Sun in the fully

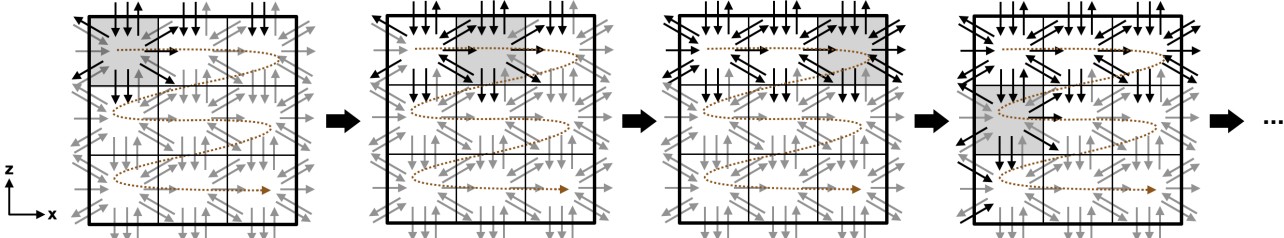

**Figure 3.** Two-dimensional schematic illustration of the first four steps of a Gauß-Seidel iteration, showing both diffuse and direct TenStream fluxes in case of Sun shining from the west or left-hand side. As one sequentially iterates through the grid boxes, ingoing fluxes are used to update the outgoing fluxes of the corresponding grid box (highlighted in grey). Grey arrows in contrast to black arrows indicate fluxes that have not yet been updated in this Gauß-Seidel iteration. Ingoing fluxes at the domain borders are dependent on the type of boundary conditions used. For this schematic, we applied periodic boundary conditions in the horizontal direction, while fluxes entering at the top of the domain are updated right from the beginning.

three-dimensional case, of course. If the Sun is shining from the south-west, for example, we would hence first iterate from south to north and west to east in the horizontal, before iterating from top to bottom. In the thermal spectral range, however, emitted radiation is larger in the lower part of the domain due to the vertical temperature gradient in the atmosphere. Hence, we iterate from bottom to top in the vertical there. Independent of the spectral range, we however still need to consider that diffuse
radiation spreads in all directions simultaneously, which we do not account for by using a fixed iteration direction. Thus, every time we finish iterating through all the grid boxes, which completes a Gauß-Seidel iteration step, we reverse the direction of iteration in all three dimensions to not favor propagation of information in one direction.

Combined, these efforts should allow us to very efficiently calculate radiative transfer in three dimensions: First, the time-stepping scheme allows us to already start with a reliable first guess instead of calculating everything from scratch. Next, we
speed up the rate of convergence by choosing a proper order in which to proceed through the linear equation system. And since the updated solution should not be radically different from the previous one, we furthermore just perform a limited number of Gauß-Seidel iterations, essentially exiting the algorithm before fully converged, arguing that an incomplete 3D solution should still be better than a 1D solution neglecting all 3D effects – as we will also see later on in Sect. 4. And finally, updating the outgoing radiative fluxes of any grid box within a Gauß-Seidel iteration just requires access to fluxes entering the exact same
grid box, which facilitates model parallelization. Implemented into the method, incomplete dynamic TenStream solves with $N$ Gauß-Seidel iterations each would then be calculated in parallel for the different subdomains, with communication between these subdomains ideally taking place just once afterwards at the end of the radiation scheme call. In this case, the spread of information would be limited to the scopes of the individual subdomains for every call of the radiation scheme.

### 2.2.3 Calculation of heating rates

In the end though, we are not just interested in calculating radiative fluxes, but especially in three-dimensional heating rates. They quantify local changes in temperature with time due to sources and sinks of radiative energy in the atmosphere and can

be calculated using the net irradiance divergence (Mayer, 2018):

$$\frac{\partial T}{\partial t} = \frac{1}{\rho \cdot c_p} \nabla \cdot \boldsymbol{E} = \frac{1}{\rho \cdot c_p} \left( \frac{\partial E_x}{\partial x} + \frac{\partial E_y}{\partial y} + \frac{\partial E_z}{\partial z} \right) \tag{3}$$

Here, $T$ denotes temperature, $t$ time, $\rho$ air density, $c_p$ specific heat capacity of air at constant pressure and $\boldsymbol{E}$ net irradiance (in units of W m$^{-2}$) with components $E_x$, $E_y$ and $E_z$ when expressed in Cartesian coordinates. Applied to the TenStream fluxes (in units of W) outlined in Sect. 2.1, we have to find expressions for the net flux in all three dimensions and then divide these by the area of the grid box surface they refer to. For the calculation of net fluxes we have to recall that TenStream features two streams to describe the transport of diffuse radiation on each of its sides. Since these two streams describe the flux entering and exiting a grid box in the upper and lower hemispheres, respectively, the total flux entering or exiting a grid box on one of its sides is given by the sum of these two streams. The net flux in any of the three dimensions is thus given by adding up all diffuse and direct fluxes entering the grid box in that dimension, minus those exiting it in the very same dimension. The heating rate of a grid box can thus be expressed as

$$\left( \frac{\Delta T}{\Delta t} \right)_{i,j,k} = \frac{1}{\rho \cdot c_p} \left[ \frac{1}{\Delta x} \cdot \frac{1}{\Delta y \cdot \Delta z} \left( \underbrace{\sum_{m=2}^{5} \left( \Phi_{\text{in},m,i,j,k} - \Phi_{\text{out},m,i,j,k} \right)}_{\substack{\text{net diffuse radiative flux} \\ \text{in x direction}}} + \underbrace{\left( \Phi_{\text{in},11,i,j,k} - \Phi_{\text{out},11,i,j,k} \right)}_{\substack{\text{net direct radiative flux} \\ \text{in x direction}}} \right) \right.$$

$$+ \frac{1}{\Delta y} \cdot \frac{1}{\Delta x \cdot \Delta z} \left( \underbrace{\sum_{m=6}^{9} \left( \Phi_{\text{in},m,i,j,k} - \Phi_{\text{out},m,i,j,k} \right)}_{\substack{\text{net diffuse radiative flux} \\ \text{in y direction}}} + \underbrace{\left( \Phi_{\text{in},12,i,j,k} - \Phi_{\text{out},12,i,j,k} \right)}_{\substack{\text{net direct radiative flux} \\ \text{in y direction}}} \right)$$

$$\left. + \frac{1}{\Delta z} \cdot \frac{1}{\Delta x \cdot \Delta y} \left( \underbrace{\sum_{m=0}^{1} \left( \Phi_{\text{in},m,i,j,k} - \Phi_{\text{out},m,i,j,k} \right)}_{\substack{\text{net diffuse radiative flux} \\ \text{in z direction}}} + \underbrace{\left( \Phi_{\text{in},10,i,j,k} - \Phi_{\text{out},10,i,j,k} \right)}_{\substack{\text{net direct radiative flux} \\ \text{in z direction}}} \right) \right]$$

$$= \frac{1}{\rho \cdot c_p} \cdot \frac{1}{\Delta x \cdot \Delta y \cdot \Delta z} \cdot \sum_{m=0}^{12} \left( \Phi_{\text{in},m,i,j,k} - \Phi_{\text{out},m,i,j,k} \right) \tag{4}$$

with $\Delta x$, $\Delta y$ and $\Delta z$ quantifying the size of the grid box. However, this formula raises some problems in combination with the incomplete solves introduced in Sect. 2.2.2. To explain this, we can once more look at Fig. 3. While for example fluxes exiting the upper-left grid box are updated in the very first step, the diffuse flux entering that exact same grid box from the bottom is updated much later in the fourth step. Hence, when the whole Gauß-Seidel iteration is completed, the fluxes exiting a certain grid box do not necessarily match the ones entering it anymore, i.e., the fluxes are not consistent anymore. This can lead to heating rates that are unphysically large or negative in the solar spectral range. To avoid this problem, we have to rephrase the outgoing fluxes in Eq. (4) in terms of ingoing fluxes, as it is given by the equations in Eq. (1):

$$\left( \frac{\Delta T}{\Delta t} \right)_{i,j,k} = \frac{1}{\rho \cdot c_p} \cdot \frac{1}{\Delta x \cdot \Delta y \cdot \Delta z} \cdot \sum_{m=0}^{12} \left( \Phi_{\text{in},m,i,j,k} - \sum_{n=0}^{12} t_{mn,i,j,k} \cdot \Phi_{\text{in},n,i,j,k} - B_{m,i,j,k} \right) \tag{5}$$

Since this expression incorporates the radiative transfer throughout the corresponding grid cell, it ensures that all fluxes involved in the calculation of the heating rate are consistent with each other and thus provides physically correct three-dimensional heating rates.

## 3 Evaluation method

The dynamic TenStream solver outlined in Sect. 2 was implemented into the libRadtran library for radiative transfer (Emde
et al., 2016; Mayer and Kylling, 2005), which allows for testing the performance of the new solver with respect to other solvers using an otherwise identical framework. Using this environment, our goal is to demonstrate that the new dynamic TenStream solver produces more accurate results than 1D independent column solvers employed nowadays while still being noticeably faster than typical 3D solvers. Therefore, this section will first introduce our test setup as well as the solvers we compare dynamic TenStream with. Then, we will explain how we determine its performance both in terms of speed and accuracy. Since
3D solvers are computationally much more demanding than 1D solvers, a special emphasis in this analysis will also be put on the calling frequency of the radiative transfer calculations in order to elaborate whether the dynamic TenStream solver is still performing better when operated with a similar computational demand as current 1D solvers.

### 3.1 Cloud and model setup

Our test setup is centered around a shallow cumulus cloud time series prepared by Jakub and Gregor (2022), which was com-
275 puted using the University of California, Los Angeles (UCLA) large eddy simulation (LES) model (Stevens et al., 2005). Dynamics in this LES simulation were not driven by radiation, but by a constant net surface flux as described in the corresponding namelist input files. Originally, the data set features both a high temporal resolution of 10 s and $256 \times 256$ grid boxes with a high spatial resolution of 25 m in the horizontal. It is 6 h long and characterized by a continuously increasing cloud fraction, starting with a clear sky situation and ending up with a completely overcast sky. In addition, a southerly wind at a
280 speed between 3 m s$^{-1}$ and 4.7 m s$^{-1}$ transports the clouds through the domain (Gregor et al., 2023).

We have chosen this data set for two reasons: First, the high temporal resolution allows us to investigate the effect of incomplete solves in the dynamic TenStream solver with regard to the calling frequency of the solver. As we outlined in Sect. 2.2.2, we expect these incomplete solves to perform best if the cloud field mainly determining the radiative field does not change much in between two radiation time steps. Due to the high temporal resolution, we can investigate how well the
285 incomplete solves perform if we call the solver less often by comparing runs with low calling frequencies to runs with the highest possible calling frequency of 10 s. On the other hand, we need the high spatial resolution of the data since dynamic TenStream does not yet take sub-grid scale cloud variability into account. However, we may not need a horizontal resolution of 25 m for that. Thus, to test the dynamic TenStream solver on a resolution that is closer to that of operational weather models without having to account for sub-grid scale cloud variability, we decided to reduce the horizontal resolution of the cloud fields
to 100 m. To avoid problems with artificially low liquid water content (LWC) at cloud edges when averaging the cloud field to that resolution, we constructed these lower resolved cloud fields by simply using the data of just every fourth grid box in both

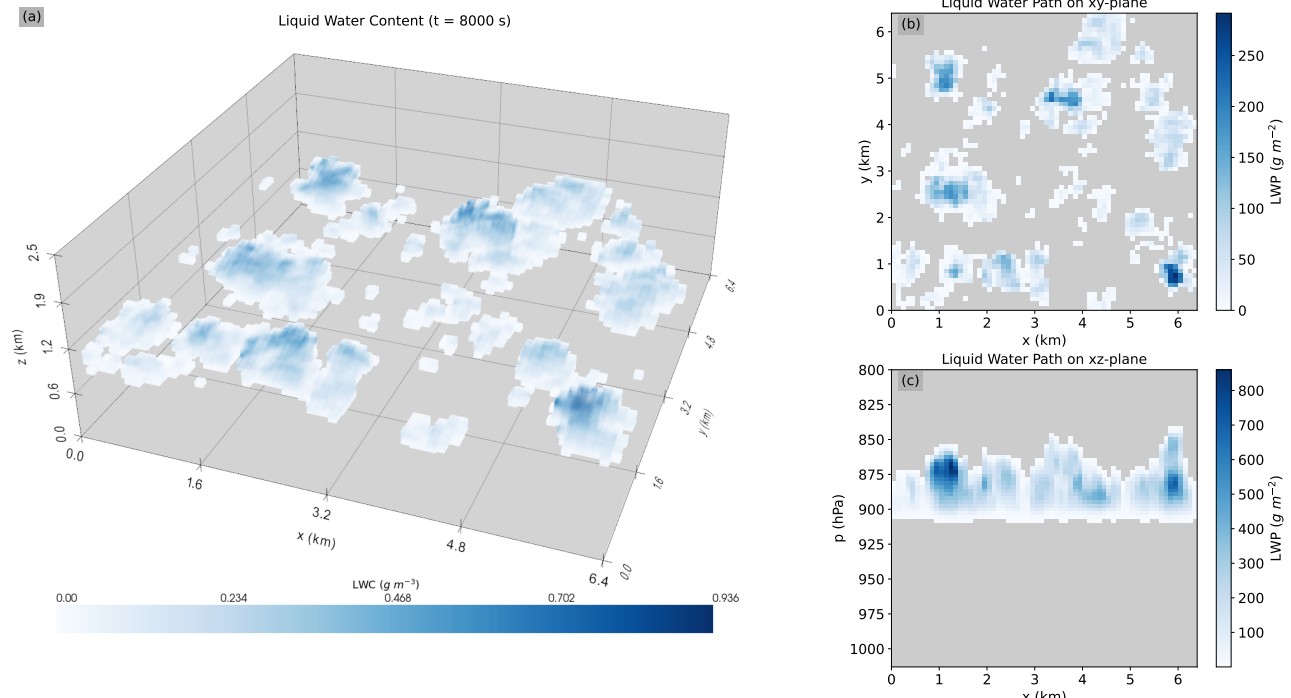

**Figure 4.** First time step of the shallow cumulus cloud field used in the evaluation. Panel (a) shows a 3D visualization of the liquid water content in the cloud field, whereas panels (b) and (c) display the vertically and horizontally integrated liquid water content for the same cloud field, respectively.

horizontal dimensions. The resulting time series still features a temporal resolution of 10 s, but the cloud data grid is reduced to $64 \times 64$ grid boxes with a resolution of 100 m in the horizontal. In the vertical, the modified cloud data set consists of 220 layers with a constant height of 25 m, thus reaching up to a height of 5.5 km. Using this modified grid, the shape of the grid boxes is also closer to the one in NWP models, with their horizontal extent being larger than their vertical extent.

For our test setup, we focus on the 100 time steps between 8000 s and 9000 s into the simulation, where the shallow cumulus cloud field has already formed, but not yet reached a very high cloud fraction, as both a clear as well as a completely overcast sky are not beneficial for 3D cloud-radiative effects. Figure 4 shows the modified cloud field for the very first time step in this time frame. Especially looking at the vertically integrated liquid water content in panel (b), one can see that our reduced horizontal resolution of 100 m allows us to still resolve the structure of the clouds.

Apart from the cloud field, the 1976 US standard atmosphere (Anderson et al., 1986) interpolated onto the vertical layers given by the cloud data grid serves as background atmosphere. Above the cloud data grid, the native US standard atmosphere levels as they are provided by libRadtran are used, so that the full grid features 264 layers in the vertical up to a height of 120 km. Both in the solar and the thermal spectral range, the simulations are carried out using the molecular absorption parameterization by Fu and Liou (1992, 1993). In the solar spectral range, the Sun is placed at a constant zenith angle of $50°$

and in the east. The zenith angle was chosen to be quite low so that 3D effects such as cloud side illumination and shadow displacement are more pronounced, representing a typical morning scene. Furthermore, the surface albedo in the solar spectral range is set to 0.125, resembling the global mean value of Trenberth et al. (2009), whereas the ground emissivity was set to 0.95 in the thermal spectral range.

## 3.2 Overview of the radiative transfer solvers

We apply four different radiative transfer solvers onto the aforementioned shallow cumulus cloud time series: the newly developed dynamic TenStream solver, the original TenStream solver, a classic one-dimensional $\delta$-Eddington approximation and a fully three-dimensional Monte Carlo solver.

Let us discuss the setup of the dynamic TenStream solver first. As we outlined in Sect. 2.2.2, it has to be provided with a first guess the very first time it is called due to the unavailability of a previously calculated result at this point in time. To evaluate the performance of the new solver, it is a good idea to use the best possible solution for this first guess. This way, one can examine whether from there on the results obtained using dynamic TenStream featuring incomplete solves diverge from those retrieved by the original TenStream solver using full solves. Hence, we initially perform 2,000 iterations for the clear sky spin-up described in Sect. 2.2.2 followed by $N_0 = 500$ Gauß-Seidel iterations also involving the cloud field to ensure that the radiative field is fully converged in the beginning of the time series. These two steps are visualized by the spin-up and arrow with the first $N_0$ Gauß-Seidel iterations in Fig. 2. From there on, we just use a minimum of two Gauß-Seidel iterations every time the solver is called. Two instead of just one iteration ensures that the iteration direction mentioned in Sect. 2.2.2 is at least altered once per call. This way, we guarantee that information is not preferably transported in one specific direction. To investigate the effect of using more than just two Gauß-Seidel iterations per call, we furthermore also performed nine additional runs with integer multiples of two Gauß-Seidel iterations, i.e., with four, six, ..., twenty Gauß-Seidel iterations per call.

Since the dynamic TenStream solver is based on the original TenStream solver, reproducing its results despite applying incomplete solves is the best outcome that we can expect. Thus, the original TenStream solver (Jakub and Mayer, 2015) serves as a best-case benchmark for our new solver. On the other hand, our goal is to significantly outperform currently employed one-dimensional independent column approximations. Consequently, the $\delta$-Eddington solver incorporated in the libRadtran radiative transfer library serves as a worst-case benchmark for our new solver that we should definitely surpass.

Finally, we also apply the 3D Monte Carlo solver MYSTIC (Mayer, 2009) onto the shallow cumulus cloud time series. When operated with a large enough number of photons, it allows to determine the most accurate three-dimensional heating rates possible. Hence, these results can be used as a benchmark for all the other solvers. For our MYSTIC simulations, we used a total of 400,000,000 photons for every time step – that is about 100,000 per vertical column, resulting in domain-average mean absolute errors in both heating rates and irradiances that are not larger than 1 % of their respective domain averages.

### 3.3 Speed and accuracy evaluation

As we mentioned earlier, our goal is to evaluate the performance of dynamic TenStream both in terms of speed and accuracy. However, especially determining the speed of a solver with respect to others is not a straightforward task, as it is highly depen-

dent on the environment the code is executed in. Since the dynamic TenStream solver is still in an early stage of development

and this work is primarily focused on demonstrating the feasibility of the main concepts of the solver, we wanted to keep the speed analysis as simple as possible. We decided to perform three radiative transfer computations for each of the previously mentioned solvers on the same workstation for the first cloud field in our time series. The average of these three run times for every solver should at least provide a rough estimate of the relative speed of the different solvers to each other. All calculations were performed on a single core. We compare the computational time of incomplete dynamic TenStream solves with

two Gauß-Seidel iterations per call, which is the same setup as the one we will use for the investigation of the performance of the new solver later on, to run times of full solves by the 1D $\delta$-Eddington approximation, the original TenStream solver and MYSTIC. This comparison is not entirely fair for the original TenStream solver though, as this solver can be run in a time-stepping scheme as well and thus rely on previously calculated results, noticeably increasing its speed compared to calculations from scratch that we are using. However, this time-stepping option for the original TenStream solver is not yet available within

libRadtran.

To assess the accuracy of the dynamic TenStream solver, we study the entire time series. We focus our analysis on how well the solvers perform in determining three-dimensional heating rates and net irradiances at the upper and lower domain boundaries. As mentioned in Sect. 3.2, values derived by MYSTIC serve as benchmark values. We evaluate the performance of the other three solvers compared to MYSTIC using two different error measures: a mean absolute error (MAE) and a mean

bias error (MBE). The mean absolute error describes the amount by which the heating rate or net irradiance of an individual grid box on average deviates from the benchmark solution:

$$
\text{MAE} = \left\langle \left| \xi - \xi_{\text{ref}} \right| \right\rangle \quad \text{with} \quad \xi \in \left\{ \left( \frac{\Delta T}{\Delta t} \right)_{i,j,k} ; \Delta E_{\text{net},i,j,\text{sfc}} ; \Delta E_{\text{net},i,j,\text{TOA}} \right\} \tag{6}
$$

$$
\text{and} \quad \Delta E_{\text{net},i,j,\text{sfc}} = (S_{0,i,j,0} + \Phi_{1,i,j,0} - \Phi_{0,i,j,0}) \cdot \Delta x \cdot \Delta y
$$

$$
\Delta E_{\text{net},i,j,\text{TOA}} = (S_{0,i,j,N_z+1} + \Phi_{1,i,j,N_z+1} - \Phi_{0,i,j,N_z+1}) \cdot \Delta x \cdot \Delta y
$$

In here, $\langle ... \rangle$ denotes a spatial average, whereas the subscript "ref" refers to a reference value, i.e., the MYSTIC values in our case. Since the mean absolute error is sensitive to how the values of an individual grid box deviate from the benchmark solution, it is a measure of whether a solver gets the overall heating rate or net irradiance pattern right. It is sensitive to double-penalty errors, i.e., gets large when local minimums and maximums in this pattern are displaced between the benchmark solution and the investigated solver. We have chosen an absolute error measure rather than a relative one here, because individual heating

rates or net irradiances can be close to zero and thus blow up a relative error measure. The mean bias error on the other hand is an error measure targeted towards the domain mean heating rate or net irradiance:

$$
\text{MBE} = \langle \xi \rangle - \langle \xi_{\text{ref}} \rangle \quad \text{with} \quad \xi \in \left\{ \left( \frac{\Delta T}{\Delta t} \right)_{i,j,k} ; \Delta E_{\text{net},i,j,\text{sfc}} ; \Delta E_{\text{net},i,j,\text{TOA}} \right\} \tag{7}
$$

In contrast to the MAE, the MBE compares domain-average values to each other and is thus a measure for whether we get the domain-average heating rate or net irradiance right. It is not sensitive to the spatial pattern of these quantities, but rather tells

us whether there is on average too much or too little absorption in the domain compared to the benchmark solution. Domain

averages of heating rates and net irradiances are usually not close to zero, so that we can also take a look at the relative error measure here:

$$\text{RMBE} = \frac{\langle \xi \rangle - \langle \xi_{\text{ref}} \rangle}{\langle \xi_{\text{ref}} \rangle} \quad \text{with} \quad \xi \in \left\{ \left( \frac{\Delta T}{\Delta t} \right)_{i,j,k} ; \Delta E_{\text{net},i,j,\text{sfc}} ; \Delta E_{\text{net},i,j,\text{TOA}} \right\} \tag{8}$$

Applied to the shallow cumulus cloud time series at its full temporal resolution of 10 s, these two error measures allow us to

determine the accuracy of the dynamic TenStream, original TenStream and $\delta$-Eddington calculations compared to the MYSTIC benchmark run at any point in time. We can also ensure that the benchmark solution itself has a significantly smaller error than the other solvers when compared to this benchmark. Therefore, we use the standard deviation $\sigma_{\text{ref}}$ that can be determined for every single MYSTIC value. However, this standard deviation describes the mean squared deviation of a MYSTIC value from its mean, whereas the MAE that we are using is looking at the mean absolute deviation. For normally distributed random

variables, however, the mean absolute deviation is simply given by

$$\text{MAD} = \sqrt{\frac{2}{\pi}} \cdot \sigma \tag{9}$$

with $\sigma$ being the standard deviation (Geary, 1935). We can assume that the benchmark run is of sufficient quality if this MAD of the MYSTIC values is much smaller than the corresponding mean absolute deviations between MYSTIC and the values of the other solvers. Hence, we can use the domain-average MAD of the benchmark solution to quantify the domain-average

MAE of the benchmark solution at any point in time:

$$\text{MAE}_{\text{ref}} = \left\langle \sqrt{\frac{2}{\pi}} \cdot \sigma_{\text{ref}} \right\rangle \tag{10}$$

We cannot provide a number for the MBE of the benchmark solution, though, as we only know how much the individual MYSTIC values are scattered around their mean, but not whether this mean has an inherent bias. Hence, we simply have to assume that our benchmark simulation is unbiased.

So far, this evaluation would only tell us the accuracy of the different solvers compared to the benchmark run when operated at the same, highest possible calling frequency of 10 s. However, radiation is usually called far less often than the dynamical core of the model. Also, 3D radiative transfer solvers are computationally much more demanding than 1D solvers, raising the question of how good dynamic TenStream performs when operated with a similar computational demand as the 1D $\delta$-Eddington approximation. To address these questions, we also investigate the effect of the radiation calling frequency on the

temporal evolution of the aforementioned error measures.

In order to explain our approach to this, we take a look at Fig. 5. It demonstrates how we determine the aforementioned error measures for a solver operated at a lower calling frequency of 30 s with respect to the MYSTIC benchmark run, that is computed at the highest possible calling frequency of 10 s. At $t = 8020$ s, that is 20 s into our time series, these error measures are given by comparing the not yet updated solution that has been originally calculated at $t = 8000$ s to the values

of the benchmark solution obtained at exactly $t = 8020$ s. This way, we can investigate how the error metrics of a not updated radiative field grow until it is eventually updated again. This investigation is particularly important for the dynamic TenStream

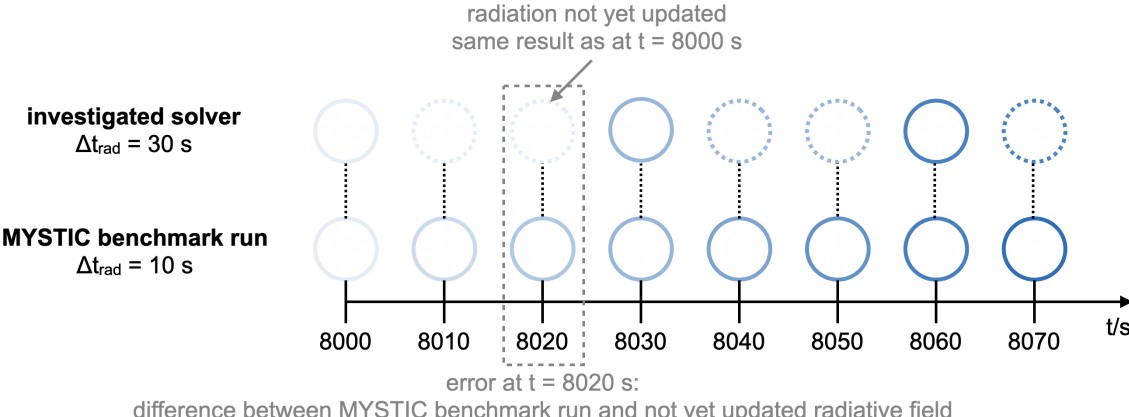

**Figure 5.** Schematic illustration of how we determine the error of a solver operated at a lower calling frequency of $\Delta t_{rad} = 30\,s$ compared to the benchmark solution computed at the highest possible calling frequency of $\Delta t_{rad} = 10\,s$ at any point in time. To this end, the circles in the figure indicate the results of the corresponding solvers at any given time, with the colors symbolizing the time step at which these results have been calculated and the dotted circles in contrast to the full circles indicating results that have not been updated at that point in time. The color of these dotted circles is thus equal to that of the corresponding last full circle.

solver, as it just performs incomplete solves every time it is called. As we expect these to work best if the overall properties determining the radiative field have not changed much in between two calls of the solver, this method allows us to investigate whether our new solver still converges towards the results of the original TenStream solver when operated at lower calling frequencies. For this paper, we decided to take calling frequencies of 10 s, 30 s and 60 s into account. These calling frequencies are still very high for operational weather forecasts, where the radiation time step is typically around one hour (Hogan and Bozzo, 2018), but we have to consider that our cloud field also features a significantly higher spatial resolution of 100 m in the horizontal compared to 2.2 km in the DWD ICON-D2 model (DWD, 2024) and 9 km in the ECMWF high-resolution deterministic forecasts (Hogan and Bozzo, 2018). At the LES resolution of 100 m that we use for our evaluation, the Weather Research and Forecasting (WRF) model for example recommends a radiation time step as high as 1 minute per kilometer of horizontal resolution (UCAR, 2024), resulting in a suggested radiation time step of 6 s for our test case. Our highest calling frequency of 10 s is at least close to that number, with the other two calling frequencies of 30 s and 60 s definitely representing scenarios where radiation is called less often than recommended.

## 4 Results and discussion

### 4.1 Solver speed

The relative speed of the different radiative transfer solvers introduced in Sect. 3.2 compared to the run time of the 1D $\delta$-Eddington approximation is shown in Table 1. As we described in Sect. 3.3, all solvers for this test were executed on a single

**Table 1.** Computing time of the different solvers relative to those of the 1D $\delta$-Eddington approximation, taken as an average over three runs performed on the same workstation for the very first time step of the LES cloud time series

| | solar spectral range | thermal spectral range |
| --- | --- | --- |
| **$\delta$-Eddington**<br>*1D two-stream solver* | 1.0 | 1.0 |
| **dynamic TenStream**<br>*incomplete 3D solver with two Gauß-Seidel iterations* | 3.6 | 2.6 |
| **original TenStream**<br>*full 3D solver* | 50.8 | 24.1 |
| **MYSTIC**<br>*full 3D benchmark solver using 400,000,000 photons* | 1068.9 | 1611.3 |

core of the same workstation and therefore in a very similar environment. This workstation featured an Intel Xeon W-2245 CPU and 64 GB RAM, with performance primarily limited by the network storage where all the data has been placed. We can see that in this experiment, the newly developed dynamic TenStream solver with two Gauß-Seidel iterations is 3.6 times slower than the 1D $\delta$-Eddington approximation in the solar spectral range and just 2.6 times slower in the thermal spectral range. Comparing these numbers to the findings in Jakub and Mayer (2016), they are in line with what we could have expected in terms of the speed of the new solver: According to this paper, retrieving the coefficients of the TenStream linear equation system from the look-up tables in both the solar and thermal spectral range takes about as long as performing one $\delta$-Eddington calculation. On top of that, we have to calculate the fluxes for every grid box of the domain, just as in a $\delta$-Eddington calculation. However, for the dynamic TenStream calculation, we have to determine fluxes for ten instead of two streams per grid box and calculate all these fluxes twice as we perform two Gauß-Seidel iterations whenever the solver is called. And even though the number of streams in particular will most likely not scale linearly with run time, we can thus certainly expect that the new solver is at least twice as slow as a $\delta$-Eddington approximation with factors of 3.6 and 2.6 being on par with that.

Although the dynamic TenStream solver is thus noticeably slower than a 1D solver, it is still significantly faster when compared to other 3D solvers executed under similar circumstances – namely the original TenStream and MYSTIC solvers in Table 1. The original TenStream solver for example is at least 24 times slower in this experiment than the 1D $\delta$-Eddington approximation, with MYSTIC being even slower. As we pointed out earlier, this comparison is not entirely fair for the original TenStream solver though, as it can also be run in a time-stepping scheme. Jakub and Mayer (2016) showed that in this case, the original TenStream solver can only be up to a factor of 5 slower than 1D $\delta$-Eddington solves, which however is still noticeably slower than the dynamic TenStream solver presented here.

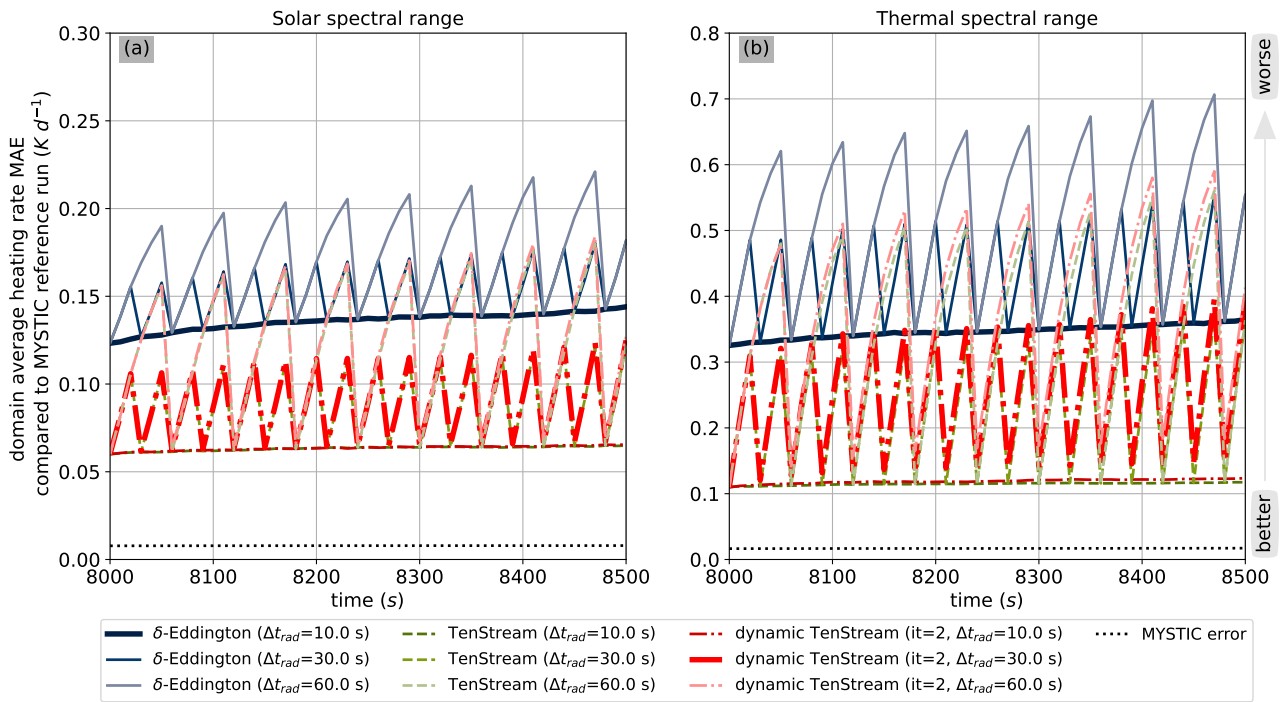

**Figure 6.** Temporal evolution of the mean absolute error in heating rates of the 1D $\delta$-Eddington approximation (blue lines), the original TenStream solver (green dashed lines) and the newly developed dynamic TenStream solver (red dash-dotted lines) with respect to the MYSTIC benchmark run at calling frequencies of 10 s, 30 s and 60 s (different shades of the corresponding color) for both the solar (panel a) and thermal (panel b) spectral range. Due to the statistical nature of Monte Carlo simulations, the MYSTIC benchmark run itself is subject to some uncertainty. The corresponding MAE calculated using Eq. (10) is visualized by the dotted black line. For reasons of visual clarity, we show only the first half of the time series here.

## 4.2 Performance in determining heating rates

Next, let us have a look at how the dynamic TenStream solver performs in calculating heating rates. Since we are primarily interested in its performance in the surroundings of the continuously evolving clouds, we only use the LES part of the domain
for this evaluation, i.e., the part between the surface and 5.5 km height. As mentioned in Sect. 3.3, the analysis will be centered around two different error measures: a mean absolute error and a mean bias error. Figure 6 shows the temporal evolution of the MAE for the different solvers at calling frequencies of 10 s, 30 s and 60 s. At this point, we should recall that the mean absolute error is a measure of how good a solver on average performs in determining the heating rate for a certain grid box.

When operated at the highest possible calling frequency of 10 s, we can see that the MAE is relatively constant in time for
all the solvers, as we compare radiative transfer calculations carried out at a certain point in time to benchmark calculations obtained at the exact same time step. The MAE in this case is solely determined by the error made by the solvers themselves when applied to relatively similarly structured shallow cumulus clouds, so this behavior is expected. Looking at the magnitude

of the MAE for the different solvers, we can see that for both spectral regions, the $\delta$-Eddington approximation (dark blue line) performs worst, whereas the 3D TenStream solver is a noticeable improvement. Pleasingly, the MAE of our dynamic TenStream solver at a calling frequency of 10 s (dark red dash-dotted line) is almost on par with the error obtained with the original TenStream solver. It is only in the thermal spectral range where its error is getting slightly larger with time. This shows that in this example, at a calling frequency of 10 s, just two Gauß-Seidel iterations per call are already sufficient to almost reproduce the results of the original TenStream solver.

At lower calling frequencies, the radiative field is not updated at every time step of the cloud time series anymore. Consequently, the MAE of the different solvers rises until the solver is called again. The resulting saw tooth structure can be observed in the MAE time series of all the solvers at calling frequencies of 30 s and 60 s. In case of the traditional solvers, a full solve is performed every time they are called. Thus, the MAE at lower calling frequencies always reduces to the value obtained at a calling frequency of 10 s when the corresponding solver is called. This is not necessarily true for the dynamic TenStream solver, however, as it is just performing an incomplete solve involving two Gauß-Seidel iterations every time it is called. If this incomplete solve would not be sufficient, it could lead to a divergent behavior of the MAE time series for this solver. Looking closely, we can also see that for both lower calling frequencies, the MAE of the dynamic TenStream solver does not always match the errors obtained at a calling frequency of 10 s when updated. However, even at a calling frequency of 60 s, we cannot observe a divergent behavior and the newly developed solver is almost able to reproduce the results of the original TenStream solver whenever called.

Moreover, we have seen that our new solver is about three times slower than a traditional 1D $\delta$-Eddington approximation. Looking at Fig. 6, we can now see that on time-average, dynamic TenStream even performs better than the $\delta$-Eddington approximation at a calling frequency of 10 s (bold blue line) when it is operated at a calling frequency of 30 s (bold red dash-dotted line) and thus with a similar computational demand as the 1D solver – both in the solar, as well as in the thermal spectral range.

Switching to the other error measure, Fig. 7 visualizes the temporal evolution of the mean bias error for the different solvers. In contrast to the MAE discussed before, this error metric describes whether we get the domain-average heating rate right. As we can clearly see, the MBE is again largest for the 1D $\delta$-Eddington approximation and once more significantly smaller for the original TenStream solver. When operated at the highest possible calling frequency of 10 s, the mean bias error of dynamic TenStream is also very similar to that of the original TenStream solver. However, at lower calling frequencies, we can clearly see that the bias increases with time, although never getting larger than the bias of the 1D results. It is also clearly visible that the bias is more negative than the original TenStream bias (green dashed lines) in the solar spectral range, whereas it is less positive in the thermal spectral range. Since the domain-average heating rate in the solar spectral range is positive, this implies that our new solver underestimates absorption in this spectral range, especially compared to the original TenStream solver it is based on. This underestimation is getting larger the less the dynamic TenStream solver is called. As the liquid water content in the domain gradually increases with time and more liquid water in the clouds leads to more absorption, this could imply that the dynamic TenStream solver does not fully take this increase into account. This does not explain the behavior of the new solver in the thermal spectral range, though, where domain-average heating rates are negative. Hence, the positive MBE

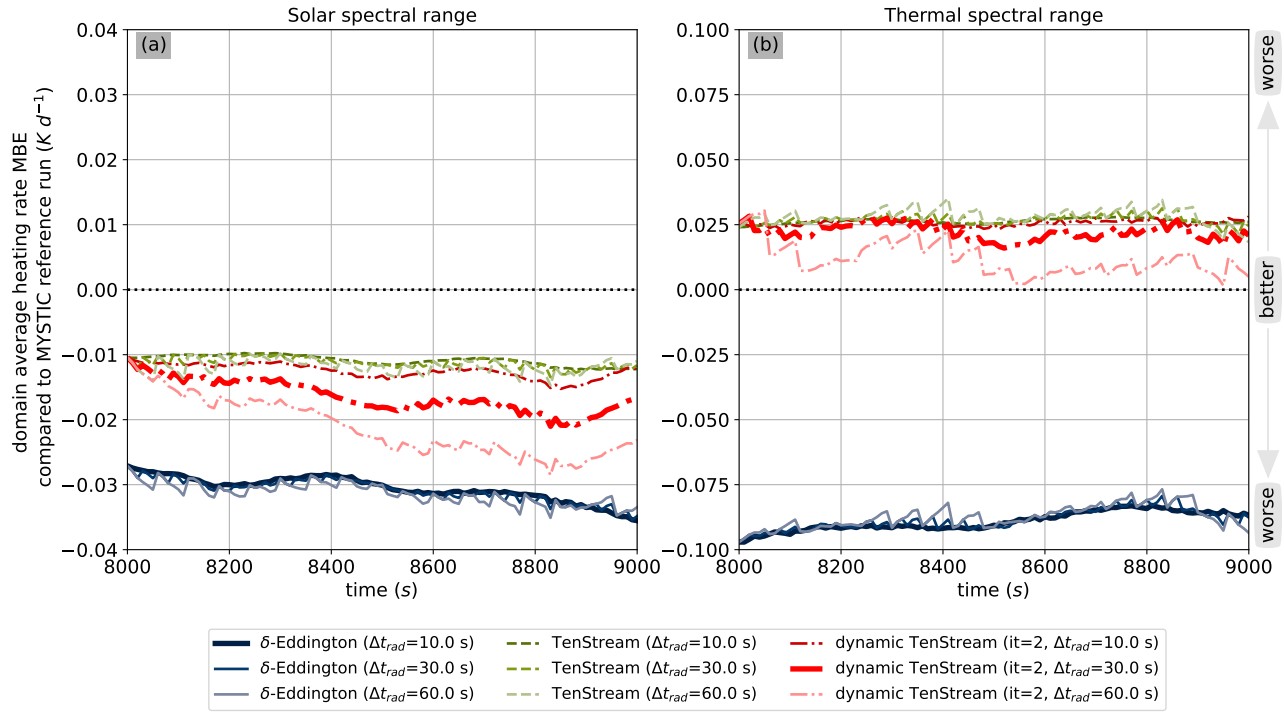

**Figure 7.** Temporal evolution of the mean bias error in heating rates for the different solvers with respect to the MYSTIC benchmark run at calling frequencies of 10 s, 30 s and 60 s for both the solar (panel a) and thermal (panel b) spectral range. A run with no bias is visualized by the dotted black line.

values observed for both the original as well as the dynamic TenStream solver imply that the heating rates are not as negative there as they should be. But in contrast to the solar spectral range, these heating rates get more negative the less the dynamic
TenStream solver is called, so that the dynamic TenStream solver overestimates the magnitude of these thermal heating rates when compared to the original TenStream solver it is based on. Using this and the results obtained from the MAE time series, we can draw a first few conclusions:

1. For an individual grid box, our new solver is able to determine heating rates much more accurately than current 1D solvers, even when operated with a similar computational demand.

2. When looking at domain averages, the dynamic TenStream solver begins to develop a bias compared to the original TenStream solver it is based on. This bias becomes larger the lower the calling frequency is, but remains smaller than the bias of the 1D $\delta$-Eddington calculations at any point in time.

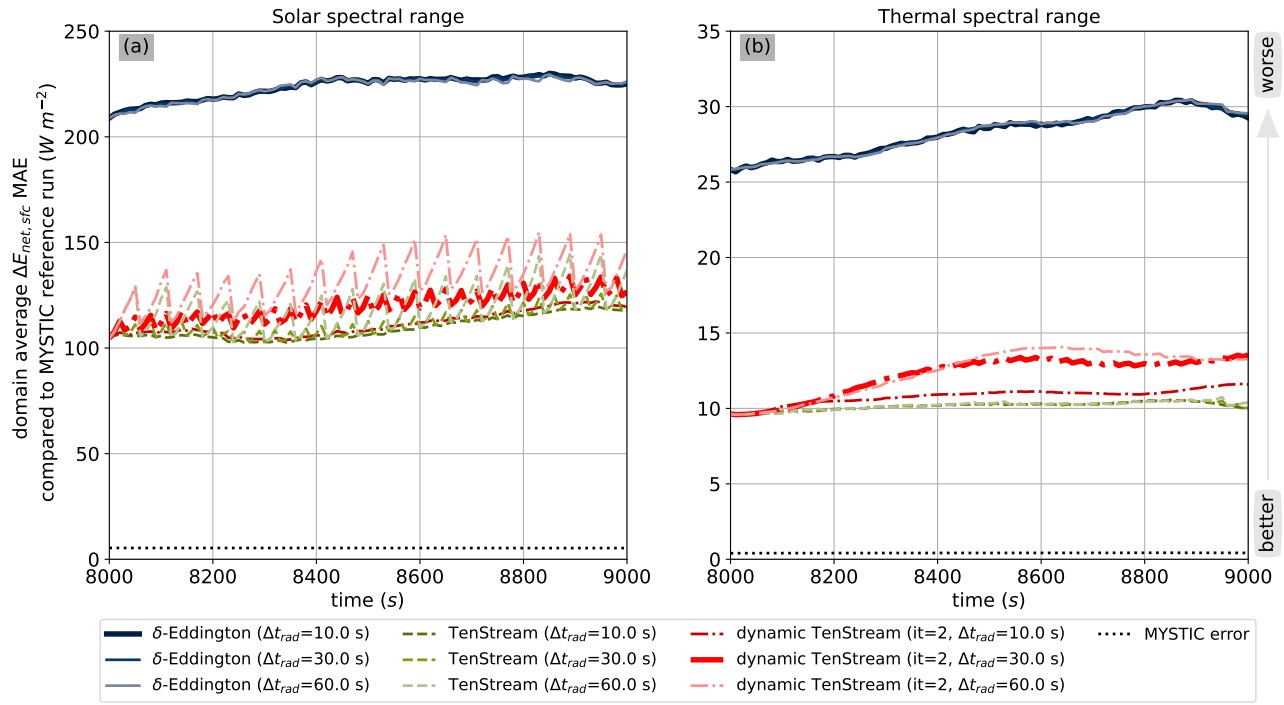

**Figure 8.** Temporal evolution of the mean absolute error in the net surface irradiance for the different solvers with respect to the MYSTIC benchmark run at calling frequencies of 10 s, 30 s and 60 s for both the solar (panel a) and thermal (panel b) spectral range. The MAE of the MYSTIC benchmark run itself is visualized by the dotted black line.

## 4.3 Performance in determining net irradiances at the upper and lower domain boundaries

Besides heating rates, we are also interested in how well dynamic TenStream performs in determining net irradiances at the
top and bottom of its domain. We will start by looking at the results for the net surface irradiances and thus absorption at the ground. Figure 8 shows the temporal evolution of the mean absolute error for this quantity in an otherwise similar fashion as Fig. 6 and Fig. 7. As for the heating rates, we can see that the 1D $\delta$-Eddington approximation (blue lines) performs worst, with the original TenStream solver (green dashed lines) once more being a noticeable improvement, remaining significantly below the errors of all 1D runs throughout the entire time series even at lower calling frequencies. Again, our newly developed
dynamic TenStream solver (red dash-dotted lines) is able to almost maintain the MAE of the full TenStream calculations at the highest possible calling frequency of 10 s, whereas its error slightly increases with time for the two lower calling frequencies. However, this slight divergence from the original TenStream MAE quickly stabilizes and also remains significantly below every single $\delta$-Eddington run even when just called every 60 s. What is interesting though is that the temporal evolution of the MAE in the thermal spectral range does not show a sawtooth structure at lower calling frequencies, in contrast to all other plots
involving the MAE so far. As we discussed earlier, this sawtooth structure is mainly caused by the fact that at lower calling

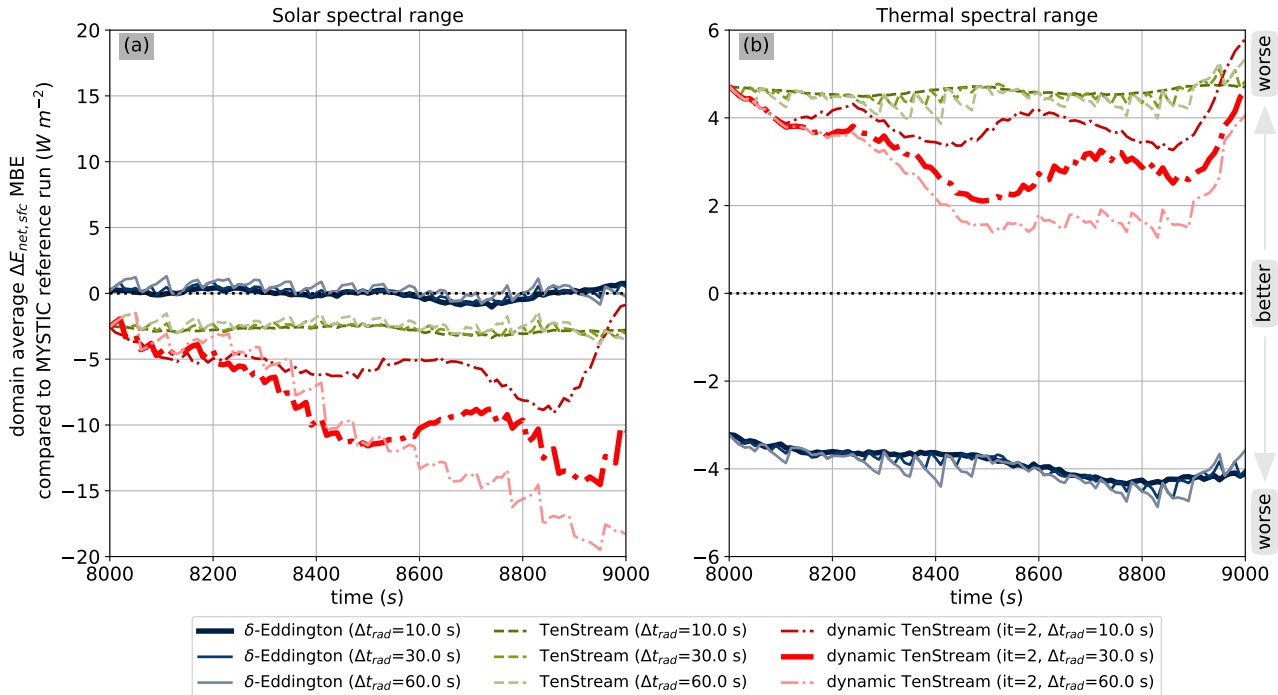

**Figure 9.** Temporal evolution of the mean bias error in the net surface irradiance for the different solvers with respect to the MYSTIC benchmark run at calling frequencies of 10 s, 30 s and 60 s for both the solar (panel a) and thermal (panel b) spectral range. A run with no bias is indicated by the dotted black line.

frequencies, we do not update the radiative field for some time steps while the clouds are still moving through the domain, resulting in gradually increasing double-penalty errors. The fact that this behavior is not observed in the thermal spectral range indicates that the net surface irradiance field does not feature such small-scale structures in the thermal.

To conclude the analysis for the surface, let us once more also have a look at the mean bias error. Again, in contrast to the
MAE, this error measure does not tell us how well dynamic TenStream performs in determining the net surface irradiance for a single grid box, but rather whether we get the domain-average surface absorption right. The corresponding plot is shown in Fig. 9 and reveals a current weakness of both the original TenStream solver as well as our new solver, as we can clearly see that the MBE is almost always larger for these two solvers than it is for the 1D $\delta$-Eddington approximation. And as we have already seen in the results for the heating rates, the lower the calling frequency, the more the MBE of the dynamic TenStream
solver diverges from the MBE of the original TenStream solver. Here, however, this behavior is more severe than it was for the heating rates, since already the benchmark for our new solver – the original TenStream solver – performs a bit worse than the 1D solver. Its MBE of about $-2.5\,\mathrm{W\,m^{-2}}$ in the solar and $5\,\mathrm{W\,m^{-2}}$ in the thermal spectral range translates to a RMBE of about $-0.5\%$ and $-6\%$, respectively (not shown here), compared to numbers of around $0\,\mathrm{W\,m^{-2}}$ $(0\%)$ in the solar and $-4\,\mathrm{W\,m^{-2}}$ $(5\%)$ in the thermal spectral range for the $\delta$-Eddington approximation. However, it should be noted that the almost

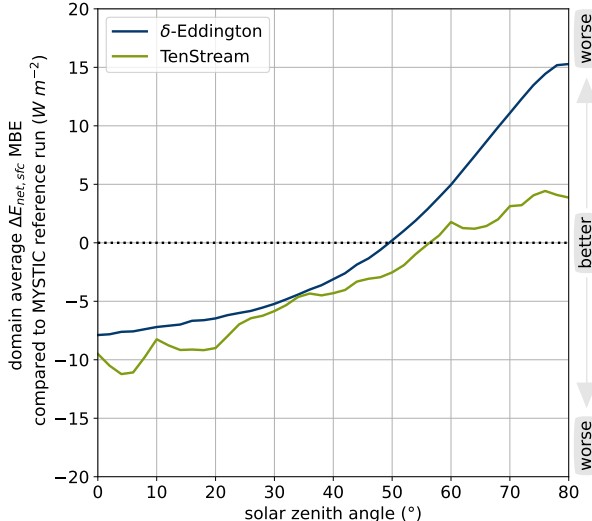

**Figure 10.** Mean bias error in the net surface irradiance as a function of the solar zenith angle for both the 1D $\delta$-Eddington approximation (blue line) and the original TenStream solver (green line), evaluated at the first time step of the shallow cumulus cloud time series.

non-existent MBE of the $\delta$-Eddington approximation in the solar spectral range is primarily caused by two counteracting 3D radiative effects that happen to cancel each other out at almost exactly the solar zenith angle of $50°$ that we are using.

To show that, Fig. 10 visualizes the MBE for both the $\delta$-Eddington approximation and the original TenStream solver as a function of the solar zenith angle for the first time step of our time series. By looking at the blue line, we can see that the $\delta$-Eddington approximation underestimates the mean net surface irradiance for solar zenith angles below $50°$, while it 525    overestimates it for angles above $50°$. This is most likely because at low solar zenith angles, 1D solvers typically overestimate cloud shadows due to the lack of transport of diffuse radiation into these shadow regions, leading to an underestimation of the mean net surface irradiance. At high solar zenith angles on the other hand, i.e., when the Sun is close to the horizon, 1D solvers severely underestimate the size of these shadows, as they cast them directly underneath the clouds instead of at a slant angle, leading to an overestimation of the mean net surface irradiance. As we can see in Fig. 10, both of these effects cancel out at an 530    angle of about $50°$, which is the one we use, resulting in the almost perfect MBE of the $\delta$-Eddington approximation in the solar spectral range in Fig. 9. Despite this coincidence, however, Fig. 10 also shows us that the original TenStream solver performs slightly worse than the $\delta$-Eddington approximation for any zenith angle below about $50°$. However, the difference in the MBEs between the two solvers is quite small, and the magnitude of their respective RMBEs does not get much larger than $-1\,\%$ for any angle below $50°$ (not shown here).

The dynamic TenStream solver though underestimates surface absorption in the solar spectral range even more than the original TenStream solver does, with the effect increasing the less frequently the new solver is called. Especially looking at the runs with calling frequencies of 10 s and 30 s, one can however clearly see that this divergence from the original TenStream MBE quickly stabilizes itself at values around $-5\,\mathrm{W\,m^{-2}}$ ($-1\%$) and $-12\,\mathrm{W\,m^{-2}}$ ($-2\%$), indicating that the bias will not

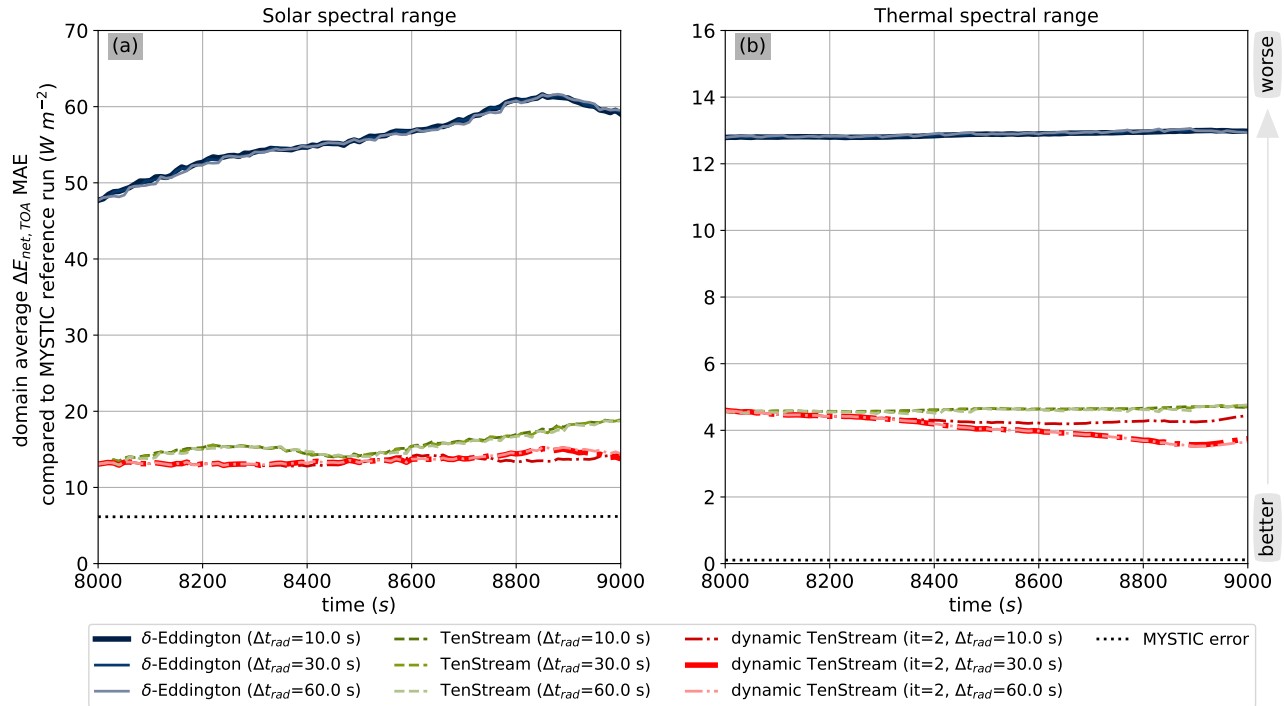

**Figure 11.** Temporal evolution of the mean absolute error in the net irradiance at top of atmosphere for the different solvers with respect to the MYSTIC benchmark run at calling frequencies of 10 s, 30 s and 60 s for both the solar (panel a) and thermal (panel b) spectral range. The MAE of the MYSTIC benchmark run itself is visualized by the dotted black line.

grow continuously. The same behavior can be observed in the thermal spectral range, only that, similar to the behavior in
the heating rates, the build-up of the bias compared to the original TenStream solver actually improves the MBE of the new solver at lower calling frequencies there. Since net surface irradiances in the thermal spectral range are negative, the positive MBE values for the original TenStream solver in Fig. 9 indicate an underestimation, i.e., not negative enough values, in the net surface irradiance, with the dynamic TenStream solver counteracting this bias the less often it is called – although this is, of course, more of a coincidence.
Finally coming to the upper boundary of our domain, Fig. 11 shows the temporal evolution of the MAE in the net irradiance at top of atmosphere (TOA) in an otherwise similar fashion as Fig. 8. Again, the incomplete solves in the dynamic TenStream solver lead to a slight divergence of the MAE of this solver (red lines) compared to the original TenStream solver (green lines) in both spectral ranges. However, this divergence remains small compared to the difference between the 3D TenStream solver and the 1D $\delta$-Eddington approximation, even at the lowest investigated calling frequency of 60 s. This indicates that also at
TOA, the dynamic TenStream solver is much better in capturing the spatial structure of the net irradiances than the traditional $\delta$-Eddington approximation is.

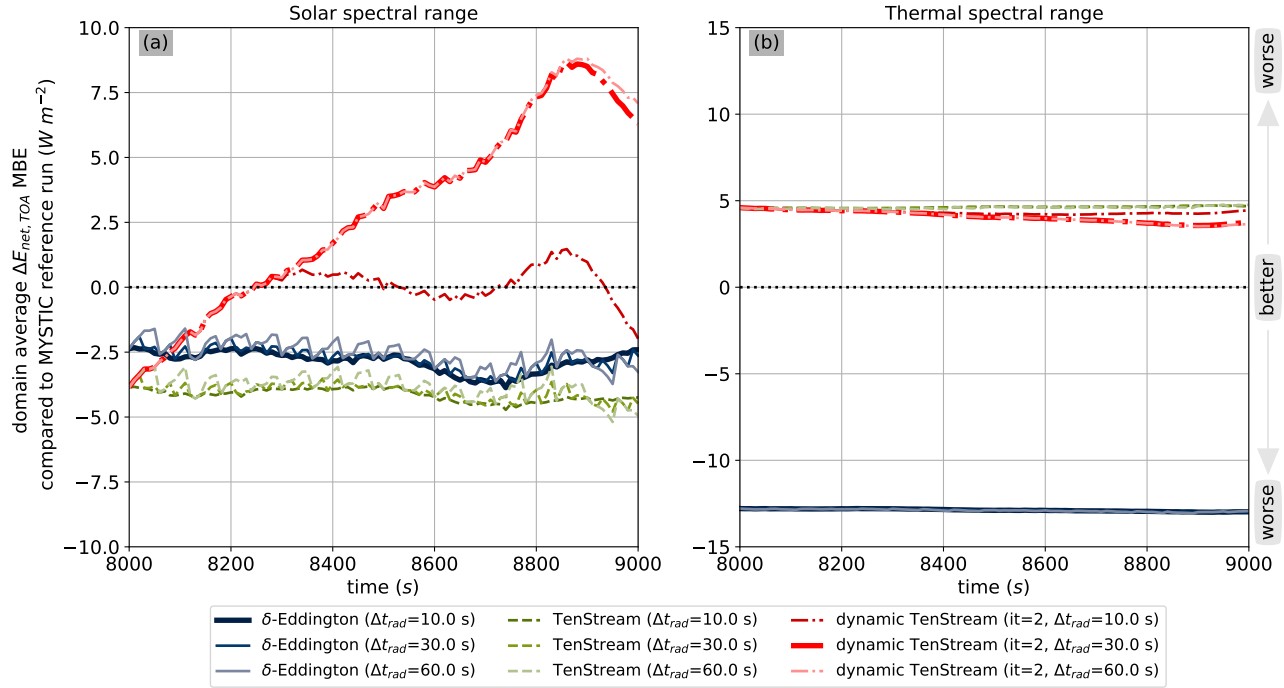

**Figure 12.** Temporal evolution of the mean bias error in the net irradiance at top of atmosphere for the different solvers with respect to the MYSTIC benchmark run at calling frequencies of 10 s, 30 s and 60 s for both the solar (panel a) and thermal (panel b) spectral range. A run with no bias is indicated by the dotted black line.

Similar to the surface, however, this does not fully apply in terms of domain averages. The corresponding temporal evolution of the MBE is shown in Fig. 12. Starting with the thermal spectral range displayed in panel (b), our new solver again just shows a comparatively small divergence from the original TenStream solver with time and performs significantly better than the $\delta$-Eddington approximation throughout the entire time series, regardless of the calling frequency used. In the solar spectral range, however, the original TenStream solver already performs a bit worse than the $\delta$-Eddington approximation does with time-average MBEs of about $-4\,\mathrm{W\,m^{-2}}$ for the TenStream solver compared to $-3\,\mathrm{W\,m^{-2}}$ for the 1D solver. More noticeably though, the incomplete solves in the dynamic TenStream solver lead to a fairly pronounced divergence in terms of the MBE from the original TenStream solver when compared to the difference between the 1D and original TenStream solvers. However, for every calling frequency investigated, this divergent behavior peaks at values that translate to RMBEs no larger than 1.25 % (not shown here). Taking both domain boundaries into account, we can thus draw similar conclusions as for the heating rates:

1. On the grid box level, our new solver determines far better net irradiances at both the surface and TOA than current 1D solvers do, even when operated at much lower calling frequencies.

2. Looking at domain averages, however, the incomplete solves within the dynamic TenStream solver lead to the build-up of a bias with time. In terms of magnitude relative to the original TenStream solver, this bias becomes larger the lower the calling frequency is and exceeds the bias of current 1D solvers, especially in the solar spectral range.

## 4.4 Dependence on the number of Gauß-Seidel iterations

So far, we have just looked into dynamic TenStream runs performed with only two Gauß-Seidel iterations whenever the solver is called. We focused on this computationally affordable setup as it already lead to promising results. To investigate how the results presented so far change when applying more than two Gauß-Seidel iterations, we have performed nine additional runs using integer multiplies of two Gauß-Seidel iterations, i.e., up to 20 iterations per call. Following the explanation given in Sect. 3.2, we use integer multiples of two instead of one in order to ensure that information is not preferably transported into one specific direction of the domain.

**Table 2.** Computing time of dynamic TenStream runs with $N$ Gauß-Seidel iterations per call relative to those with 2 Gauß-Seidel iterations, taken as an average over three runs performed on the same workstation for the very first time step of the LES cloud time series

| number $N$ of Gauß-Seidel iterations | 2 | 4 | 6 | 8 | 10 | 12 | 14 | 16 | 18 | 20 |
|---|---|---|---|---|---|---|---|---|---|---|
| **solar spectral range** | 1.0 | 1.2 | 1.4 | 1.6 | 1.7 | 1.9 | 2.1 | 2.3 | 2.5 | 2.6 |
| **thermal spectral range** | 1.0 | 1.1 | 1.3 | 1.4 | 1.5 | 1.6 | 1.8 | 1.9 | 2.0 | 2.2 |

In order to evaluate the improved performance of these additional runs, it is important to have a rough estimate of their additional computational cost. Therefore, we have measured the computing time of these runs exactly as we did it in Sect. 4.1 for all the other solvers. Table 2 shows these computing times relative to a calculation with two Gauß-Seidel iterations per call. As we can see, using four instead of two iterations does not double the computational cost, as there is a considerable amount of overhead that always takes the same amount of time before even starting with the Gauß-Seidel iterations, such as retrieving the TenStream coefficients from the corresponding look-up tables. However, apart from this offset, computing time scales roughly linearly with the number of Gauß-Seidel iterations, as two more iterations always add about 10 % to 20 % of the baseline cost of a calculation with two Gauß-Seidel iterations to the computing time. This fraction is smaller for the thermal spectral range because of a larger overhead due to the additional calculation of thermal emission.

Having this additional computational burden in mind, we can now have a look at Fig. 13. Panels (a) and (b) in this figure show the time- and domain-average MAE of the dynamic TenStream solver in heating rates for the shallow cumulus cloud time series as a function of the number $N$ of Gauß-Seidel iterations. Correspondingly, the values at the very left at $N = 2$ are the time-averages of the sawtooth curves in Fig. 6 for the corresponding calling frequencies. The dashed lines represent the temporal mean MAEs for the original TenStream solver. The MAEs of the dynamic TenStream solver converge towards these dashed lines in the limit of a large number of iterations. For lower calling frequencies, this limit the dynamic TenStream solver is converging to is larger than it is for higher calling frequencies, because the solver is called less often, leading to the

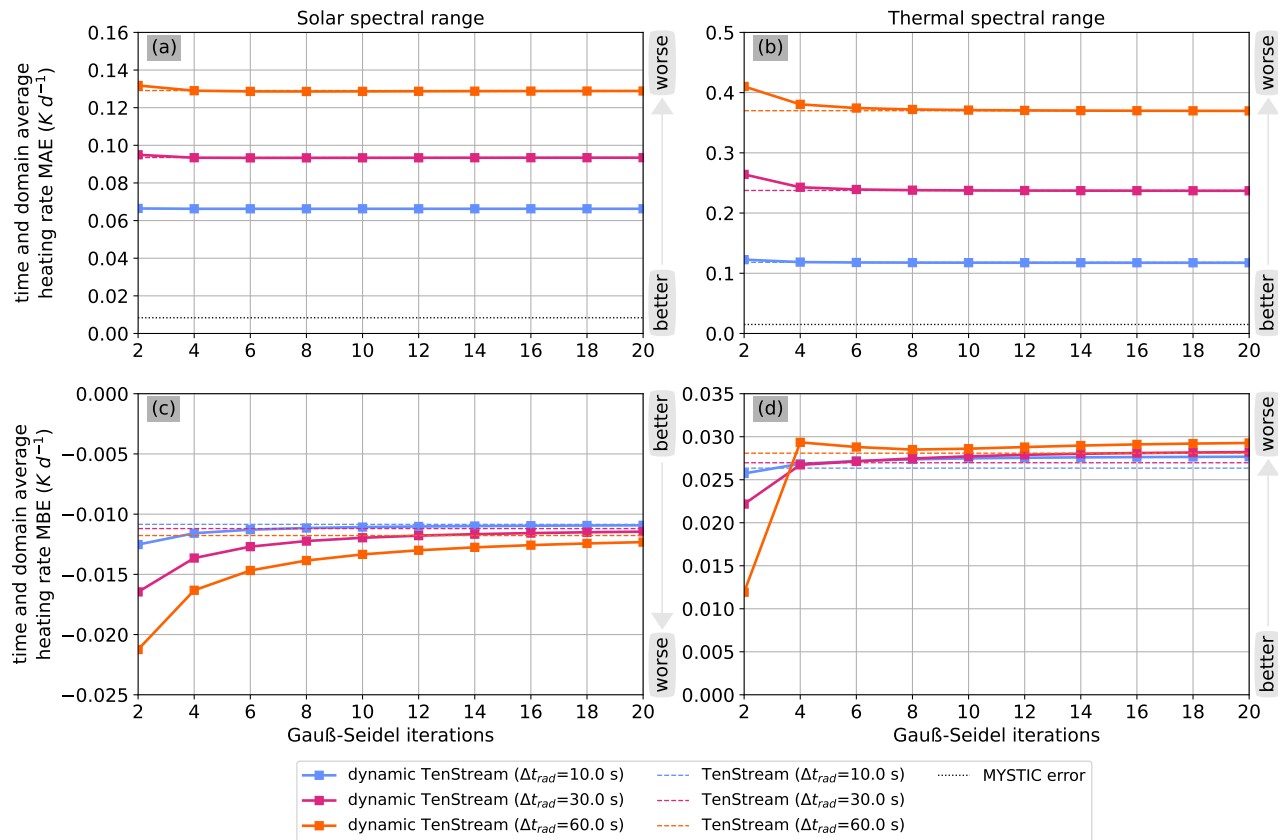

**Figure 13.** Time- and domain-average mean absolute error (panels a and b) and mean bias error (panels c and d) in heating rates with respect to the MYSTIC benchmark run as a function of the number of Gauß-Seidel iterations used in the dynamic TenStream solver for both the solar (left panels) and thermal spectral range (right panels). The three different colors show the errors for calling frequencies of 10 s (blue), 30 s (purple) and 60 s (orange). Solid lines connect the values for the dynamic TenStream solver, while the constant dashed lines represent the errors of a full TenStream solve at the corresponding calling frequency, towards which the dynamic TenStream values are converging to. In panels (a) and (b), the MAE of the MYSTIC benchmark run itself is visualized by the black dotted line.

build-up of a larger MAE with time until the solver is eventually called again, as we have seen in Fig. 6. Since the MAEs of the dynamic TenStream solver were already almost on par with the original TenStream solver when using just two Gauß-Seidel iterations, the MAE is already nearly converged at $N = 2$ and does not greatly improve when using more iterations. It is only in the thermal spectral range and at lower calling frequencies that we see a slight improvement in the mean MAE when applying more iterations, especially by doubling the number of iterations from two to four.

In contrast to the MAE, however, we observed a noticeable build-up of bias with time for the dynamic TenStream solver that is larger the less the solver is called. Consequently, the MBE in panels (c) and (d) of Fig. 13 starts at values significantly apart from convergence at $N = 2$, especially for the lowest two calling frequencies. The more Gauß-Seidel iterations we apply, the

more this difference in bias compared to the original TenStream solver disappears. We can also see that the incidentally better bias of our new solver in the thermal spectral range at a calling frequency of 60 s quickly converges towards the bias of the original TenStream solver, as dynamic TenStream is simply based on this solver. To evaluate whether it is worth to decrease the magnitude of the MBE compared to the original TenStream solver by applying more iterations, let us have a look at the additional computational cost of these iterations in Table 2. Using four instead of two Gauß-Seidel iterations only adds 10 % to 20 % to the total computational time, while leading to a noticeable decrease in the both the MAE and especially the MBE. In this regard, one could even think about calling dynamic TenStream less frequently, but with more Gauß-Seidel iterations. As we have seen in Sect. 4.1, using our new solver at a calling frequency of 30 s is about as expensive as calling a $\delta$-Eddington approximation every 10 s. Taking Table 2 into account, we can see that using $N = 20$ instead of $N = 2$ iterations is a bit more than twice as expensive. Hence, we could argue that a dynamic TenStream configuration with $N = 20$ at a lower calling frequency of 60 s also imposes about the same computational cost as a $\delta$-Eddington approximation at a calling frequency of 10 s. However, while such a setup would lead to a better time-average MBE than our configuration with $N = 2$ and a calling frequency of 30 s, it would also lead to a very noticeable increase in the mean MAE. To put it figuratively, using more iterations at a lower calling frequency reduces the bias, but at the expense of the spatially correct representation of the heating rates. In terms of these heating rates, we can thus draw two main conclusions:

1. Using more Gauß-Seidel iterations per call primarily counteracts the build-up of a bias with time, as the incomplete solves with two Gauß-Seidel iterations per call already resemble the spatial structure of the full TenStream results very accurately.

2. When using more Gauß-Seidel iterations, but a lower calling frequency in order to maintain the total computational cost, one improves the representation of domain averages at the expense of the spatial structure of the results.

Especially at the surface, however, one should definitely think about using more than just two Gauß-Seidel iterations per call. To motivate that, Fig. 14 shows the same plots as Fig. 13, but for net surface irradiances instead of heating rates. As for the heating rates, we can see that the use of more than two Gauß-Seidel iterations per call primarily counteracts the build-up of the MBE with time. In contrast to the heating rates, however, lower calling frequencies do not impact the magnitude of the MAE as much. This indicates that even at lower calling frequencies, the dynamic TenStream solver is able to adequately capture the spatial structure of the net surface irradiances. Consequently, using our new solver with $N = 20$ iterations at a calling frequency of 60 s leads to better results than at $N = 2$ and a 30 s calling frequency here – both in the solar, as well as in the thermal spectral range.

We can thus conclude that even though the computationally most affordable runs using just two Gauß-Seidel iterations per call lead to promising results, it might be beneficial to use configurations involving slightly more iterations, as they add a comparatively small additional computational cost to the solver while significantly counteracting the build-up of a bias with time. The results for the net irradiance at top of atmosphere only underline the statements for the surface and are thus not shown in here.

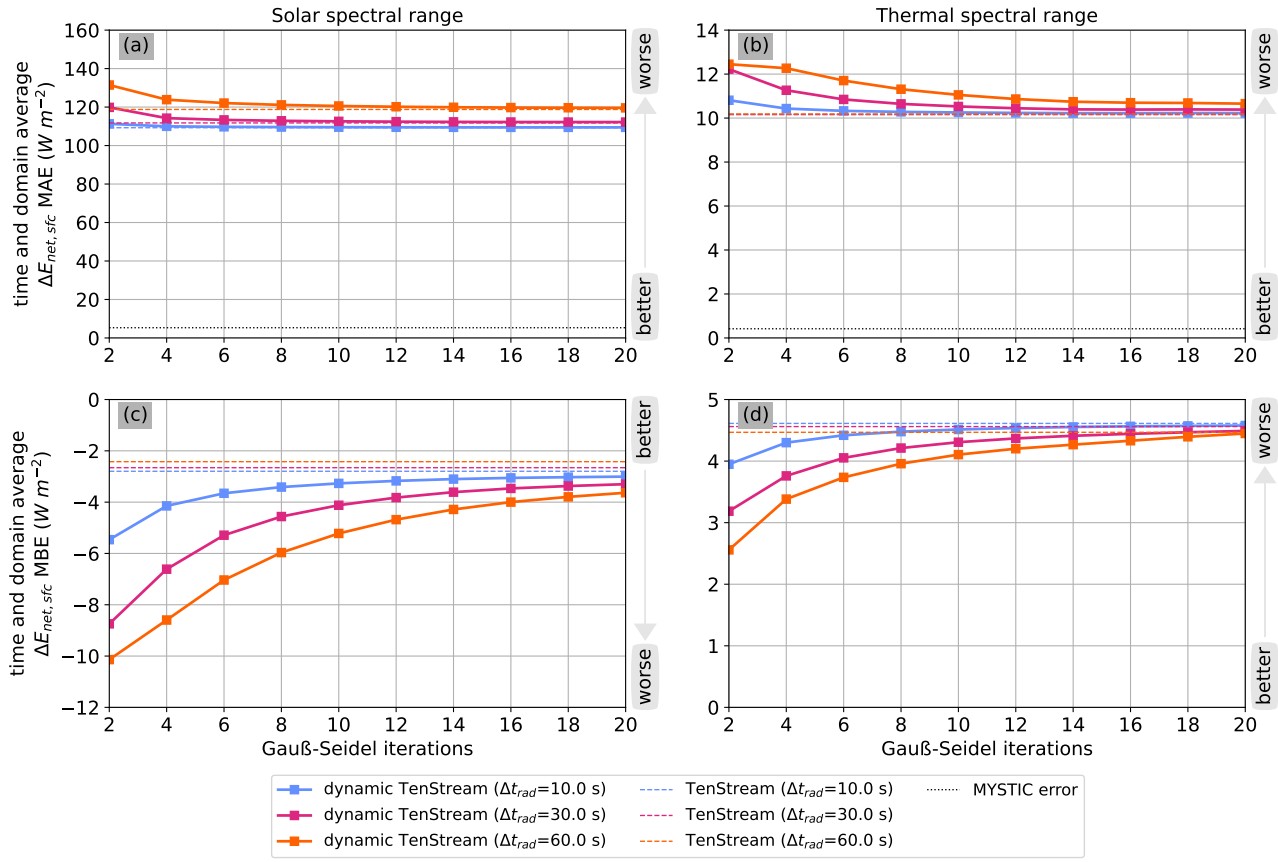

**Figure 14.** Time- and domain-average mean absolute error (panels a and b) and mean bias error (panels c and d) in the net surface irradiance with respect to the MYSTIC benchmark run as a function of the number of Gauß-Seidel iterations at calling frequencies of 10 s, 30 s and 60 s. Solid lines connect the values for the dynamic TenStream solver, while the constant dashed lines represent the errors of a full TenStream solve at the corresponding calling frequency. In panels (a) and (b), the MAE of the MYSTIC benchmark run itself is visualized by the black dotted line.

## 4.5 Visualization of dynamic TenStream heating rate fields

We want to conclude our evaluation by visually comparing the dynamic TenStream results to those calculated by the other solvers introduced in Sect. 3.2. In contrast to the previous section, we restrict ourselves to dynamic TenStream runs with just two Gauß-Seidel iterations per call here. A special focus of this comparison will be on how well dynamic TenStream visually

performs in updating the radiative field depending on the calling frequency. To make this comparison as hard as possible for our new solver, we decided to look at the last time step where the radiative field is simultaneously updated for all three calling frequencies that we consider: that is at $t = 8960$ s. Instead of this point in time, we could of course also take a look at a point in time where the different dynamic TenStream solves have just not been updated. By doing so, one would focus more on how

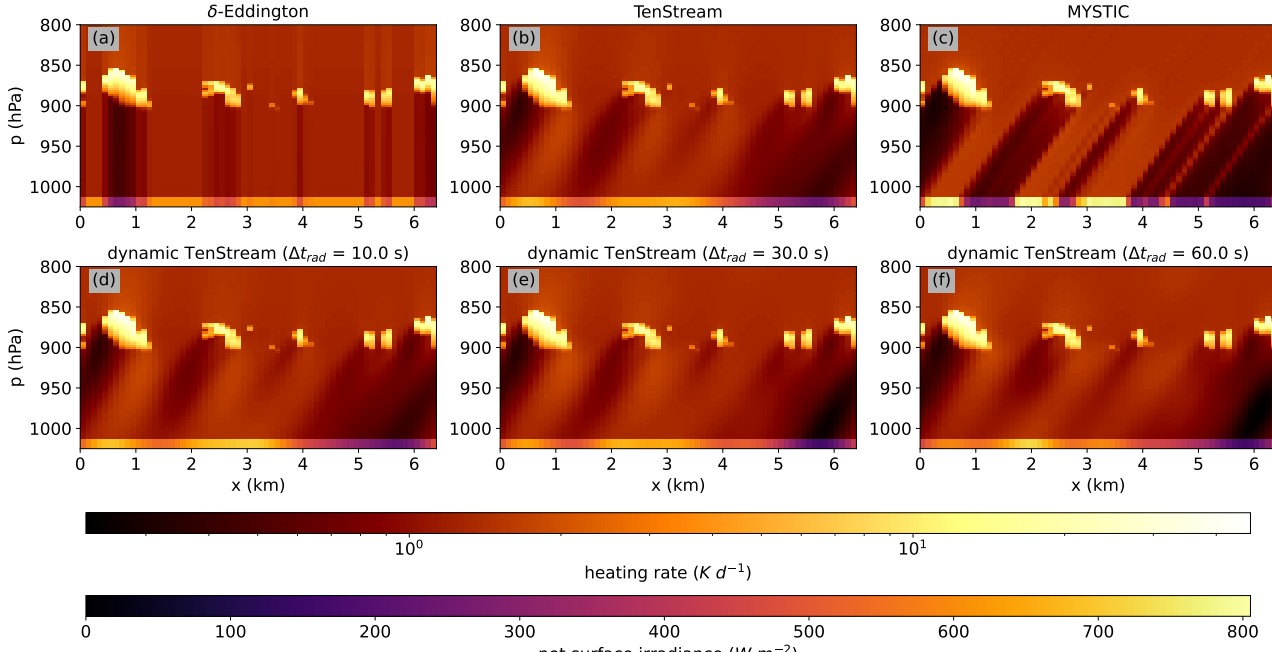

**Figure 15.** xz cross section of the heating rate fields obtained by the different RT solvers in the solar spectral range at $t = 8960$ s. Heating rates are visualized by a logarithmic color scale for the $\delta$-Eddington (panel a), original TenStream (panel b) and MYSTIC (panel c) solver, as well as for the dynamic TenStream solver when operated at calling frequencies of 10 s (panel d), 30 s (panel e) or 60 s (panel f). Additionally, the horizontal line at the bottom in all the plots visualizes the corresponding net surface irradiances obtained by the solvers.

good not yet updated radiative fields still resemble the benchmark result. Here, however, we want to focus more on how well
our new solver performs in updating the radiative field depending on how much it has changed before the radiation scheme
was called again. From this point of view, $t = 8960$ s is the last point in time where all three dynamic TenStream runs have just
been updated. Hence, they are subject to the most incomplete solves and furthest away from the initial spin-up there, enlarging
the chance for potential artifacts in the radiative field due to not fully solving the TenStream linear equation system for quite a
while.

Figure 15 shows xz cross sections for this point in time for the solar spectral range, with the colors indicating the heating
rates along the cross section using a logarithmic color scale – except for the lowermost row in all the panels, which visualizes
the net surface irradiance. In general, the bright yellow areas with correspondingly large heating rates indicate the position of
clouds, while the dark areas signify shadows below the clouds. Right from the start we can see that the largest visual differences
do not occur in between the different incomplete dynamic TenStream solves, but between the 1D $\delta$-Eddington approximation
in panel (a) and the 3D solvers in panels (b)–(f). As 1D radiation does not allow for horizontal transport of energy, shadows
in panel (a) cannot be cast according to the angle of solar incidence, but just right underneath the clouds. This also affects
absorption at the ground, with regions of low surface absorption located right below the clouds, rather than displaced like in

the MYSTIC benchmark run. We can see that the visual structure of this benchmark result is much better resembled by the TenStream solver shown in panel (b). Clouds here are also illuminated at their sides and horizontal transport of energy allows for shadows being cast in the direction of the solar incidence angle. However, we can see that both these shadows and regions of low surface absorption are much more diffuse than in the MYSTIC benchmark run – although they are still a much better representation of the benchmark than the 1D solution.

Having these characteristics in mind, we can now discuss the results for the new dynamic TenStream solver, that are shown in the last row of Fig. 15. The three panels show the results for the new solver if it has been called every 10 s (panel d), 30 s (panel e) or 60 s (panel f) before. At first glance, we can see that the new solver almost perfectly matches the results obtained by the original TenStream solver in panel (b), even when operated at the low calling frequency of 60 s. Remember that in this run just two Gauß-Seidel iterations towards convergence were carried out at only (8960 - 8000) / 60 = 16 points in time since the spin-up. Since our solver is based on the TenStream solver, this is almost the best result we could have obtained. We can see that just like the TenStream solver, dynamic TenStream allows for full three-dimensional transport of energy, with shadows and regions of low surface absorption being cast not just directly underneath clouds. Looking closely, one can however indeed see differences between the results obtained at different calling frequencies. Panel (d) showing the results for a calling frequency of 10 s most accurately resembles the original TenStream result, which becomes most visible within the shadows cast by the clouds on the right-hand side of the domain. They are overestimated by both lower calling frequency runs in between about 5 and 6 km in x-direction, with heating rates being too low there compared to the original TenStream result. Also, surface absorption differs quite a bit in between the different dynamic TenStream runs. The structure obtained by the original TenStream solver is again most accurately resembled by the dynamic TenStream run with a calling frequency of 10 s, whereas the surface absorption is overestimated a bit around 5 km in x-direction in the 30 s run and featuring a much more pronounced region of high absorption at around 2 km in the 60 s run.

Before making a closing statement, let us also have a look at the results in the thermal spectral range shown in Fig. 16. Again, we can see that the result for the 1D $\delta$-Eddington approximation in panel (a) features the most differences when compared to all the other panels showing results obtained by 3D solvers. Compared to the MYSTIC benchmark run, we can see that thermal shadows cast by the clouds are much more pronounced in 1D and not weakened in direction of the ground due to interaction with neighboring columns. This also leads to a very distinct pattern of strongly negative and not so negative net surface irradiance areas at the ground in the 1D results, whereas the net surface irradiance is almost uniform in the MYSTIC benchmark result. This also provides proof to our observation in Sect. 4.3, where we have noted that the benchmark results for the net surface irradiance in the thermal spectral range should be pretty uniform in order to avoid the sawtooth pattern in the MAE time series that we typically saw when evaluating solvers at lower calling frequencies. Furthermore, we can also see that the 1D $\delta$-Eddington approximation is not able to consider cloud-side cooling due to its lack of horizontal transport of energy, leading to a much more pronounced cloud bottom warming in the 1D results than in the MYSTIC benchmark. The original TenStream solver depicted in panel (b) is able to consider almost all of these 3D effects and is therefore once more visually close to the MYSTIC result. Looking closely, we can however see that the thermal shadows are a bit more pronounced there,

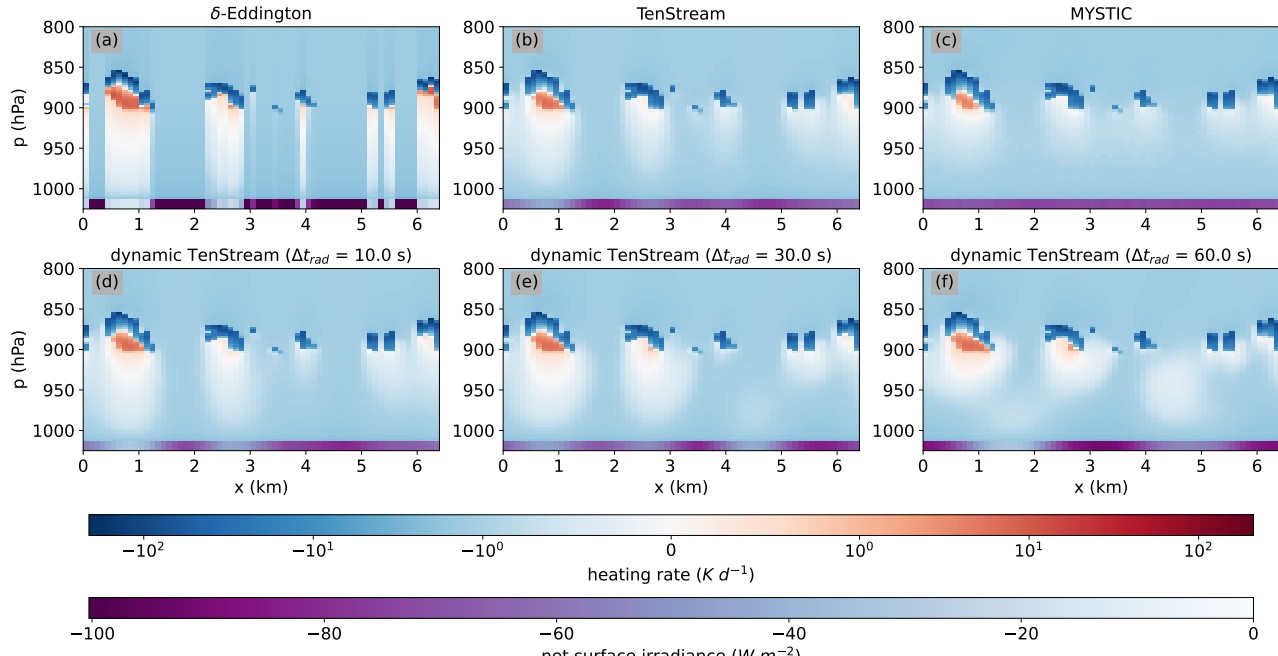

**Figure 16.** xz cross section of the heating rate fields obtained by the different RT solvers in the thermal spectral range at $t = 8960$ s. The structure of the plot is identical to Fig. 15, only that the color scale is logarithmic for heating rates both above 1 K d$^{-1}$ and below –1 K d$^{-1}$ and linear in between.

which also leads to regions of a bit weaker net surface irradiance below the clouds in contrast to the very uniform pattern produced by the MYSTIC benchmark solver.

Comparing these results to those of our newly developed dynamic TenStream solver, we can see that also in the thermal spectral range, it is almost able to reproduce the results of the original TenStream solver, even when operated at lower calling frequencies. However, the result obtained with a calling frequency of 10 s shown in panel (d) clearly resembles the original TenStream result most closely. At lower calling frequencies, we can see small artifacts, most noticeably in the form of larger or completely floating thermal shadows (the white areas in the plots) that do not seem to belong to any cloud at all, while they are normally placed directly underneath them. These regions are residual shadows of already dissolved clouds, which the

incomplete solves where not able to get rid of yet. Evidence for this hypothesis is provided by looking at the same plot at previous time steps (not shown here). These residual shadows also influence the net surface irradiance pattern, which is most prominently visible in between 1 and 2 km in panel (f). In total, these residual shadows are minor artifacts, though, as we have to consider that we were only able to visualize them by using a logarithmic color scale. And we also have to keep in mind that especially panel (e) showing the results at a calling frequency of 30 s has been obtained using a similar computational demand

like performing 1D $\delta$-Eddington calculations every 10 s. In contrast to these results, however, the dynamic TenStream result

features horizontal transport of radiative energy, resulting in much more realistically distributed heating rates and net surface irradiance patterns.

In summary, we can hence say that for both the solar and the thermal spectral range, dynamic TenStream is almost perfectly able to visually reproduce the results obtained by the original TenStream solver, even when operated at lower calling frequencies. At those, however, minor artifacts like residual shadows are introduced. The reason for these artifacts are the incomplete solves, which can delay lower-order 3D effects, such as feedback effects from other clouds or the surface. The term "feedback effects" thereby refers to the fact that the 3D radiative effects of a cloud can theoretically alter the conditions determining the 3D radiative effects of any other cloud in the domain. Because these feedback effects require multiple back and forth transports of information, they cannot be fully accounted for when solving radiation incompletely. For example, incomplete solves can perfectly consider three-dimensional radiative effects of an emerging cloud at the location of the cloud itself, but the feedback on these heating rates due to lower upward facing radiative fluxes from the shadow this cloud casts may be delayed to a later call of the scheme, if the two Gauß-Seidel iterations that we perform per call are not sufficient to transport this feedback back to the cloud itself.

## 5  Summary and outlook

Based upon the TenStream solver, we presented a new radiative transfer model currently designed for the use at subkilometer-scale horizontal resolutions that allows us to calculate 3D radiative fluxes and heating rates at a significantly increased speed by utilizing two main concepts that both rely on the idea that the radiative field does not completely change in between two calls of the scheme: First, radiation in this method is not solved from scratch every time it is called, but rather uses a time-stepping scheme to update the radiative field based on the result from the previous radiation time step. Secondly, the model is based on incomplete solves, performing just a few Gauß-Seidel iterations towards convergence every time it is called.

To demonstrate the feasibility of the dynamic TenStream solver incorporating these two concepts, we implemented it into the libRadtran library for radiative transfer and applied it onto 100 time steps of a shallow cumulus cloud time series prepared by Jakub and Gregor (2022). Its high temporal resolution of 10 s allowed us to investigate the effect of the calling frequency on the performance of our new solver by comparing results obtained at this high calling frequency to those retrieved at lower calling frequencies, where the radiative field changes more noticeably in between two time steps. Four different solvers were applied to this time series: Besides our newly developed dynamic TenStream solver with a low number of two Gauß-Seidel iterations per call, a traditional 1D $\delta$-Eddington approximation was used as a worst-case benchmark that we should definitely surpass, whereas the original TenStream solver served as a best-case benchmark – since our new solver is based on the TenStream solver, retrieving the exact same results while relying on incomplete solves would have been the best outcome that we could have expected. Simulations performed by the 3D Monte Carlo solver MYSTIC furthermore served as benchmark results for all the solvers, essentially providing a ground truth.

Using these results, we evaluated the performance of our new solver in determining heating rates and net irradiances at the upper and lower domain boundaries both in terms of speed and accuracy. In terms of speed, we saw that the dynamic TenStream

solver is about three times slower than a traditional 1D $\delta$-Eddington approximation, but delivers a noticeable increase in
performance when compared to the other two 3D solvers, which are at least a factor of 5 slower. To evaluate the accuracy of
the aforementioned solvers with respect to the MYSTIC benchmark run, we used two different error measures: While a mean
absolute error allowed us to investigate the average error a certain solver makes for an individual grid box, a mean bias error
allowed us to observe whether the domain-average results of a certain solver deviate from the domain-average benchmark
results. In terms of heating rates, we saw that our new solver is almost perfectly able to reproduce the results of the original
TenStream solver, even when operated at lower calling frequencies. At these lower calling frequencies, we observed that our
incomplete solves lead to the build-up of a bias with time that is the larger the lower the calling frequency is. However, even
at the lowest calling frequency investigated, this build-up stabilized itself at some point and remained lower than the bias of
any 1D run in any point in time. More importantly, time-average dynamic TenStream results were better in terms of both error
measures when compared to 1D simulations carried out with a similar computational demand. Next, we saw that mean absolute
errors in the net irradiance at the top and bottom of the domain were also significantly lower than the corresponding 1D errors,
even when operated at lower calling frequencies. Only the bias in the net irradiances was larger than in the 1D simulations at
any point in time. However, this bias could already be observed in the original TenStream results and was thus not expected to
be improved by applying incomplete solves. Finally, we observed that using more than just two Gauß-Seidel iterations per call
primarily counteracts the build-up of this bias with time at a computationally relatively small cost.

Overall, the results of this test case clearly demonstrated the capabilities of the dynamic TenStream solver. Using a first
example, we were able to show that the introduction of a time-stepping scheme and the application of incomplete solves
are able to retrieve both heating rates and net irradiances at the upper and lower domain boundaries that are much closer to
the 3D benchmark results than to currently employed 1D solvers, even when operated with a similar computational demand.
These results become even more interesting when we consider recent developments in the field of another major computational
bottleneck in radiative transfer calculations, namely the number of quadrature points required to calculate accurate integrated
longwave and shortwave heating rates. In our evaluation, we used the wavelength parameterization by Fu and Liou (1992, 1993)
that features a total of 54 and 67 quadrature points in the solar and thermal spectral range, respectively (Oreopoulos et al.,
2012). That is already a pretty low number considering that most models currently use the newer and more precise RRTMg
parameterization (Mlawer et al., 1997; AER, 2024), that even takes a total of 112 and 140 quadrature points in the solar and
thermal spectral range into account, respectively. However, recent developments showed that these numbers can be dramatically
reduced without a significant loss in precision in the calculation of both radiative fluxes and heating rates. de Mourgues et al.
(2023) for example showed that in the thermal spectral range, even 30 quadrature points are sufficient to almost perfectly
reproduce heating rates obtained by a line-by-line calculation. Compared to RRTMg, this is more than four times less the
number of quadrature points. Similar results have been obtained by Hogan and Matricardi (2022), who showed that just 32
quadrature points in both the solar and the thermal spectral range are able to produce very accurate irradiances and heating rates,
with more quadrature points adding little to no further precision. These more efficient spectral parameterizations, together with
the speed improvements achieved with the dynamic TenStream solver, would allow to accelerate 3D radiative transfer towards

the speed of currently employed 1D solvers, potentially for the first time ever allowing the use of 3D radiative transfer in NWP models.

Before this vision becomes reality, however, the dynamic TenStream solver needs more work. First of all, further performance tests should include multiple layer cloud fields – e.g. shallow cumulus clouds with cirrus clouds above – as well as deep convective clouds to investigate whether two Gauß-Seidel iterations per call as used in this paper are still sufficient under these circumstances, as more complex cloud fields involve more radiative interaction in the vertical. Earlier simulations carried out with the dynamic TenStream solver have shown that incomplete solves can lead to "ping-pong" effects in these cases,

where distant grid boxes update radiative influences on each other back and forth in between different dynamic TenStream calls. These "ping-pong" effects were vastly reduced due to the use of the Gauß-Seidel method, but it will be interesting to see whether vertically more complex cloud fields pose a greater challenge to our solver. In addition, the derivation of a rule on how many Gauß-Seidel iterations to use depending on the model setup to ensure reliable results is another main future target. In this context, it would also be interesting to investigate whether occasional full solves are a computationally feasible means of

ensuring that the results of the dynamic TenStream solver do not deviate too much from those of the original TenStream solver. Additionally, we could think about an even more sophisticated first guess for the incomplete solves by advecting the radiative field of the previous time step with the rest of the atmospheric fields. As we assume that the radiative field does not totally change in between two different calls of the radiation model, such a first guess should already better account for the updated position of the clouds, so that the incomplete solves could primarily focus on correcting for the changed optical properties of

the clouds, which could speed up convergence even more. Coupled to dynamics, it will also be very interesting to investigate how the incomplete solves in the dynamic TenStream solver influence the development of clouds. Finally, going to the NWP scale, we will certainly need to consider sub-grid scale cloud variability, for example by extending the TenStream look-up tables to account for cloud fraction. And on top of that, we will certainly have to parallelize the solver in order to make it computationally more efficient. As in the development of this new solver, we can build upon the original TenStream solver for

that, as this solver is already fully parallelized (Jakub and Mayer, 2015) and can be interactively coupled to LES models (Jakub and Mayer, 2016). Using the native TenStream framework is especially suitable for these purposes since the main features of the new dynamic TenStream solver – among them the ability to perform incomplete solves, the correct calculation of 3D heating rates in this case, as well as the speed-up in convergence by properly iterating through the underlying linear equation system – have already been included as options for the native TenStream solver in the meantime.

*Code and data availability.* The newly developed dynamic TenStream solver presented in this paper was developed as part of the libRadtran library for radiative transfer (Emde et al., 2016) and can be accessed via Mayer et al. (2023). The user manual can be found in the "doc" folder of the library. The shallow cumulus cloud time series used to evaluate the performance of the new solver has been published by Jakub and Gregor (2022), with the modifications and methods applied to it to reproduce the results of Sect. 4 described in Sect. 3 of this paper.

*Author contributions.* RM developed the dynamic TenStream solver and carried out its evaluation with extensive support of BM and FJ. BM

contributed the foundational idea of the solver and the concept behind the correct calculation of three-dimensional heating rates, while FJ

provided a lot of technical support regarding the original TenStream solver, the methods to access the TenStream look-up tables, the iterative

method used in the solver as well as methods to evaluate the performance of the new solver. CE, MM and AV took part in regular discussions

regarding the development of the dynamic TenStream solver. RM prepared the manuscript with contributions from all co-authors.

*Competing interests.* The authors declare that they have no conflict of interest.

*Acknowledgements.* The research leading to these results has been done within the subproject "B4 – Radiative interactions at the NWP

scale and their impact on midlatitude cyclone predictability" of the Transregional Collaborative Research Center SFB / TRR 165 "Waves to

Weather" (www.wavestoweather.de) funded by the German Research Foundation (DFG).

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
