# Peer review of "A dynamic approach to three-dimensional radiative transfer in subkilometer-scale numerical weather prediction models: the dynamic TenStream solver v1.0"

_EGUsphere, 2023_

## Author Comment (AC2)

**Response to Review Comment 4 (RC4)**

**Manuscript:** egusphere-2023-2129
**Title:** A dynamic approach to three-dimensional radiative transfer in numerical weather prediction models: the dynamic TenStream solver v1.0
**Authors:** Richard Maier, Fabian Jakub, Claudia Emde, Mihail Manev, Aiko Voigt, and Bernhard Mayer

We thank Anonymous Referee #3 for his or her comments on our manuscript, which we will respond to below. To structure our response, the referee's comments are printed on a gray background color, while our answers are displayed on ordinary white background.

* General comments:

This is a welcome update of the TenStream 3D radiative transfer (RT) code that already fills a major gap in LES modeling capability, namely, to perform 3D RT broadband radiation budget estimation for Large-Eddy Simulation (LES) models. LES is now routinely used in cloud-scale process modeling to address some of predictive climate science's most stubborn issues, such as cloud feedbacks and aerosol-cloud interactions.

However, in spite of generating fully 3D (i.e., vertically-developed) clouds driven by convective dynamics, the RT parameterizations used in LES are still too often heritage codes from Global Climate Models (GCMs) where nothing less than ~50 to 100 km in scale is resolved, hence all clouds and many cloud systems. A typical aspect ratio for a GCM cloudy column is therefore on the order of 1-to-10, thus, some form of 1D RT that captures the internal variability of the clouds (e.g., McICA) is justified since little radiation will be leaked through the horizontal boundaries anyway. In sharp contrast, a cloudy column in an LES has the opposite aspect ratio: say, 5 km by 50 m, hence about 100-to-1. Even cloud-resolving models (CRMs), say, at 5 km by 0.5 km are 10-to-1. NWP models are heading into that kind of spatial resolution as well. So there is plenty of opportunity for net horizontal fluxes to develop across grid-cell facets, starting with direct shadowing of neighboring cells in the anti-solar direction. The TenStream model is purposefully designed to account for this 3D RT in terms of radiation energetics, hence fluxes, not radiances, as required for computing heating rates profiles and net fluxes through the top and bottom boundaries.

The new _dynamic_ TenStream model is designed to address the issue of computational efficiency that is in the way of the general acceptance of TenStream in the LES community for operational implementation. Specifically, it brings CPU time allocation down to ~3x the baseline cost of 1D RT, and does so by cutting a few corners, which could carry a cost in accuracy. Therefore, dynamic TenStream is benchmarked for accuracy against the original TenStream, as well as 1D RT (delta-Eddington) and full 3D RT (MYSTIC). Its accuracy is at par with the original TenStream, which is already a vast improvement in accuracy for radiation budget estimation using standard 1D RT.

The paper is well written and illustrated. It should be published by GMD after a minor revision that addresses the following issues.

* Specific comments:

(1) Careful attention is paid to the heating-rate profile and surface irradiance/flux. However, it seems to me that the outgoing TOA flux is also important. Maybe TenStream enforces radiant

energy conservation is such a way that the TOA flux is as accurate as the rest, but that isn't obvious to this reviewer. At a minimum, some kind of statement on TOA flux accuracy is in order.

You are right that we focused our evaluation on heating rates and net surface irradiances, as they are the main drivers of the weather. For the revised version, we extended Section 4.3 to also account for the performance of the new solver in determining net irradiances at top of atmosphere (TOA). The content of this extension is centered around two new plots. The first one shows the temporal evolution of the mean absolute error (MAE) in the net irradiance at TOA in an otherwise similar fashion as Fig. 8 in the preprint:

[Figure]

The plot shows that also at TOA, the 1D delta-Eddington solver performs worst (blue lines) in terms of the MAE, with the original TenStream solver (green lines) once more being a noticeable improvement, remaining significantly below the error of all 1D runs in both the solar and the thermal spectral range. Our newly developed dynamic TenStream solver (red lines) shows just slight deviations from the MAE of the full TenStream calculations, almost independent of the calling frequency used, and thus also stays significantly smaller than the MAE of any 1D delta-Eddington run throughout the entire time series – even at the lowest calling frequency of 60 s.

However, similar to the results obtained for the net surface irradiance, the performance of both the original, as well as our new dynamic TenStream solver is worse in terms of our other error measure, the mean bias error (MBE). The temporal evolution of this error measure in terms of the net irradiance at TOA is shown in the other new figure below. It shows that in the solar spectral range, the MBE for the new dynamic TenStream solver (red lines) clearly diverges from the MBE of the original TenStream solver (green lines). This spread from the original TenStream solver gets significantly larger at a calling frequency of 30 s compared to the dynamic TenStream run at a calling frequency of 10 s. Interestingly, however, the spread does not further increase when calling dynamic TenStream even more infrequently (bright red line). And in both cases, the MBE of the dynamic TenStream runs does not continuously increase, but stabilizes itself at some point in time.

And even for calling frequencies of 30 s and 60 s, the MBE peaks at values of around 8.5 W m$^{-2}$, which translates into a RMBE of about 1.2 % (not shown here).

[Figure]

In the thermal spectral range, on the other hand, the MBE in the net TOA irradiance for both the original, as well as the new dynamic TenStream solver stays significantly below the error of the 1D delta-Eddington runs throughout the entire time series, peaking at values of around 5 W m$^{-2}$ (-2 %) for the 3D solvers compared to -13 W m$^{-2}$ (5 %) for the delta-Eddington solver.

(2) The temporal down-sampling and the incomplete solves naturally cause the new model to drift away from the original counterpart. Would it not be beneficial to occasionally "reset" this drift to zero by calling the original TenStream? Of course there is a whole study to perform about when to do this operationally, without the benchmark information at hand.

This is a good idea, and one that we definitely had in mind when thinking about future couplings of our new solver to LES or NWP models. The implications of such resets would be relatively straightforward, as the error metrics would simply reduce to those of the original TenStream solver whenever such a reset was performed. For this paper, however, we wanted to focus on how our new solver performs when applied with the lowest computational cost possible – that is, with a low number of two Gauß-Seidel iterations per call and no intermediate resets of the new model.

In the future, however, it would certainly be interesting to investigate the trade-off between increased accuracy due to occasional model resets on one side, and the additional computational cost that these resets introduce on the other side. We have included this thought in the outlook of the revised version of the paper: "In this context, it would also be interesting to investigate whether occasional full solves are a computationally feasible means of ensuring that the results of our new solver do not deviate too much from those of the original TenStream solver."

(3) Although it should have been done when documenting the original TenStream model, it would be good to look into the past to find models with similar mathematical structure in terms of radical angular simplification compared to standard 3D RT solvers, more precisely with improved efficiency in mind. Can I suggest a few?

- an original "6-flux" model, applied to homogeneous plane-parallel media (but with potential for heterogeneous media):

Chu, C.M. and Churchill, S.W., 1955. Numerical solution of problems in multiple scattering of electromagnetic radiation. The Journal of Physical Chemistry, 59(9), pp.855-863.

- a discrete-angle RT formalism predicated on regular tessellations of 2D and 3D spaces, seeking the minimal number of directions to capture 3D RT effects:

Lovejoy, S., Davis, A., Gabriel, P., Schertzer, D. and Austin, G.L., 1990. Discrete angle radiative transfer: 1. Scaling and similarity, universality and diffusion. Journal of Geophysical Research: Atmospheres, 95(D8), pp.11699-11715.

- a 2D (4-stream) RT model in a deterministic fractal medium, emphasizing numerical implementation (successive over-relaxation scheme):

Davis, A., Gabriel, P., Lovejoy, S., Schertzer, D. and Austin, G.L., 1990. Discrete angle radiative transfer: 3. Numerical results and meteorological applications. Journal of Geophysical Research: Atmospheres, 95(D8), pp.11729-11742.

- the same 2D (4-stream) RT model but in a random multifractal medium, emphasizing numerical implementation (Monte Carlo scheme):

Davis, A.B., Lovejoy, S. and Schertzer, D., 1991, November. Discrete-angle radiative transfer in a multifractal medium. In Wave Propagation and Scattering in Varied Media II (Vol. 1558, pp. 37-59). SPIE.

- vastly faster solution of the 4-stream model using sparse matrix inversion:

Lovejoy, S., Watson, B.P., Grosdidier, Y. and Schertzer, D., 2009. Scattering in thick multifractal clouds, Part II: Multiple scattering. Physica A: Statistical Mechanics and its Applications, 388(18), pp.3711-3727.

Thank you for these suggestions. We also think that these papers should have been primarily mentioned in the documentation of the original TenStream solver. Nonetheless, we included some of these papers into the introduction of the revised version of our paper.

* Technical corrections:

Title: The application to NWP models is both inspirational and aspirational. Here, however, the authors only get as far as LES, or CRM (100 m grid spacing). A more accurate title is in order.

You are certainly right with that. We will change the title to "A dynamic approach to three-dimensional radiative transfer in subkilometer-scale numerical weather prediction models: the dynamic TenStream solver v1.0" for the revised version.

l. 99: i.e., e.g., (need commas, I think)

We have changed the sentence containing this expression to clarify the next comment and added commas behind "i.e." and "e.g." elsewhere in the document.

l. 100: "n1" --> what is the "1" for?

We have clarified the meaning of the "1" by adding more explanation to the corresponding example: "For example, the emissivity $e_{0,i,j,k}$ of grid box (i,j,k) in upward direction is equal to the fraction of the downward facing radiative flux $\Phi_{1,i,j,k+1}$ that is absorbed on the way through that grid box, which in turn is one minus the sum of all fractions $a_{n1,i,j,k}$ of $\Phi_{1,i,j,k+1}$ exiting grid box (i,j,k), i.e. $e_{0,i,j,k} = 1 - \Sigma_{n=0}^{9} a_{n1,i,j,k}$, where $a_{n1,i,j,k}$ refers to the corresponding entries in the second column of matrix $T_{i,j,k}$."

l. 104: first "out" --> not italics

We have changed that as suggested.

Fig. 3: For SZA near 45 deg, one could use a diagonal sweep through the grid? Same for ~45 deg in azimuth? Admittedly more tricky to code, but it would follow more closely the propagation of direct sunlight. No? [...]

```
 ____  ____  ____  ____  ___  etc.

|     |     |     |     |     |

|  1  |  2  |  6  |  7  |  15

|     |     |     |     |     |

 ____  ____  ____  ____  ___

|     |     |     |     |     |

|  3  |  5  |  8  |  14 |

|     |     |     |     |     |

 ____  ____  ____  ____  ___

|     |     |     |     |     |

|  4  |  9  |  13 |     |

|     |     |     |     |     |

etc.
```

Thank you for this suggestion. Indeed, one could think about more sophisticated patterns of propagating through the model grid boxes in order to follow the propagation of direct radiation even more closely. However, it would likely not improve convergence, since direct radiation is only represented by three independent streams pointing in x, y and z direction in the dynamic TenStream solver. By properly sorting the resulting three loops due to solar incidence angle, one already ensures that the ingoing direct fluxes of any grid box are always updated before calculating the

corresponding outgoing fluxes – even at 45° zenith and azimuth angle. To illustrate that, let us look at a simplified version of Fig. 3 showing only direct streams:

[Figure]

Similar to Fig. 3, this sketch shows the first four steps of one Gauß-Seidel iteration in two dimensions only. In every step, ingoing fluxes are used to update the outgoing fluxes of the corresponding grid box (highlighted in grey). Grey arrows in contrast to black arrows indicate fluxes that have not yet been updated in this Gauß-Seidel iteration. We consider a solar zenith angle of 45° with the Sun shining from the top-right. We can clearly see that even with our not as sophisticated way of iterating through the domain, we always use already updated ingoing fluxes to update the corresponding outgoing fluxes – except for fluxes at the borders of the domain, that are subject to boundary conditions. Due to the definition of the direct streams in the solver, a diagonal sweep through the grid boxes would actually even slow down convergence, as we would not always use already updated ingoing fluxes following such a pattern, although these diagonal sweeps seem to follow the propagation of solar incidence more closely at first.

l. 190: "this direction" -->? horizontal scan

We actually reverse the iteration direction in every other Gauß-Seidel iteration in all three dimensions. To clarify that, we have adjusted the corresponding sentence: "Thus, we reverse the direction of iteration in every other Gauß-Seidel iteration in all three dimensions to not favor propagation of information in one direction."

S. 3.1 (beginning): specify domain size in cells _and_ km

We have changed that as suggested: "The data set originally features both a high temporal resolution of 10 s and 256 × 256 grid boxes with a high spatial resolution of 25 m in the horizontal."

l. 262: specify domain height (in km too)

As we pointed out in the paper, the total domain is constructed using two different sources: For the first 220 layers, we use the high vertical resolution of 25 m provided by the LES runs. We clarified the domain height of that part referred to in l. 262: "In the vertical, the modified cloud data set consists of 220 layers with a constant height of 25 m, thus reaching up to a height of 5.5 km."

Above this vertically highly resolved grid, we use atmospheric levels provided by the 1976 US standard atmosphere, as pointed out further down. To clarify the total domain height, we have thus also extended that part in l. 271: "Apart from the cloud field, the 1976 US standard atmosphere (Anderson et al., 1986) interpolated onto the vertical layers given by the cloud data grid serves as background atmosphere. Above the cloud data grid, the native US standard atmosphere levels as they are provided by libRadtran are used, so that the full grid features 264 vertical layers up to a height of 120 km.".

Eqs. (6)-(7): why not look at TOA fluxes as well?

We have changed that as suggested.

l. 546: My first encounter with the notion of thermal "shadows". Is there a reference in the literture?

For now, we have not found a reference to these thermal "shadows" in the literature, as investigations are often solely focused on cloudy regions. In addition to that, these thermal shadows are also very small in magnitude, as one has to keep in mind that we used a logarithmic color scale in order to visualize them, as we explicitly stated in the paper.

l. 575: Clarify "feedback effect". Are the LES dynamics driven by a 3D RT model? Or is this a purely (instantaneous) 3D RT effects? BTW, what radiation scheme was used in the LES runs? Should be specified in Section 3.1 (I'm assuming a standard 1D RT model, but may be wrong).

The term "feedback effects" is now explained in more detail: "The reason for these artifacts are the incomplete solves, which can delay lower-order 3D effects, such as feedback effects from other clouds or the surface. The term "feedback effects" thereby refers to the fact that the 3D radiative effects of a cloud can theoretically alter the conditions determining the 3D radiative effects of any other cloud in the domain. Because these feedback effects require multiple back and forth transports of information, they cannot be fully accounted for when solving radiation incompletely."

We also clarified how the dynamics were driven in the original LES data set by adding the following sentence to the beginning of Section 3.1: "Dynamics in this LES simulation were not driven by radiation, but by a constant net surface flux as described in the namelist input files in Jakub and Gregor (2022)."

---

## Author Response (AR1)

**Response to Review Comment 1 (RC1)**

**Manuscript:** egusphere-2023-2129
**Title:** A dynamic approach to three-dimensional radiative transfer in numerical weather prediction models: the dynamic TenStream solver v1.0
**Authors:** Richard Maier, Fabian Jakub, Claudia Emde, Mihail Manev, Aiko Voigt, and Bernhard Mayer

We thank Anonymous Referee #1 for his or her comments on our manuscript, which we will respond to below. To structure our response, the referee's comments are printed on a gray background color, while our answers are displayed on ordinary white background.

This paper describes a method for 3D radiative transfer that could be computationally affordable enough to be used in high-resolution models. The idea of treating radiation more akin to dynamics is intriguing and as far as I know novel. The results presented are state-of-the-art in terms of speed-accuracy tradeoff (at least for 3D solvers) and potentially very significant for the advancement of NWP models, which are already configured at resolutions where 3D radiative effects are notable yet are currently ignored in all operational models.

My major comments are provided below and relate mainly to the computational aspects, which deserve more attention. Some of my questions may be adequate to address in the review and not in the paper, as it's already long (and concerned mainly with demonstrating the feasibility of the method - which it does excellently!), but a few clarifying sentences and providing absolute runtimes and/or measures of floating point operations in the paper would go a long way in informing the reader how fast dynamic tenStream potentially is, and whether it could be a real contender to operational radiation schemes outside of LES.

Besides this, I think the paper would really benefit if the authors tried to make it more concise by avoiding repetition and removing unnecessary words and sentences. The results shown are relevant but they are sometimes described in a very wordy manner.

Finally, the code does not seem to be actually available to download at current time which I understand is against GMD policy.

**Other major comments:**

1a. In general it's a bit difficult to fully understand the method (although Figure 3 does a good job at illustrating it) especially when it comes its implementation in code and its parallelism. The future tense used in L198-204 implies that the parallelism is not yet implemented. My understanding of dynamic TenStream would be something like this for a simplified 1D case:

```
! Downwelling flux; boundary condition

fd(1) = incsol

fd(2:nlev) = fd_prev_timestep(2:nlev)

! Gauss seidel incomplete solves, not parallelizable

for jiter in 1,niter
```

! Vectorization or other parallelism, array notation

fd(2:nlev) = T(1:nlev-1)*fd(1:nlev-1)

This would correspond to the radiative flows in individual grid boxes being computed concurrently i.e. in parallel within a single step of Fig 3, is this right?

Yes, your simplified 1D case does indeed illustrate the concept of the dynamic TenStream solver, although you use the Jacobi method instead of the Gauß-Seidel method to update the outgoing fluxes of the grid boxes. In contrast to the Gauß-Seidel method, this Jacobi method always uses ingoing fluxes from the previous time step to calculate the updated outgoing fluxes of a grid box. It would thus also allow for concurrent calculation of these outgoing fluxes. On the downside, information can only be propagated to the neighboring grid boxes in every single Jacobi iteration, leading to slow convergence despite high parallelizability.

This is why we have chosen to use the Gauß-Seidel method instead. In your simplified 1D case, this Gauß-Seidel method would look something like this:

for jiter in 1, niter

for iz in 2, nlev

fd(iz) = T(iz-1) * fd(iz-1)

In contrast to the Jacobi method you described, it uses updated ingoing fluxes wherever possible in the calculation of outgoing fluxes, leading to much faster convergence. When calculating fd(iz), for example, we can already use the value of fd(iz-1) determined in the very same iteration jiter. However, this implies that the Gauß-Seidel method does not allow for concurrent calculation of outgoing fluxes for all the grid boxes, as it would simply lead to the Jacobi method in that case: when doing the calculations for all the grid boxes in parallel, we would always have to use ingoing fluxes of the previous iteration instead of the current iteration.

In order to parallelize the Gauß-Seidel method, our idea is thus – as it is described in l. 199 to 202 of the preprint – to apply parallelization to subdomains of the full 3D domain that are larger than an individual grid box. Within every one of these subdomains, the use of the Gauß-Seidel method would ensure that already updated ingoing fluxes are used in the calculation of outgoing fluxes wherever possible, speeding up convergence. Updates between different subdomains would only happen in between different calls of the radiation scheme. This treatment would represent a balance between convergence speed and parallelizability.

But as you correctly noted, parallelization has not yet been implemented into the dynamic TenStream solver by now.

1B. How should the reader interpret the reported speed numbers in terms of effective speed against operational radiation schemes? Is the 1D delta-Eddington reference based on efficient, vectorized code? It is unclear how efficient dynamic TenStream is or could be compared to widely used two-stream codes such as ecRad, which expresses parallelism across g-points, or the RTE+RRTMGP scheme which vectorizes the column dimension instead. Comparison to other schemes could be

greatly facilitated by reporting absolute runtimes, or you could run one of the aforementioned schemes. Potential lack of parallelism and optimization in its current stage can be stressed explicitly and of course, even if dynamic tenStream is currently much slower than operational schemes then it's not a bad result considering full 3D solvers have until now been many orders of magnitudes more expensive. Finally, it could be very useful to report the number of floating point operations (whether absolute or relative to delta-Eddington) but may require a library such as GPTL to estimate, and is perhaps not necessary if the other aspects are clarified.

The relative numbers in Table 1 can indeed not be used to compare the speed of these solvers with respect to those in operational radiation schemes. However, providing such numbers was never the intention of this paper, as the dynamic TenStream solver is still in an early stage of development.

The main point we wanted to make in terms of speed was that a solver using incomplete solves is pretty fast by its design, as it only updates the fluxes in the radiative field a limited amount of times, which is much closer to the way 1D independent column approximations work, where you only update the fluxes of every grid box once every time the radiation model is called.

In order to provide a rough estimate of how fast the solver currently is, we performed this simple speed comparison to other solvers in libRadtran that are indeed not based on highly efficient code and not parallelized either.

We think that simply providing absolute runtimes instead would not really add any value, as these runtimes are highly dependent on the environment the code is executed in: the retrieval of the TenStream coefficients from the corresponding look-up tables is for example highly dependent on where these coefficients are stored. On top of that, the dynamic TenStream solver is not yet parallelized, making comparisons to highly efficiently written and parallelized solvers not particularly useful.

2. Can you discuss whether you see dynamic TenStream to be a potentially viable scheme for global or regional NWP models as they approach kilometer scale resolution? And on cost again: as these models currently use a very coarse radiation time step compared to the ones reported in the paper, such as 15 minutes (AROME 2.5 km regional model) or 1 hour (IFS, but 9 km so not yet km-scale), does this mean that dynamic TenStream would in fact incur a much bigger cost increase for such models than those given in Table 1, or does the coarser spatial resolution compared to LES mean that dynamic TenStreams convergence would still be adequate with relatively coarse radiation time steps?

As the radiative field changes much less rapidly at the lower resolutions used in global or regional scale NWP models, we would assume that also a much coarser radiation time step is needed to achieve comparable results as for the high-resolution test case presented in this paper. On top of that, performing more Gauß-Seidel iterations per radiation call does in fact not scale linearly with computational time, as the computational time is mainly determined by overhead such as retrieving the TenStream coefficients from the look-up tables when performing such a low number of iterations. For our test case, using 10 instead of 2 Gauß-Seidel iterations for example is less than two times more expensive. One could hence easily try to perform a bit more iterations per radiaton call if 2 Gauß-Seidel iterations would not be sufficient to run into proper convergence.

However, this is just speculation at this point in time. To really figure out how incomplete solves perform on the NWP scale, we will have to adapt the model for kilometer-scale resolutions and thoroughly test it, which is beyond the scope of this paper.

To clarify that the paper focuses on subkilometer-scale models for now, we have changed to title to "A dynamic approach to three-dimensional radiative transfer in subkilometer-scale numerical weather prediction models: the dynamic TenStream solver v1.0".

**Minor comments:**

Section 2.1. For the direct radiation, what is the advantage of having 3 streams in the independent x,y,z directions rather than two streams to/from the direction of the sun?

Finite volume algorithms such as the TenStream solver require the calculation of radiative fluxes for at least all the surfaces of the underlying grid boxes – for cuboids, which is the type of grid boxes used in the libRadtran library, that would add up to a total of six streams. Since direct radiation propagates into just one specific direction at every cuboid face, the number of streams can be further reduced to three. That, however, is the minimum amount of streams possible for direct radiation.

L114: Does TenStreams use of an external linear algebra library mean that its implementation is computationally efficient and exploits parallelism but dynamic TenStream currently does not, if so can the speed-up reported in Table 1 be improved further in the future?

Indeed, the use of PETSc allows the original TenStream solver to use computationally efficient methods to solve its system of linear equations. In addition to that, it is also parallelized. However, one of the main aims in the development of the dynamic TenStream solver was to get rid of complex libraries such as PETSc to allow for easier integration into operational models.

Besides that, we do not assume that the numbers in Table 1 will improve when parallelizing the dynamic TenStream solver, as they all refer to the single core performances of the corresponding solvers. Regarding multi-core performance, the TenStream solver is usually much more memory bandwidth limited than 1D delta-Eddington solvers are. As shown in Jakub and Mayer (2016), this leads to the original TenStream solver actually scaling worse to more cores than traditional 1D solvers do.

L114: Does PETSc run on GPUs? Do you think GPU acceleration is promising for (dynamic) tenStream?

Yes, PETSc does indeed run on GPUs. That being said, the Gauß-Seidel method used in the dynamic TenStream solver is a notoriously bad solver on GPU compute architectures. Other solvers such as the Jacobi method are better suited towards the high parallelization on GPUs and have been tested for the original TenStream solver using PETSc on GPUs. However, run times turned out to be only on par or just slightly better than when using CPUs.

In addition to that, the computing time of the dynamic TenStream solver is mainly determined by CPU-based overhead such as the retrieval of the TenStream coefficients and not so much by the

actual Gauß-Seidel solve. Hence, we do not assume a notable increase in speed by just performing the Gauß-Seidel iterations on GPUs.

 Has TenStream been evaluated across a wider range of solar zenith angles and is its performance sensitive to it?

Yes, the TenStream solver has been evaluated at a wide range of solar zenith angles and its performance is sensitive to it (Jakub and Mayer, 2016). Especially when considering small (sub)domains combined with high zenith angles, information has to be transported over multiple subdomains in case of parallelization, slowing down convergence as communication between different cores is required.

 Interesting, what is the reason for tenStream having a worse surface irradiance bias than delta-Eddington?

As Anonymous Reviewer #4 pointed out, the solar zenith angle that we have used in our calculations is very beneficial for the 1D delta-Eddington solver, as there are two different 3D radiative effects at the surface that cancel out for solar zenith angels around 45°. Initially, we have not evaluated the time series for different zenith angles, which is why we did not give an explanation for that in the paper. For the revised version, we have changed that by adding the following part to Sect. 4.3:

"However, it should be noted that the almost non-existent MBE of the delta-Eddington approximation in the solar spectral range is primarily caused by two counteracting 3D radiative effects that happen to cancel each other out at almost exactly the solar zenith angle of 50° that we are using.

[Figure]

Fig. 10: Mean bias error in the net surface irradiance as a function of the solar zenith angle for both the 1D delta-Eddington approximation (blue line) and the original TenStream solver (green line), evaluated at the first time step of the shallow cumulus cloud time series

To show that, Fig. 10 visualizes the MBE for both the delta-Eddington approximation and the original TenStream solver as a function of the solar zenith angle for the first time step of our time series. By looking at the blue line, we can see that the delta-Eddington approximation underestimates the mean net surface irradiance for solar zenith angles below 50°, while it overestimates it for angles above 50°. This is most likely because at low solar zenith angles, 1D solvers typically overestimate cloud shadows due to the lack of transport of diffuse radiation into these shadow regions, leading to an underestimation of the mean net surface irradiance. At high solar zenith angles on the other hand, i.e., when the Sun is close to the horizon, 1D solvers severely underestimate the size of these shadows, as they cast them directly underneath the clouds instead of at a slant angle, leading to an overestimation of the mean net surface irradiance. As we can see in Fig. 10, both of these effects cancel out at an angle of about 50°, which is the one we use, resulting in the almost perfect MBE of the delta-Eddington approximation in the solar spectral range in Fig. 9. Despite this coincidence, however, Fig. 10 also shows us that the original TenStream solver performs slightly worse than the delta-Eddington approximation for any zenith angle below about 50°. However, the difference in the MBEs between the two solvers is quite small, and the magnitude of their respective RMBEs does not get much larger than -1 % for any angle below 50° (not shown here)."

L540-544. This is an example of probably unnecessarily detail and wordiness (4 lines of text to introduce a plot similar to one already shown)

You are absolutely right that Fig. 11 is introduced far too detailed. For the revised version, we significantly shortened this part as follows: "Before making a closing statement, let us also have a look at the results in the thermal spectral range shown in Fig. 16." Note that Fig. 11 is Fig. 16 in the revised version.

**References:**

Jakub, F. and Mayer, B.: 3-D radiative transfer in large-eddy simulations – experiences coupling the TenStream solver to the UCLA-LES, Geosci. Model Dev., 9, 1413–1422, https://doi.org/10.5194/gmd-9-1413-2016, 2016

**Response to Review Comment 3 (RC3)**

**Manuscript:** egusphere-2023-2129
**Title:** A dynamic approach to three-dimensional radiative transfer in numerical weather prediction models: the dynamic TenStream solver v1.0
**Authors:** Richard Maier, Fabian Jakub, Claudia Emde, Mihail Manev, Aiko Voigt, and Bernhard Mayer

We thank Anonymous Referee #2 for his or her comments on our manuscript, which we will respond to below. To structure our response, the referee's comments are printed on a gray background color, while our answers are displayed on ordinary white background.

This is a very interesting paper on speeding up three-dimensional (3D) radiative transfer calculations toward potential use in numerical weather prediction (NWP).

I am very impressed by the paper. It is an important topic, as 3D radiative transfer will require attention as NWP models move to higher resolution.

The methodological advances are carefully designed and effective. I like that the basic ideas are simple and clever and intuitive (e.g., using time-stepping to update the radiative field, and using incomplete solves), while careful attention to details is also crucial to the success of the method (e.g., in the details of the Gauss-Seidel iterations).

The comparisons in the paper are also thorough and include comparisons to a 1D delta-Eddington solver, a 3D Monte Carlo solver, and the original TenStream solver. It is very valuable to have each one of these comparisons, since they span a range of options for speed and accuracy.

The limitations of different methods are also discussed. For instance, the new method is slightly slower than 1D delta-Eddington, and not as accurate 3D Monte Carlo when operated at lower calling frequencies. I appreciate the attention given to these limitations.

It is a very good paper in all aspects: comprehensive, careful, well-written. I appreciated the schematic illustrations, which are helpful for clarifying technical details and main ideas.

I think the paper could be accepted in its current form, but I will mention one specific comment that the authors may wish to address.

**Specific comment:**

The title mentions NWP as the aim. Then the paper presents results for hectometer-scale grid spacings of large-eddy simulations. On the other hand, I would imagine that NWP will be operating at kilometer-scale horizontal grid spacings for quite some time into the future. If that is the case, then will a major modification of your methods be required in order to work effectively with kilometer-scale horizontal grid spacings, where propagation of radiation in horizontal directions is not well-resolved? I would think so.

While the main conceptual ideas of using a time-stepping scheme and incomplete solves will stay the same on kilometer-scale horizontal resolutions, we will certainly have to make some adjustments to the dynamic TenStream solver. Currently, we think that two main modifications will be needed, which we both addressed in the "Summary and Outlook" section of the paper: First, we will have to consider sub-grid scale inhomogeneities such as cloud fraction inside a certain grid

box. Secondly, we will also have to parallelize the model in order to run efficiently on the large domain sizes that come with global or regional-scale NWP models.

If you agree that major modification of your methods will be required in order to work effectively with kilometer-scale horizontal grid spacings, then I would suggest a change to the title (and also possibly some changes in the Introduction section and Summary and outlook section). For instance, in the title, possibly change 'A dynamic approach to' to 'A dynamic approach toward', or change 'in NWP' to 'in LES' or 'in hectometer-scale NWP'. Then you could save the NWP emphasis for a later paper when you can address the difficulties that will arise in using dynamic TenStream on actual NWP models with kilometer-scale grid spacing.

Thank you for this suggestion. To clarify that the solver is currently only designed for the use on subkilometer-scale horizontal resolutions, we have changed the title to "A dynamic approach to three-dimensional radiative transfer in subkilometer-scale numerical weather prediction models: the dynamic TenStream solver v1.0".

The revised version also includes minor adjustments in the "Introduction" and "Summary and Outlook" sections of the paper to stress that this first version of the dynamic TenStream solver is only designed for the use on subkilometer-scale horizontal resolutions:

In the introduction, we have modified the penultimate paragraph as follows: "[…] To address this high computational cost of current 3D solvers, this paper presents a first step towards a new, "dynamic" 3D radiative transfer model. Currently designed for the use at subkilometer-scale horizontal resolutions, where model grid boxes can be assumed to be homogeneous, this new, fully three-dimensional model is based on the TenStream solver. […]".

A similar modification was applied in the "Summary and Outlook" section of the paper: "Based upon the TenStream solver, we presented a new radiative transfer model currently designed for the use at subkilometer-scale horizontal resolutions that allows us to calculate 3D radiative fluxes and heating rates at a significantly increased speed by utilizing two main concepts that both rely on the idea that the radiative field does not completely change in between two calls of the scheme: [...]"

This was the only issue I want to mention, and I otherwise was pleased and impressed by the careful comparisons and discussions of limitations.

**Technical correction:**

Line 505: "In here" should be just "Here"

We changed that as suggested.

**Response to Review Comment 4 (RC4)**

**Manuscript:** egusphere-2023-2129
**Title:**   A dynamic approach to three-dimensional radiative transfer in numerical weather prediction models: the dynamic TenStream solver v1.0
**Authors:**  Richard Maier, Fabian Jakub, Claudia Emde, Mihail Manev, Aiko Voigt, and Bernhard Mayer

We thank Anonymous Referee #3 for his or her comments on our manuscript, which we will respond to below. To structure our response, the referee's comments are printed on a gray background color, while our answers are displayed on ordinary white background.

* General comments:

This is a welcome update of the TenStream 3D radiative transfer (RT) code that already fills a major gap in LES modeling capability, namely, to perform 3D RT broadband radiation budget estimation for Large-Eddy Simulation (LES) models. LES is now routinely used in cloud-scale process modeling to address some of predictive climate science's most stubborn issues, such as cloud feedbacks and aerosol-cloud interactions.

However, in spite of generating fully 3D (i.e., vertically-developed) clouds driven by convective dynamics, the RT parameterizations used in LES are still too often heritage codes from Global Climate Models (GCMs) where nothing less than ~50 to 100 km in scale is resolved, hence all clouds and many cloud systems. A typical aspect ratio for a GCM cloudy column is therefore on the order of 1-to-10, thus, some form of 1D RT that captures the internal variability of the clouds (e.g., McICA) is justified since little radiation will be leaked through the horizontal boundaries anyway. In sharp contrast, a cloudy column in an LES has the opposite aspect ratio: say, 5 km by 50 m, hence about 100-to-1. Even cloud-resolving models (CRMs), say, at 5 km by 0.5 km are 10-to-1. NWP models are heading into that kind of spatial resolution as well. So there is plenty of opportunity for net horizontal fluxes to develop across grid-cell facets, starting with direct shadowing of neighboring cells in the anti-solar direction. The TenStream model is purposefully designed to account for this 3D RT in terms of radiation energetics, hence fluxes, not radiances, as required for computing heating rates profiles and net fluxes through the top and bottom boundaries.

The new _dynamic_ TenStream model is designed to address the issue of computational efficiency that is in the way of the general acceptance of TenStream in the LES community for operational implementation. Specifically, it brings CPU time allocation down to ~3x the baseline cost of 1D RT, and does so by cutting a few corners, which could carry a cost in accuracy. Therefore, dynamic TenStream is benchmarked for accuracy against the original TenStream, as well as 1D RT (delta-Eddington) and full 3D RT (MYSTIC). Its accuracy is at par with the original TenStream, which is already a vast improvement in accuracy for radiation budget estimation using standard 1D RT.

The paper is well written and illustrated. It should be published by GMD after a minor revision that addresses the following issues.

* Specific comments:

(1) Careful attention is paid to the heating-rate profile and surface irradiance/flux. However, it seems to me that the outgoing TOA flux is also important. Maybe TenStream enforces radiant

energy conservation is such a way that the TOA flux is as accurate as the rest, but that isn't obvious to this reviewer. At a minimum, some kind of statement on TOA flux accuracy is in order.

You are right that we focused our evaluation on heating rates and net surface irradiances, as they are the main drivers of the weather. For the revised version, we extended Sect. 4.3 to also account for the performance of the new solver in determining net irradiances at top of atmosphere (TOA):

"Finally coming to the upper boundary of our domain, Fig. 11 shows the temporal evolution of the MAE in the net irradiance at top of atmosphere (TOA) in an otherwise similar fashion as Fig. 8. Again, the incomplete solves in the dynamic TenStream solver lead to a slight divergence of the MAE of this solver (red lines) compared to the original TenStream solver (green lines) in both spectral ranges. However, this divergence remains small compared to the difference between the 3D TenStream solver and the 1D δ-Eddington approximation, even at the lowest investigated calling frequency of 60 s. This indicates that also at TOA, the dynamic TenStream solver is much better in capturing the spatial structure of the net irradiances than the traditional δ-Eddington solver is."

[Figure]

**Figure 11.** Temporal evolution of the mean absolute error in the net irradiance at top of atmosphere for the different solvers with respect to the MYSTIC benchmark run at calling frequencies of 10 s, 30 s and 60 s for both the solar (panel a) and thermal (panel b) spectral range. The MAE of the MYSTIC benchmark run itself is visualized by the dotted black line.

Similar to the surface, however, this does not fully apply in terms of domain averages. The corresponding temporal evolution of the MBE is shown in Fig. 12. Starting with the thermal spectral range displayed in panel (b), our new solver again just shows a comparatively small divergence from the original TenStream solver with time and performs significantly better than the δ-Eddington approximation throughout the entire time series, regardless of the calling frequency used. In the solar spectral range, however, the original TenStream solver already performs a bit worse than the δ-Eddington approximation does with time-average MBEs of about −4 W m$^{-2}$ for the TenStream solver compared to −3 W m$^{-2}$ for the 1D solver. More noticeably though, the incomplete

solves in the dynamic TenStream solver lead to a fairly pronounced divergence in terms of the MBE from the original TenStream solver when compared to the difference between the 1D and original TenStream solvers. However, for every calling frequency investigated, this divergent behavior peaks at values that translate to RMBEs no larger than 1.25 % (not shown here). Taking both domain boundaries into account, we can thus draw similar conclusions as for the heating rates:

1. On the grid box level, our new solver determines far better net irradiances at both the surface and TOA than current 1D solvers do, even when operated at much lower calling frequencies.

2. Looking at domain averages, however, the incomplete solves within the dynamic TenStream solver lead to the build-up of a bias with time. In terms of magnitude relative to the original TenStream solver, this bias becomes larger the lower the calling frequency is and exceeds the bias of current 1D solvers, especially in the solar spectral range."

[Figure]

**Figure 12.** Temporal evolution of the mean bias error in the net irradiance at top of atmosphere for the different solvers with respect to the MYSTIC benchmark run at calling frequencies of 10 s, 30 s and 60 s for both the solar (panel a) and thermal (panel b) spectral range. A run with no bias is indicated by the dotted black line."

Apart from this large addition, net irradiance at TOA is now mentioned in various parts of the paper.

(2) The temporal down-sampling and the incomplete solves naturally cause the new model to drift away from the original counterpart. Would it not be beneficial to occasionally "reset" this drift to zero by calling the original TenStream? Of course there is a whole study to perform about when to do this operationally, without the benchmark information at hand.

This is a good idea, and one that we definitely had in mind when thinking about future couplings of our new solver to LES or NWP models. The implications of such resets would be relatively straightforward, as the error metrics would simply reduce to those of the original TenStream solver

whenever such a reset was performed. For this paper, however, we wanted to focus on how our new solver performs when applied with the lowest computational cost possible – that is, with a low number of two Gauß-Seidel iterations per call and no intermediate resets of the new model.

In the future, however, it would certainly be interesting to investigate the trade-off between increased accuracy due to occasional model resets on one side, and the additional computational cost that these resets introduce on the other side. We have included this thought in the outlook of the revised version of the paper: "In this context, it would also be interesting to investigate whether occasional full solves are a computationally feasible means of ensuring that the results of the dynamic TenStream solver do not deviate too much from those of the original TenStream solver."

(3) Although it should have been done when documenting the original TenStream model, it would be good to look into the past to find models with similar mathematical structure in terms of radical angular simplification compared to standard 3D RT solvers, more precisely with improved efficiency in mind. Can I suggest a few?

- an original "6-flux" model, applied to homogeneous plane-parallel media (but with potential for heterogeneous media):

Chu, C.M. and Churchill, S.W., 1955. Numerical solution of problems in multiple scattering of electromagnetic radiation. The Journal of Physical Chemistry, 59(9), pp.855-863.

- a discrete-angle RT formalism predicated on regular tessellations of 2D and 3D spaces, seeking the minimal number of directions to capture 3D RT effects:

Lovejoy, S., Davis, A., Gabriel, P., Schertzer, D. and Austin, G.L., 1990. Discrete angle radiative transfer: 1. Scaling and similarity, universality and diffusion. Journal of Geophysical Research: Atmospheres, 95(D8), pp.11699-11715.

- a 2D (4-stream) RT model in a deterministic fractal medium, emphasizing numerical implementation (successive over-relaxation scheme):

Davis, A., Gabriel, P., Lovejoy, S., Schertzer, D. and Austin, G.L., 1990. Discrete angle radiative transfer: 3. Numerical results and meteorological applications. Journal of Geophysical Research: Atmospheres, 95(D8), pp.11729-11742.

- the same 2D (4-stream) RT model but in a random multifractal medium, emphasizing numerical implementation (Monte Carlo scheme):

Davis, A.B., Lovejoy, S. and Schertzer, D., 1991, November. Discrete-angle radiative transfer in a multifractal medium. In Wave Propagation and Scattering in Varied Media II (Vol. 1558, pp. 37-59). SPIE.

- vastly faster solution of the 4-stream model using sparse matrix inversion:

Lovejoy, S., Watson, B.P., Grosdidier, Y. and Schertzer, D., 2009. Scattering in thick multifractal clouds, Part II: Multiple scattering. Physica A: Statistical Mechanics and its Applications, 388(18), pp.3711-3727.

Thank you for these suggestions. We also think that these papers should have been primarily mentioned in the documentation of the original TenStream solver. Nonetheless, we included some of them into the introduction of the revised version of our paper:

"[...] On the other hand, a lot of work went into the speed-up of inter-column radiative transport at subkilometer-scale resolutions, where model grid boxes can be gradually treated homogeneously. A large group of these models simplifies the expensive angular part of 3D radiative transfer calculations by just using a discrete number of angles (e.g., Lovejoy et al., 1990; Gabriel et al., 1990; Davis et al., 1990). Most recently, the TenStream solver (Jakub and Mayer, 2015) built upon this idea. It is capable of calculating 3D radiative fluxes and heating rates in both the solar and the thermal spectral range. To do so, it extends the 1D two-stream formulation to ten streams to consider horizontal transport of energy. [...]"

\* Technical corrections:

Title: The application to NWP models is both inspirational and aspirational. Here, however, the authors only get as far as LES, or CRM (100 m grid spacing). A more accurate title is in order.

You are certainly right with that. We changed the title to "A dynamic approach to three-dimensional radiative transfer in subkilometer-scale numerical weather prediction models: the dynamic TenStream solver v1.0" for the revised version.

l. 99: i.e., e.g., (need commas, I think)

We have changed the sentence containing this expression to clarify the next comment and added commas behind "i.e." and "e.g." elsewhere in the document.

l. 100: "n1" --> what is the "1" for?

We have clarified the meaning of the "1" by adding more explanation to the corresponding example: "For example, the emissivity $e_{0,i,j,k}$ of grid box $(i,j,k)$ in upward direction is equal to the fraction of the downward facing radiative flux $\Phi_{1,i,j,k+1}$ that is absorbed on the way through that grid box, which in turn is one minus the sum of all fractions $a_{n1,i,j,k}$ of $\Phi_{1,i,j,k+1}$ exiting grid box $(i,j,k)$, i.e., $e_{0,i,j,k} = 1 - \Sigma_{n=0}^{9} a_{n1,i,j,k}$, where $a_{n1,i,j,k}$ refers to the corresponding entries in the second column of matrix $T_{i,j,k}$."

l. 104: first "out" --> not italics

We have changed that as suggested.

Fig. 3: For SZA near 45 deg, one could use a diagonal sweep through the grid? Same for ~45 deg in azimuth? Admittedly more tricky to code, but it would follow more closely the propagation of direct sunlight. No? [...]

```
 _____ _____ _____ _____ ___  etc.

|     |     |     |     |     |

|  1  |  2  |  6  |  7  |  15

|     |     |     |     |     |
```

```
 ____ ____ ____ ____ ___
|    |    |    |    |    |
| 3  | 5  | 8  | 14 |
|    |    |    |    |    |

 ____ ____ ____ ____ ___
|    |    |    |    |    |
| 4  | 9  | 13 |    |
|    |    |    |    |    |
etc.
```

Thank you for this suggestion. Indeed, one could think about more sophisticated patterns of propagating through the model grid boxes in order to follow the propagation of direct radiation even more closely. However, it would likely not improve convergence, since direct radiation is only represented by three independent streams pointing in x, y and z direction in the dynamic TenStream solver. By properly sorting the resulting three loops due to solar incidence angle, one already ensures that the ingoing direct fluxes of any grid box are always updated before calculating the corresponding outgoing fluxes – even at 45° zenith and azimuth angle. To illustrate that, let us look at a simplified version of Fig. 3 showing only direct streams:

[Figure]

Similar to Fig. 3, this sketch shows the first four steps of one Gauß-Seidel iteration in two dimensions only. In every step, ingoing fluxes are used to update the outgoing fluxes of the corresponding grid box (highlighted in grey). Grey arrows in contrast to black arrows indicate fluxes that have not yet been updated in this Gauß-Seidel iteration. We consider a solar zenith angle of 45° with the Sun shining from the top-right. We can clearly see that even with our not as sophisticated way of iterating through the domain, we always use already updated ingoing fluxes to update the corresponding outgoing fluxes – except for fluxes at the borders of the domain, that are subject to boundary conditions. Due to the definition of the direct streams in the solver, a diagonal sweep through the grid boxes would actually even slow down convergence, as we would not always use already updated ingoing fluxes following such a pattern, although these diagonal sweeps seem to follow the propagation of solar incidence more closely at first.

l. 190: "this direction" -->? horizontal scan

We actually reverse the iteration direction in all three dimensions in every following Gauß-Seidel iteration. To clarify that, we have adjusted the corresponding sentence: "Thus, every time we finish iterating through all the grid boxes, which completes a Gauß-Seidel iteration step, we reverse the direction of iteration in all three dimensions to not favor propagation of information in one direction."

S. 3.1 (beginning): specify domain size in cells _and_ km

We have changed that as suggested: "Originally, the data set features both a high temporal resolution of 10 s and 256 × 256 grid boxes with a high spatial resolution of 25 m in the horizontal."

l. 262: specify domain height (in km too)

As we pointed out in the paper, the total domain is constructed using two different sources: For the first 220 layers, we use the high vertical resolution of 25 m provided by the LES runs. We clarified the domain height of that part referred to in l. 262 of the preprint: "In the vertical, the modified cloud data set consists of 220 layers with a constant height of 25 m, thus reaching up to a height of 5.5 km."

Above this vertically highly resolved grid, we use atmospheric levels provided by the 1976 US standard atmosphere, as pointed out further down. To clarify the total domain height, we have thus also extended that part in l. 271 of the preprint: "Apart from the cloud field, the 1976 US standard atmosphere (Anderson et al., 1986) interpolated onto the vertical layers given by the cloud data grid serves as background atmosphere. Above the cloud data grid, the native US standard atmosphere levels as they are provided by libRadtran are used, so that the full grid features 264 vertical layers up to a height of 120 km.".

Eqs. (6)-(7): why not look at TOA fluxes as well?

We have changed that as suggested.

l. 546: My first encounter with the notion of thermal "shadows". Is there a reference in the literture?

For now, we have not found a reference to these thermal "shadows" in the literature, as investigations are often solely focused on cloudy regions. In addition to that, these thermal shadows are also very small in magnitude, as one has to keep in mind that we used a logarithmic color scale in order to visualize them, as we explicitly stated in the paper.

l. 575: Clarify "feedback effect". Are the LES dynamics driven by a 3D RT model? Or is this a purely (instantaneous) 3D RT effects? BTW, what radiation scheme was used in the LES runs? Should be specified in Section 3.1 (I'm assuming a standard 1D RT model, but may be wrong).

The term "feedback effects" is now explained in more detail: "The reason for these artifacts are the incomplete solves, which can delay lower-order 3D effects, such as feedback effects from other clouds or the surface. The term "feedback effects" thereby refers to the fact that the 3D radiative effects of a cloud can theoretically alter the conditions determining the 3D radiative effects of any other cloud in the domain. Because these feedback effects require multiple back and forth transports of information, they cannot be fully accounted for when solving radiation incompletely."

We also clarified how the dynamics were driven in the original LES data set by adding the following sentence to the beginning of Section 3.1: "Dynamics in this LES simulation were not driven by radiation, but by a constant net surface flux as described in the namelist input files."

**Response to Review Comment 5 (RC5)**

**Manuscript:** egusphere-2023-2129
**Title:** A dynamic approach to three-dimensional radiative transfer in numerical weather prediction models: the dynamic TenStream solver v1.0
**Authors:** Richard Maier, Fabian Jakub, Claudia Emde, Mihail Manev, Aiko Voigt, and Bernhard Mayer

We thank Anonymous Referee #4 for his or her comments on our manuscript, which we will respond to below. To structure our response, the referee's comments are printed on a gray background color, while our answers are displayed on ordinary white background.

**Summary**

This paper describes an updated version of the TenStream solver, which can be used to solve radiation in high-resolution numerical models such as atmospheric Large-Eddy Models. This new "dynamic" version represents an improvement in terms of computational speed compared to the original TenStream. It relies on the same radiative transfer model but its resolution is accelerated using two fundamental ideas. This first one is that previously computed radiation fields can be used as a first guess in the numerical resolution of the linear system corresponding to the TenStream model, which is refered to as a "dynamic" approach or "time-stepping" scheme because of the similarity with the resolution of advection in the dynamical core of atmospheric models. The second idea is that using an iterative method, namely, the Gauss-Seidel method, to solve the linear system starting from this first guess offers the possibility to stop the calculation after a few iterations, using the resulting field even if it has not converged toward the solution. This is refered to as "incomplete solve". After exposing these ideas and describing their implementation in the dynamical TenStream solver, the authors examine the errors introduced by the fact that infrequent calls to radiation will lead to starting from a "bad" first guess, increasing the error associated with incomplete solves compared to more frequent calls, for the same number of Gauss-Seidel iterations; as well as errors introduced by the fact that the solves are incomplete, by comparing their results with those predicted by the full TenStream solver given the same input fields. Their conclusions are that the dynamic TenStream is significantly faster than the original TenStream, while being mostly as accurate even using as few as 2 Gauss-Seidel iterations at each radiation call.

**General comments**

I find the paper of great interest. It reports important advances in the field of 3D radiation modeling and its numerical resolution, working towards replacing overly simplified and strongly biased 1D radiation models by their 3D counterparts. I find the manuscript very clear and well organized, and the demonstration of the capabilities of the dynamical TenStream solver convincing. I appreciated the detailed explanations on the models and evaluation methods. I found the part where the results are discussed a little less satisfying but I understand that much more work might be needed to really understand the biases of the different models and that it probably falls out of scope of the present study.

In the following I list some questions and suggestions that I think would make the manuscript even clearer. They are given in a chronological manner rather than per importance. I trust the authors' judgement in the relevance of my suggestions and questions and would recommend publication even if not all my comments are addressed in the revised version.

**Specific comments**

- Mostly in the Abstract and Introduction but also elsewhere in the paper: the distinction between sub-grid and inter-column "3D effects" is not clear enough and I am afraid it might be confusing for a non-expert reader. For instance in the Introduction L.30-33, it is mentioned that NWP models still use 1D ICA RT schemes, by which I think you mean "solve radiation independently in each model column". Immediatly after this statement comes "such as the McICA" which is indeed a 1D RT solver but here the ICA refers to the neglect of *subgrid* 3D effects (that is, between stochastically generated 1D profiles or "subcolumns"). Later on, you describe SPARTACUS, which is of a very different nature from the TenStream and NCA models, and only there the distinction between inter-column and sub-grid 3D effects is mentioned. I suggest you clarify since the beginning of the Introduction that this distinction exists and that your work relies to the resolution of inter-column horizontal transport. I also feel this distinction is lacking when you write that the 3D effects are becoming more important as the horizontal resolution of NWP models increases. I would rather say that the partition between subgrid and inter-column 3D effects depends on the host model horizontal resolution and that, as we go toward higher resolution, it becomes more important to solve horizontal transfers between columns and less so at the subgrid scale.

Thank you for pointing this out. Subgrid and inter-column 3D effects were indeed not clearly separated in our paper. To account for this differentiation, we have modified the introduction as follows:

"Depending on scale, we can differentiate between two different regimes of 3D radiative transport: On the model grid scale, 3D radiative transfer allows for horizontal transport of energy between adjacent model columns, whereas on the subgrid scale, it refers to the three-dimensional transport of radiative energy within a heterogeneous model grid box. The calculation of both of these effects is computationally expensive, largely preventing their representation in operational weather forecasting. This is why up to this date, numerical weather prediction (NWP) models still use one-dimensional (1D) independent column approximations (ICA), such as the Monte Carlo Independent Column Approximation (McICA; Pincus et al. (2003)) currently employed at both DWD and ECMWF (DWD, 2021; Hogan and Bozzo, 2018). These models assume that radiative transport between grid boxes only takes place in the vertical and neglect any horizontal transport of energy – both in between different model columns and within individual model grid boxes.

However, both of these effects have been shown to be important for the correct calculation of radiative transfer in the atmosphere. While subgrid-scale 3D effects primarily act at coarser resolutions, where an individual grid box incorporates both cloudy and clear-sky regions and should thus not be treated homogeneously, the increasing horizontal resolution of numerical weather prediction models makes inter-column radiative transfer more and more important (O'Hirok and Gautier, 2005). [...]"

We also clarified that our new solver is specifically designed for considering inter-column 3D radiative effects on the subkilometer-scale:

"To address this high computational cost of current 3D solvers, this paper presents a first step towards a new, "dynamic" 3D radiative transfer model. Currently designed for the use at subkilometer-scale horizontal resolutions, where model grid boxes can be assumed to be homogeneous, this new, fully three-dimensional model is based on the TenStream solver. It

accelerates inter-column 3D radiative transfer towards the speed of currently employed 1D solvers by utilizing two main concepts."

Furthermore, we have clarified which 3D radiative effect we refer to in various parts of the paper.

- One condition for the Dynamic TenStream to work is that the radiation field does not change too much between two radiation calls, so that the field used as first guess is already close enough to the solution that only a few iterations of the Gauss-Seidel algorithm are needed. It made me wonder if the radiation field was advected with the rest of the atmospheric fields so that it still matched an advected cloud field and the largest errors were mostly limited to cloud birth and death between two radiation time steps?

We have not investigated that, but we would assume that the general structure of the radiative field is indeed to a large part advected with the rest of the atmospheric fields, if the time step does not get too large in a sense that cloud birth and death, but also major changes in the structure of the clouds dominate the differences in the radiative field in between two radiation time steps.

But that is actually a very interesting aspect to investigate in the future, as one could choose a more intelligent first guess that already considers advection as a starting point of the incomplete solves. This might speed up convergence even more.

We have added this into the outlook of the revised version of the paper: "Additionally, we could think about an even more sophisticated first guess for the incomplete solves by advecting the radiative field with the rest of the atmospheric fields. As we assume that the radiative field does not totally change in between two different calls of the radiation model, such a first guess should already better account for the updated position of the clouds, so that the incomplete solves could primarily focus on correcting for the changed optical properties of the clouds, which could speed up convergence even more."

- How are the thermal sources handled by the Gauss-Seidel method? I imagine they are calculated at the beginning of the iterations and somehow part of the first guess but could you explain how it works exactly? Maybe comment on the fact that B is absent from eq. (2)?

The thermal source terms are calculated right before starting with the Gauß-Seidel iterations. They are not part of the first guess, but calculated from scratch within the same routine that retrieves the TenStream coefficients from the corresponding look-up tables. Whenever this routine is called for a grid box, it calculates both the Planck emission and emissivities for every stream, the latter following the pattern lined out in l. 100 of the preprint. We added a sentence at the end of section 2.2.1 to clarify that: "The thermal source terms are not part of the first guess and have to be calculated from scratch following the pattern outlined in Sect. 2.1 before starting with the Gauß-Seidel algorithm."

In Eq. (2), the thermal source term was indeed missing. We fixed that for the revised version. Thanks for pointing this out!

- In Fig. 3, I don't understand how the fluxes entering the domain at the borders would systematically be "updated right from the beginning"? From what I understand, if the BC are periodic for instance, then the incoming flux at the left-side wall would be updated only after the outgoing fluxes at the right-side wall have been calculated? In a parallelized Dynamic TenStream, the fluxes at the subdomain boundaries would only be updated at the end of the calculation as mentioned at L.201 and hence the incoming fluxes at the borders used at a given time would be the ones from the calculations at the previous radiation call?

You are perfectly right, the boundary conditions were not properly visualized in Fig. 3. Hence, we have updated Fig. 3 and its caption as follows:

[Figure]

Figure 3: Two-dimensional schematic illustration of the first four steps of a Gauß-Seidel iteration, showing both diffuse and direct TenStream fluxes in case of Sun shining from the west or left-hand side. As one sequentially iterates through the grid boxes, ingoing fluxes are used to update the outgoing fluxes of the corresponding grid box (highlighted in grey). Grey arrows in contrast to black arrows indicate fluxes that have not yet been updated in this Gauß-Seidel iteration. Ingoing fluxes at the domain borders are dependent on the type of boundary conditions used. For this schematic, we applied periodic boundary conditions in the horizontal direction, while fluxes entering at the top of the domain are updated right from the beginning.

- L.154-156, solving for a clear-sky situation does not automatically imply that there is no horizontal variability in the model, e.g. specific humidity or surface albedo could still vary on the horizontal. In which case, shouldn't the spin-up be performed on the entire model grid? Would that still be manageable? Wouldn't it be cheaper to use the classical TenStream solver for initiating the Dynamic TenStream? At L.288, it is said that the classical TenStream is not used for initialization to avoid relying on PETSc library, could you elaborate a little more on that, and maybe mention it when the spin-up is first discussed in Sec. 2.2.2?

You are right, normally there can still be some horizontal variability in the background atmosphere in the absence of clouds. However, this background atmosphere is always one-dimensional in the libRadtran library, which allows us to perform the clear-sky spin-up for a single vertical column. We clarified that directly in Sect. 2.2.2 for the revised version of the paper: "Since there is no horizontal variability in the cloud field in a clear-sky situation and our model does not feature any horizontal variability in the background atmosphere, we can perform this calculation for a single vertical column at a dramatically increased speed compared to a calculation involving the entire model grid."

In case the background atmosphere is not horizontally homogeneous, this 1D spin-up would certainly be less accurate, but still resemble a better starting point of the Gauß-Seidel algorithm than starting with values of zero for all the radiative fluxes. We also added that to the revised version: "Assigned to the radiative fluxes of all vertical columns in the entire domain, these values then

provide a first guess for all the TenStream variables that can be assumed to be much closer to the final result than starting with values of zero – even if the background atmosphere was not horizontally homogeneous and we would have to take the average of that background first."

For a better spin-up, one could in general of course also use a full TenStream solve. You are absolutely right that the reason for not using it should be given directly in Sec. 2.2.2, which we have done for the revised version, alongside with adding more background to that decision: "However, for the very first call of the radiation scheme, we cannot use a previously calculated result. In order to choose a reasonable starting point of the algorithm for this first call as well, though, we could use a full TenStream solve. However, such a solve would be computationally expensive and rely on numerical methods provided by the PETSc library, that we want to get rid of with our new solver to allow for easier integration into operational models. So instead of performing a full TenStream calculation, we decided to solve the TenStream linear equation system for a clear sky situation as a starting point."

- L.254, "our solver does not yet take sub-grid scale cloud variability into account": any idea how you would do that? This is probably of great importance for NWP and without it the TenStream solver(s) might be restricted to LES where grid boxes might be considered homogeneous?

You are right, accounting for sub-grid scale cloud variability will possibly be the most important thing to consider when going to the NWP scale. To give a first idea of how we could do that, we extended the corresponding sentence in the outlook as follows: "Finally, going to the NWP scale, we will certainly need to consider sub-grid scale cloud variability, for example by extending the TenStream look-up tables to account for cloud fraction." The implementation of these ideas however is beyond the scope of this paper.

- L.257 "to avoid problems with artificially low LWC at cloud edges [...]" were you able to quantify the error in the radiative field induced by smoothing the cloud field vs. by subsampling it at a coarser resolution? Or could you cite a study demonstrating that one is better than the other?

No, we did actually not quantify this error. The motivation to just use every fourth grid box instead of averaging the cloud fields was exactly the one given in the paper: We thought that it might be more wisely to just use data coming directly out of the LES runs instead of producing averages, where artificially low liquid water contents could lead to an underestimation of 3D radiative effects at cloud edges.

- L.426-428, I disagree with "the newly developed solver is able to almost perfectly reproduce the results of the original TenStream solver whenever called". Looking at Fig. 6b, after a few time steps it seems that the Dynamic TenStream for dtrad=30s line is always above the TenStream lines. Similarly, I disagree with "our new solver even performs better than the delta-Eddington solver at a calling frequency of 10 s when it is operated at a calling frequency of 30 s" at L.430-431. Looking at Fig.6b again, it seems that the errors associated with the Dynamic TenStream for dtrad=30s become larger than those associated with the delta-Eddington for dtrad=10s after around 8200 s.

You are right that the dynamic TenStream solver is not exactly reproducing the results of the original TenStream solver whenever called. We actually explicitly noted that in l. 424-426 of the preprint: "Looking closely, we can also see that for both lower calling frequencies, the MAE of the dynamic TenStream solver does not always match the errors obtained at a calling frequency of 10 s when updated.". However, the phrase "almost perfectly" is certainly not appropriate. We thus changed the statement to "the newly developed solver is almost able to reproduce the results of the original TenStream solver whenever called" for the revised version.

Apart from that, you are right that the maximum error caused by the dynamic TenStream solver at a calling frequency of 30 s exceeds the error of the delta-Eddington solver at a calling frequency of 10 s in the thermal spectral range (as does the original TenStream solver, by the way). To correct that, we have changed the meaning of the sentence to account for time-averages: "Looking at Fig. 6, we can now see that on time-average, dynamic TenStream even performs better than the $\delta$-Eddington approximation at a calling frequency of 10 s (bold blue line) when it is operated at a calling frequency of 30 s (bold red dash-dotted line) and thus with a similar computational demand as the 1D solver – both in the solar, as well as in the thermal spectral range."

- Looking at Fig. 7b, it is interesting that the dynamic TenStream solver bias in the thermal partially compensates the original TenStream bias and it might not be for good reasons e.g. the original TenStream is not diffusive enough in the thermal and the incomplete solving in the dynamic approach adds numerical diffusion making the solution closer to the reference but for unphysical reasons?

Thank you for this suggestion that is definitely worth looking into. However, tests conducted with the original TenStream solver involving 24 instead of 10 diffuse streams to account for more diffusion did in general not reduce its bias. However, a sophisticated answer to this question would require a much deeper analysis of the two solvers that is beyond the scope of this paper.

- In Fig. 9a, it is also interesting that the mean bias is larger in the TenStream solvers than in the delta-Eddington. I think this might be very dependent on the solar zenith angle: 3D effects on the mean surface fluxes go from positive to negative as the sun goes from zenith to horizon and are usually close to zero for angles between 40 and 50 degrees from zenith in cumulus cloud fields (depending on cloud and surface properties). This is because the overestimation by 1D models of direct flux reaching the surface compensates the underestimation of diffuse almost perfectly at these angles. This solar angle dependence would not explain Fig. 9b though, but here the TenStream and delta-Eddington errors are of the same magnitude albeit of opposite sign.

Thanks for pointing this out. We actually had the same idea that the 3D effects in the domain-average net surface flux probably cancel at the zenith angle of 50° we are investigating. Initially, we have not evaluated the time series for different zenith angles, which is why we did not give an explanation for that in the paper. For the revised version, we investigated that in more detail in Sect. 4.3:

"However, it should be noted that the almost non-existent MBE of the $\delta$-Eddington approximation in the solar spectral range is primarily caused by two counteracting 3D radiative effects that happen to cancel each other out at almost exactly the solar zenith angle of 50° that we are using.

To show that, Fig. 10 visualizes the MBE for both the δ-Eddington approximation and the original TenStream solver as a function of the solar zenith angle for the first time step of our time series. By looking at the blue line, we can see that the δ-Eddington approximation underestimates the mean net surface irradiance for solar zenith angles below 50°, while it overestimates it for angles above 50°. This is most likely because at low solar zenith angles, 1D solvers typically overestimate cloud shadows due to the lack of transport of diffuse radiation into these shadow regions, leading to an underestimation of the mean net surface irradiance. At high solar zenith angles on the other hand, i.e., when the Sun is close to the horizon, 1D solvers severely underestimate the size of these shadows, as they cast them directly underneath the clouds instead of at a slant angle, leading to an overestimation of the mean net surface irradiance. As we can see in Fig. 10, both of these effects cancel out at an angle of about 50°, which is the one we use, resulting in the almost perfect MBE of the δ-Eddington approximation in the solar spectral range in Fig. 9. Despite this coincidence, however, Fig. 10 also shows us that the original TenStream solver performs slightly worse than the δ-Eddington approximation for any zenith angle below about 50°. However, the difference in the MBEs between the two solvers is quite small, and the magnitude of their respective RMBEs does not get much larger than −1 % for any angle below 50° (not shown here).

[Figure]

**Figure 10.** Mean bias error in the net surface irradiance as a function of the solar zenith angle for both the 1D δ-Eddington approximation (blue line) and the original TenStream solver (green line), evaluated at the first time step of the shallow cumulus cloud time series."

- Even if the TenStream solvers clearly perform radically better than delta-Eddington, it is difficult to imagine how the remaining errors with respect to MYSTIC might affect the simulation once it is used online. Do you have any insights on that, from the literature maybe? For instance it is not obvious to me if it would be preferable to have the right mean flux but with the wrong spatial structure, or the opposite?

Currently, we do not really have any insights on how the errors introduced by both TenStream and the incomplete solves would affect simulations driven by our new solver. And although this topic is highly interesting for the future, it is somehow beyond the scope of this paper that was mainly

focused on exploring first steps on whether incomplete solves could be an option to consider inter-column 3D radiative effects at much lower computational cost.

As it is a very important topic, though, we have included it into the outlook of the paper: "Coupled to dynamics, it will also be very interesting to investigate how the incomplete solves in the dynamic TenStream solver influence the development of clouds."

- I find it a little frustrating that all simulations have been performed with two Gauss-Seidel iterations. No information on convergence speed is provided in the paper whereas from what I understand of the method there is a tradeoff to be found between frequency of radiation call and number of iterations of the Gauss-Seidel method?

The preprint version of the paper was indeed just presenting results for a very low number of two Gauß-Seidel iterations per call. We limited the results to this setup, as it serves as a kind of worst-case setup for the new solver and already lead to promising results. You are however right that the implications of using more iterations are also very interesting and important. For the revised version, we have thus included a new section exploring the effects 
[revised manuscript text omitted]

- L.559 I disagree with "almost perfectly". This formulation is not great anyway, as something that is not "entirely" perfect is by definition imperfect.

You are right that this formulation is not making much sense. We got rid of the word "perfectly" for the revised version: "Comparing these results to those of our newly developed dynamic TenStream solver, we can see that also in the thermal spectral range, it is almost able to reproduce the results of the original TenStream solver, even when operated at lower calling frequencies.".

- L.570 I disagree with "full three-dimensional radiative transport" as it is far from being full considering the limited number of streams and other remaining approximations.

You are certainly also right with that. For the revised version, we have thus modified the corresponding sentence as follows: "In contrast to these results, however, the dynamic TenStream result features horizontal transport of radiative energy, resulting in much more realistically distributed heating rates and net surface irradiance patterns."

**Technical corrections**

- First paragraph of Introduction, I would also mention the importance of surface fluxes and not just heating rates.

Thanks for pointing this out. We have changed the corresponding sentence to: "They are quantified by heating rates and net surface irradiances and are calculated using radiative transfer models, which describe the transport of radiative energy through Earth's atmosphere, ideally allowing for full three-dimensional (3D) transport of energy."

- L.39-40, add "in the solar spectral range"?

We changed that as suggested.

- 2.1 title: I think you describe more than the TenStream "solver"; you describe the underlying radiative transfer "model". Would it be fair to say that this same model can either be solved as in the original TenStream solver, or as in the Dynamic TenStream?

Yes, that is certainly a good point. We have changed the title of Sect. 2.1 to "The original TenStream model".

-L.88 I was bothered by the use of "transmittance" here as the a-coefficient also account for incoming scattering and I thought that transmittance was defined as the complementary to extinction along a given line sight; but I might be wrong.

You are probably right that transmittance just refers to the complementary of extinction along a given line of sight. Thus, we have changed the corresponding sentence as follows: "While the "a"-coefficients describe the transport of diffuse radiation, the "b"-coefficients quantify the fraction of direct radiation that gets scattered, thus providing a source term for the ten diffuse streams."

- In Fig. 3, it took me some time to understand that horizontal arrows between horizontally adjacent grid-boxes, as well as one of the two vertical downwelling arrows between vertically adjacent grid-boxes, represent direct solar radiation propagation. It might be worth it to mention it in the caption or to distinguish them somehow or maybe remove them from the schematics?

We added additional information to the caption of Fig. 3 clarifying that it visualizes both direct and diffuse streams: "Two-dimensional schematic illustration of the first four steps of a Gauß-Seidel

iteration, showing both diffuse and direct TenStream fluxes in case of Sun shining from the west or left-hand side."

Having these direct streams in Fig. 3 is crucial to understand the iteration direction through the domain, which is why we leave them in the figure. We also decided against distinguishing them colorwise, as we really want to focus on the information whether fluxes are updated or not and not distract the reader from that by adding another color.

- In Fig.5, consider using a more contrasted color palette for the various circles?

The color palette in Fig. 5 was chosen so that it matches the shade of blue used in various other plots such as Fig. 2. Using a darker blue as base color does not add significantly more contrast to the plot, which is why we decided to stay with that color scheme.

- Figs. 6-9 are impossible to read for color-blind people.

We invested a lot of time and tried a wide range of different color palettes to make Figs. 6-9 as accessible to color-blind people as possible. In the end, these colors achieved the best results in the Coblis color blindness simulator referred to at the GMD website, while still providing a pleasant experience for people without color deficiencies. In fact, the plots should be able to read even for people with a monochromatic color blindness, as we use different line styles for the different solvers (solid for the delta-Eddington solver, dashed for the original TenStream solver and dash-dotted lines for the dynamic TenStream solver) and different levels of brightness for the different radiation time steps, making every line in the plot unique. We are aware that the plots are certainly still not ideally suited for color-blind people, but in the end they offered the best trade-off between readability for people without major color deficiencies and color-blind people that we could find.

- L.448 "the dynamic TenStream solver overestimates thermal heating rates" is not very clear here, do you mean overestimates their magnitude knowing that they are negative (i.e. they are more negative than the classical TenStream)?

Exactly. For the revised version, we clarified that we refer the magnitude of the thermal heating rates here: "But in contrast to the solar spectral range, these heating rates get more negative the less the dynamic TenStream solver is called, so that the dynamic TenStream solver overestimates the magnitude of these thermal heating rates when compared to the original TenStream solver it is based on."

- L.548 I was bothered by the use of "emission" here as I think it might be confusing; consider sticking to "flux" or "irradiance"?

That is a good point. We changed that for the revised version: "This also leads to a very distinct pattern of strongly negative and not so negative net surface irradiance areas at the ground in the 1D results, whereas the net surface irradiance is almost uniform in the MYSTIC benchmark result.".

- Page 27, why not use the more precise term of quadrature point instead of bands?

Thank you for the suggestion. We used the term "spectral bands" instead of "quadrature points" as it seemed easier to understand for a general audience, but given that "quadrature points" is the usual term used in the literature, we have changed that for the revised version.

**Response to Community Comment 1 (CC1)**

**Manuscript:** egusphere-2023-2129
**Title:** A dynamic approach to three-dimensional radiative transfer in numerical weather prediction models: the dynamic TenStream solver v1.0
**Authors:** Richard Maier, Fabian Jakub, Claudia Emde, Mihail Manev, Aiko Voigt, and Bernhard Mayer

We thank Chiel van Heerwaarden and his group for their comments on our manuscript, which we will respond to below. To structure our response, Chiel's comments are printed on a gray background color, while our answers are displayed on ordinary white background.

I am writing this comment on behalf of our research group that works on understanding interactions between clouds, radiation, and the land surface. One of our main research topics is developing and using large-eddy simulations with coupled 3D radiation, and for that reason, we studied this paper together with great interest. Let me start by congratulating the authors with their paper. The Munich group has pioneered the coupling of large-eddy simulations with 3D radiation with their TenStream solver, and this method is a very interesting further development of the method. Based on our group discussion, we would like to share two suggestions that could help in improving the paper.

**Suggestion 1: comparison to alternatives to n-stream methods**

It would be nice if the authors could extend their introduction by adding some discussion on alternative methods to the TenStream solver. We believe that in recent years, there has been significant progress in ray tracing of large-eddy simulation fields of cloudy boundary layers, with the papers of Najda Villefranque and colleagues (JAMES, 2019) and Jake Gristey and colleagues (JAS, 2020, GRL 2020) as prominent examples. Also, in our group, we developed a GPU ray tracer, which we coupled to our large-eddy simulation code to study the evolution of shallow cumulus clouds (Veerman et al., 2022, GRL) inspired on earlier work by the Munich group. Then, the recent work of Du an Stechmann (JCP, 2023) on spectral element modeling looks rather promising as well, although coupling with cloud-resolving models remains future work there. To conclude, a more elaborate comparison of n-stream solvers to ray tracing and spectral elements methods could help the reader understand why the authors believe their method is the way to bring 3D radiation to operational weather prediction models.

Thank you for these suggestions. In the preprint version, we kept the introduction rather short. But you are certainly right that we should probably summarize the current state of research on 3D radiative transfer more thoroughly. Therefore, we thankfully used the papers you provided to rewrite our introduction for the revised version, providing a more thorough coverage of the current state of research:

"To account for these increasingly important effects, a lot of effort in recent years was put into making 3D radiative transfer models computationally more feasible. Targeted towards subgrid-scale 3D effects, the Speedy Algorithm for Radiative Transfer through Cloud Sides (SPARTACUS; Schäfer et al. (2016); Hogan et al. (2016)) for example provides a fast method to calculate 3D radiative effects at the resolutions of currently employed global atmospheric models. To this end, it introduces additional terms to the well-established two-stream scheme to account for the radiative transport between cloudy and clear regions inside an individual model column. On the other hand, a lot of work went into the speed-up of inter-column radiative transport at subkilometer-scale

resolutions, where model grid boxes can be gradually treated homogeneously. A large group of these models simplifies the expensive angular part of 3D radiative transfer calculations by just using a discrete number of angles (e.g., Lovejoy et al., 1990; Gabriel et al., 1990; Davis et al., 1990). Most recently, the TenStream solver (Jakub and Mayer, 2015) built upon this idea. It is capable of calculating 3D radiative fluxes and heating rates in both the solar and the thermal spectral range. To do so, it extends the 1D two-stream formulation to ten streams to consider horizontal transport of energy. Besides the TenStream solver, the Neighboring Column Approximation (NCA; Klinger and Mayer (2016, 2020)) provides a fast analytical method for calculating inter-column 3D heating rates in the thermal spectral range. For that purpose, it estimates cloud side effects by taking just the direct neighbors of a specific grid box into account. Apart from these two approaches, significant progress has also been made in accelerating highly accurate 3D Monte Carlo solvers for the use in LES models, with Veerman et al. (2022) for example speeding up the method through the use of graphics processing units (GPUs). This allowed them to perform LES simulations driven by a full Monte Carlo solver for the first time ever. However, despite all these efforts, all of these solvers are still too slow to be used operationally. For example, the GPU-accelerated Monte Carlo solver of Veerman et al. (2022) is at least 6.4 times slower than the two-stream model they compare it to. And even while specifically designed for the use in NWP models, the SPARTACUS model is still 5.8 times slower than the McICA paramerization currently used at ECMWF (Hogan and Bozzo, 2018). This high computational burden prohibits the use of all of these models in operational forecasting, especially given that radiation is already called far less often than the dynamical core of NWP models."

**Suggestion 2: discussion on memory usage of the solver**

The authors present a very extensive performance analysis of their method, which shows that they can deliver an excellent speed up with respect to the original TenStream solver. This in itself is a great result, and the way this is achieved – keeping the fluxes in memory – is clever, because it removes the need for global communication and for a linear system solver. The description omits, however, a discussion on the most impactful consequence of keeping fluxes in memory, namely memory usage. We did a back of the envelope calculation: if every flux (10 diffuse, 3 direct, 10 thermal) for every quadrature point needs to be kept in memory, and one uses a set of 54 (SW) and 67 (LW) quadrature points, then the dynamic TenStream solver requires (10+3) * 54 + 10 * 67 = 1372 permanent three-dimensional fields for the solver. While the authors discuss in the final sections the benefit of smaller quadrature-point sets, the exact memory footprint of the dynamic solver with respect to the original TenStream is not discussed. We believe this number is very relevant if the ultimate aim is to include this solver in an operational weather model.

You are absolutely right that we have to save these 1372 three-dimensional flux fields in order for the dynamic TenStream solver to work. However, we do currently not keep these fluxes in memory all the time, but dump them to the hard drive after calculating a spectral band, so that we only have one three-dimensional field of fluxes in the memory at the same time.

Memory usage is thus currently not dominated by storing these 3D fields in memory, but rather by the look-up tables, which we keep in memory all the time in order to be able to quickly access the TenStream coefficients when performing the Gauß-Seidel iterations.

Nevertheless, introducing a time-stepping scheme in contrast to calculating radiation from scratch will always be more memory consuming on the downside.

As we already mentioned in the response to Review Comment 1, a thorough analysis of the computational demands of our new solver however was never within the scope of this paper. The only point we wanted to make in that regard is that by design incomplete solves lead to a noticeably increase in computational speed. The numbers in Sect. 3.1 were just supposed to give a rough estimation of how fast the new solver is. A detailed investigation of computational speed and memory usage would require a much more thorough analysis of the computational aspects of the solver, whereas the paper is mainly concerned with demonstrating the feasibility of our new method. That is why we decided not to include more detailed computational aspects into the paper.

**Response to Chief Editor Comment 1 (CEC1)**

**Manuscript:** egusphere-2023-2129
**Title:**     A dynamic approach to three-dimensional radiative transfer in numerical weather prediction models: the dynamic TenStream solver v1.0
**Authors:**   Richard Maier, Fabian Jakub, Claudia Emde, Mihail Manev, Aiko Voigt, and Bernhard Mayer

We thank Juan Antonio Añel for his comment on our manuscript, which we will respond to below. To structure our response, Juan's comment is printed on a gray background color, while our answer is displayed on ordinary white background.

Dear authors,

After checking your manuscript, it has come to our attention that it does not comply with our Code and Data Policy.

https://www.geoscientific-model-development.net/policies/code_and_data_policy.html

You have archived your code in a web page (libtradtran.org) that does not comply with our trustable permanent archival policy. Therefore, you have to publish your code in one of the appropriate repositories according to our policy. In this way, you must reply to this comment with the link to the repository used in your manuscript, with its DOI. The reply and the repository should be available as soon as possible and before the Discussions stage is closed. Also, you must include in a potentially reviewed version of your manuscript the modified 'Code and Data Availability' section and the DOI of the code.

Please note that if you fail to comply with this request, we will have to reject your manuscript for publication. Actually, your manuscript should not have been accepted in Discussions, given this lack of compliance with our policy.

Juan A. Añel
Geosci. Model Dev. Executive Editor

Whilst the paper was still in discussion, we uploaded libRadtran version 2.0.5.1 including the new dynamic TenStream solver to Zenodo: https://zenodo.org/records/10288179 (DOI: 10.5281/zenodo.10288179). Furthermore, we changed the "Code and Data Availability" section of the paper as follows for the revised version:

"The newly developed dynamic TenStream solver presented in this paper was developed as part of the libRadtran library for radiative transfer (Emde et al., 2016) and can be accessed via Mayer et al. (2023). Its user manual can be found in the "doc" folder of the library. The shallow cumulus cloud time series used to evaluate the performance of the new solver has been published by Jakub and Gregor (2022), with the modifications and methods applied to it to reproduce the results of Sect. 4 described in Sect. 3 of this paper."

where Mayer et al. (2023) refers to:

Mayer, B., Emde, C., Gasteiger, J., Kylling, A., Jakub, F., and Maier, R.: libRadtran – library for radiative transfer – version 2.0.5.1, https://doi.org/10.5281/ZENODO.10288179, Zenodo [code], 2023.

---

## Referee Report (RR1)

Minor comments for revised version of:
*"A dynamic approach to three-dimensional radiative transfer in subkilometer-scale numerical weather prediction models: the dynamic TenStream solver v1.0"*

Could you try to measure the number of floating point operations? This would give a direct measure of the cost. Reporting just the speed against the ten-stream implementation in Libradtran (a non operational code) is not that informative. Whether dynamic TenStream becomes a viable contender for NWP models will largely depend on how fast it eventually is, which depends both on the cost (number of FLOPs) and whether it can effectively exploit hardware (FLOPs per second). I agree with leaving this second aspect for a future paper to address but the number of floating point operations could easily be measured with GPTL (https://github.com/jmrosinski/GPTL) or other timing library built with PAPI.

Regarding the second aspect, I am in fact a bit concerned that the Gauss Seidel implementation, even if divided into subdomains, won't be able to exploit SIMD vectorization on CPU's. (Perhaps GPU's could be a better fit). You need not address this in the paper but do you think it would be a lot of work to write a Jacobi implementation for dynamic TenStream in the future in order to explore which gives better speed/accuracy trade-off on different hardware? I understand its convergence speed is worse but it could turn out to be a reasonable trade-off if it allows SIMD vectorization and Gauss-Seidel doesn't, especially as CPU hardware is moving towards longer vector lengths with AVX-512 instructions being supported by newer CPU's.

Line 66. SPARTACUS was not designed for NWP specifically, in the original Hogan and Shonk (2013) paper the authors actually talk more about climate models. The proliferation of (sub-) kilometer-scale NWP models arguably makes SPARTACUS more relevant for climate rather than NWP.

Line 67. This speed comparison is slightly out of date since SPARTACUS was recently sped-up by ~3x via code optimization. A better comparison might be to TripleClouds, it's fully 1D-counterpart, and to say it's 3-5 times more slower than TripleClouds (citing Fig. 3 in Ukkonen and Hogan, 2024). However, this is nitpicking sightly and it's also OK to leave the older comparison (optimized SPARTACUS against an optimized McICA could still still be ~6x slower for all I know).

References:

Hogan, R. J., & Shonk, J. K. (2013). Incorporating the effects of 3D radiative transfer in the presence of clouds into two-stream multilayer radiation schemes. *Journal of the Atmospheric Sciences*, *70*(2), 708-724.
Ukkonen, P., & Hogan, R. J. (2024). Twelve Times Faster yet Accurate: A New State-Of-The-Art in Radiation Schemes via Performance and Spectral Optimization. *Journal of Advances in Modeling Earth Systems*, *16*(1), e2023MS003932.

---

## Author Response (AR2)

**Response to the Referee Comments**

**Manuscript:** egusphere-2023-2129
**Title:** A dynamic approach to three-dimensional radiative transfer in subkilometer-scale numerical weather prediction models: the dynamic TenStream solver v1.0
**Authors:** Richard Maier, Fabian Jakub, Claudia Emde, Mihail Manev, Aiko Voigt, and Bernhard Mayer

**Response to Referee #1**

We thank Peter Ukkonen for his comments on our manuscript, which we will respond to below. To structure our response, his comments are printed on a gray background color, while our answers are displayed on ordinary white background.

> Could you try to measure the number of floating point operations? This would give a direct measure of the cost. Reporting just the speed against the ten-stream implementation in Libradtran (a non operational code) is not that informative. Whether dynamic TenStream becomes a viable contender for NWP models will largely depend on how fast it eventually is, which depends both on the cost (number of FLOPs) and whether it can effectively exploit hardware (FLOPs per second). I agree with leaving this second aspect for a future paper to address but the number of floating point operations could easily be measured with GPTL (https://github.com/jmrosinski/GPTL) or other timing library built with PAPI.

By using GPTL, we were indeed able to quantify the number of floating point operations for all the solvers we use when applied to the very first time step of our shallow cumulus time series – except for the MYSTIC solver, whose timing would take a very long time:

|  | **solar spectral range** | **thermal spectral range** |
|---|---|---|
| **δ-Eddington**
*1D two-stream solver* | $4.66 \cdot 10^{10}$ FLOPS (x 1.00) | $6.77 \cdot 10^{10}$ FLOPS (x 1.00) |
| **dynamic TenStream**
*incomplete 3D solver with two Gauß-Seidel iterations* | $2.20 \cdot 10^{11}$ FLOPS (x 4.72) | $2.78 \cdot 10^{11}$ FLOPS (x 4.11) |
| **original TenStream**
*full 3D solver* | $1.79 \cdot 10^{12}$ FLOPS (x 38.44) | $1.11 \cdot 10^{12}$ FLOPS (x 16.46) |

However, we have decided to not include these numbers into the paper itself as GPTL printed out a few error messages during these tests that we were not able to get rid of. Hence, we are not completely sure how reliable these numbers are, although they in general confirm the relative speed numbers reported in Table 1 of the paper.

Besides that, we think that the number of floating-point operations is not an ideal measure of speed anyway. One of the primary computational expenditures in the (dynamic) TenStream solver for example is the retrieval of the TenStream coefficients from the corresponding look-up tables, which is mainly limited by memory bandwidth. On the other hand, calculating the Eddington coefficients in the δ-Eddington approximation is definitely faster, but features a significant amount of floating-point operations, indicating that even the number of floating-point operations is not an objective measure of performance.

So as much as we understand the desire to have an objective measure of how fast our dynamic TenStream solver actually is, providing such a number is a difficult task. And as we already noted in

the initial reply to your review, the main aim of this manuscript is to demonstrate the feasibility of the concepts behind the dynamic TenStream solver and not to compare its performance to operational radiation codes. The only statement we wanted to make in terms of speed is that a solver using incomplete solves is pretty fast by its design, as it just performs a fraction of the computations a full solve does.

Regarding the second aspect, I am in fact a bit concerned that the Gauss Seidel implementation, even if divided into subdomains, won't be able to exploit SIMD vectorization on CPU's. (Perhaps GPU's could be a better fit). You need not address this in the paper but do you think it would be a lot of work to write a Jacobi implementation for dynamic TenStream in the future in order to explore which gives better speed/accuracy trade-off on different hardware? I understand its convergence speed is worse but it could turn out to be a reasonable trade-off if it allows SIMD vectorization and Gauss-Seidel doesn't, especially as CPU hardware is moving towards longer vector lengths with AVX-512 instructions being supported by newer CPU's.

Implementing the Jacobi method into the dynamic TenStream solver is actually a pretty straightforward task and has already been done (but is not used by default).

Multicolor or red-black Gauß-Seidel/SOR solvers allow to use shared memory parallelization and SIMD instructions and should also allow for reasonable vector lengths for GPU processing, albeit again reducing convergence speed. We agree that if a host model is targeting accelerators or vector machines one should investigate the respective performance.

Line 66. SPARTACUS was not designed for NWP specifically, in the original Hogan and Shonk (2013) paper the authors actually talk more about climate models. The proliferation of (sub-) kilometer-scale NWP models arguably makes SPARTACUS more relevant for climate rather than NWP.

Thanks for pointing this out. By quantifying sub-grid scale 3D effects, SPARTACUS is indeed also relevant for climate models. But since especially global-scale NWP models still operate at resolutions where individual grid boxes contain both cloudy and cloud-free regions, these sub-grid scale 3D effects may also still play a role at the resolutions of currently employed NWP models. Hence we decided to change the expression "specifically designed for the use in NWP models" to "specifically designed for the use in large-scale models" for the second revision of the manuscript, as it was used in the introduction of the original SPARTACUS paper (Schäfer and Hogan, 2016). However, this change is no longer relevant due to the response to the following comment.

Line 67. This speed comparison is slightly out of date since SPARTACUS was recently sped-up by ~3x via code optimization. A better comparison might be to TripleClouds, it's fully 1D-counterpart, and to say it's 3-5 times more slower than TripleClouds (citing Fig. 3 in Ukkonen and Hogan, 2024). However, this is nitpicking sightly and it's also OK to leave the older comparison (optimized SPARTACUS against an optimized McICA could still still be ~6x slower for all I know).

References:
Hogan, R. J., & Shonk, J. K. (2013). Incorporating the effects of 3D radiative transfer in the presence of clouds into two-stream multilayer radiation schemes. Journal of the Atmospheric Sciences, 70(2), 708-724.
Ukkonen, P., & Hogan, R. J. (2024). Twelve Times Faster yet Accurate: A New State-Of-The-Art in Radiation Schemes via Performance and Spectral Optimization. Journal of Advances in Modeling Earth Systems, 16(1), e2023MS003932.

Thanks for pointing out this new, very interesting paper. In that regard, the speed comparison is indeed a bit outdated. Since we primarily wanted to refer to the currently relatively slow speed of

inter-column 3D radiative transfer solvers, we decided to get rid of this comparison for the second revision of the paper, because SPARTACUS is addressing sub-grid scale rather than inter-column 3D radiative effects and its significant speed-up is making the comparison pointless.

**Response to Anonymous Referee #3**

We thank Anonymous Referee #3 for his or her comments on our manuscript, which we will respond to below. To structure our response, the referee's comments are printed on a gray background color, while our answers are displayed on ordinary white background.

I am satisfied with the authors' responses to my previous review. I feel that they have made a real effort to understand my questions and to take into account my suggestions when relevant (as well as those from other reviewers). In addition to minor changes, the revised version of the manuscript includes a new section with an analysis of the behaviour of the scheme as a function of the number of Gauss-Seidel iterations; a welcome addition! Some of my questions were not answered in the manuscript because the authors considered them out of scope, which I found very reasonable. I can only hope that their next papers will investigate these compensating errors in the thermal, or the idea of advecting flux fields along with the other meteorological fields, or accounting for subgrid heterogeneity... In the meantime, I strongly recommend publication - after the four remaining occurences of "almost perfectly" are removed!

Thank you for your positive response on the revised version of our manuscript. For the second revision, we have also removed the four remaining occurrences of "almost perfectly". In particular, the following changes were made:

The sentence starting in l. 660 of the revised version was changed to: "At first glance, we can see that the results for the new solver are very similar to those obtained by the original TenStream solver in panel (b), even when operated at the low calling frequency of 60 s."

In l. 703, we modified the corresponding sentence this way: "In summary, we can hence say that for both the solar and the thermal spectral range, dynamic TenStream is able to visually almost reproduce the results obtained by the original TenStream solver, even when operated at lower calling frequencies."

The sentence starting in l. 739 was changed to: "In terms of heating rates, we saw that our new solver is almost able to reproduce the results of the original TenStream solver, even when operated at lower calling frequencies."

And finally, we also modified the sentence starting in l. 762 to avoid the phrase "almost perfectly": "de Mourgues et al. (2023) for example showed that in the thermal spectral range, even 30 quadrature points are sufficient to calculate heating rates that are very similar to those obtained by a line-by-line calculation."

**References:**

- Schäfer, S. A. K., Hogan, R. J., Klinger, C., Chiu, J. C., and Mayer, B.: Representing 3-D cloud radiation effects in two-stream schemes: 1. Longwave considerations and effective cloud edge length, Journal of Geophysical Research: Atmospheres, 121, 8567–8582, https://doi.org/10.1002/2016jd024876, 2016.